# Loops in 4+1d Topological Phases

Xie Chen[1,2], Arpit Dua[1,2], Po-Shen Hsin[1,2,3], Chao-Ming Jian[4], Wilbur Shirley[1,5], Cenke Xu[6]

[1] *Department of Physics and Institute for Quantum Information and Matter,*

*California Institute of Technology, Pasadena, CA 91125, USA*

[2] *Walter Burke Institute for Theoretical Physics,*

*California Institute of Technology, Pasadena, CA 91125, USA*

[3]*Mani L. Bhaumik Institute for Theoretical Physics,*

*475 Portola Plaza, Los Angeles, CA 90095, USA*

[4] *Department of Physics, Cornell University, Ithaca, NY 14853, USA*

[5] *School of Natural Sciences, Institute for Advanced Study, Princeton, NJ, USA*

[6] *Department of Physics, University of California, Santa Barbara, CA 93106, USA*

2+1d topological phases are well characterized by the fusion rules and braiding/exchange statistics of fractional point excitations. In 4+1d, some topological phases contain only fractional loop excitations. What kind of loop statistics exist? We study the 4+1d gauge theory with 2-form $\mathbb{Z}_2$ gauge field (the loop only toric code) and find that while braiding statistics between two different types of loops can be nontrivial, the self 'exchange' statistics are all trivial. In particular, we show that the electric, magnetic, and dyonic loop excitations in the 4+1d toric code are not distinguished by their self-statistics. They tunnel into each other across 3+1d invertible domain walls which in turn give explicit unitary circuits that map the loop excitations into each other. The $SL(2,\mathbb{Z}_2)$ symmetry that permutes the loops, however, cannot be consistently gauged and we discuss the associated obstruction in the process. Moreover, we discuss a gapless boundary condition dubbed the 'fractional Maxwell theory' and show how it can be Higgsed into gapped boundary conditions. We also discuss the generalization of these results from the $\mathbb{Z}_2$ gauge group to $\mathbb{Z}_N$.

**CONTENTS**

## I. INTRODUCTION

Fractional topological orders in 2+1d are well characterized by the universal properties of the fractional point excitations in the system: possible species of such excitations, how they fuse into each other, their exchange and braiding statistics, etc [1–3]. These features are captured by the mathematical framework of unitary tensor categories [1, 4, 5] and substantial progress has been made towards their classification [6, 7], hence the classification of 2+1d topological order. In 3+1d, apart from fractional point excitations, topological phases also have fractional loop excitations. Braiding processes among loops [8] can lead to nontrivial

statistics, but the statistics can always be reduced to or interpreted in some way as statistics among point excitations.

In 4+1d, there exist topological phases with only loop excitations and no point excitations. The 4+1d all-loop toric code [9] is one example. This model has been studied extensively as an error correction code [10, 11]. In this paper we focus on the physical properties of this model. In particular, we want to know what kind of intrinsic loop statistics exist when there are no point excitations, and how the statistics are similar to or different from that among point excitations in 2+1d.

The 4+1d all-loop toric code is very similar to the well-known 2+1d toric code [1] in certain ways. As lattice models, they are both exactly solvable stabilizer models with dual Pauli $X$ and Pauli $Z$ Hamiltonian terms. From this analogy, it is tempting to guess that the statistics are very similar too: a $(-1)$ mutual braiding statistics between the electric and magnetic excitations and a fermionic self-statistics for the composite dyonic excitation. While the first part of the conjecture can be easily checked for the 4+1d toric code, the second part turns out not to be true. Valuable insight can be gained from the field theory description of the model which is given by a 4+1d Chern-Simons theory of two *2-form* $U(1)$ gauge fields [12–14]. Field theory calculations, as discussed in this paper, show that any correlation function of the dyonic membrane operator can be written in terms of a surface integral over the membrane, indicating the triviality of any self-statistics. This is in contrast to the 2+1d Chern-Simons theory of two *1-form* $U(1)$ gauge fields which describes the 2+1d toric code where a nontrivial dyonic correlation function has invariant meaning and gives rise to fermionic self-statistics. We further show in the paper that in 4+1d the electric, magnetic, and dyonic loop excitations can tunnel into each other across invertible domain walls in the bulk of the system. The domain walls, if interpreted as a time direction boundary, give the explicit unitary circuits that map the loop excitations into each other. This is similar to the $\mathbb{Z}_2$ electromagnetic duality symmetry in 2+1d $\mathbb{Z}_2$ gauge theory that exchanges the electric and magnetic particles (but unlike in our case, the symmetry leaves the dyon invariant), or the $S_3$ permutation symmetry in the three-fermion theory in 2+1d that permutes the three fermion particles [15].

The 4+1d $\mathbb{Z}_2$ 2-form gauge theory can be generalized to the 4+1d $\mathbb{Z}_N$ 2-form gauge theory [12–14]. Many properties and results carry over from the $\mathbb{Z}_2$ gauge theory, but the nature of gapped boundaries and loop permutation symmetry is affected by the parity of $N$. For odd

$N$, different gapped boundaries correspond to condensates of different loop excitations while for even $N$ this is further enriched by a sub-type related to difference in framing dependence. Correspondingly, for odd $N$ the bulk has a $SL(2, \mathbb{Z}_N)$ loop permutation symmetry [16, 17] while for even $N$ this symmetry has an obstruction to gauging unless properly extended.

Similar to the gapless boundary of Abelian Chern-Simons theory in 2+1d where the chiral edge modes originate from the bulk gauge parameter (see *e.g.* [18, 19]), the $\mathbb{Z}_N$ 2-form gauge theory in 4+1d also has a natural gapless boundary where the boundary degrees of freedom comes from the 1-form gauge parameter in the bulk. The chiral boson theory on the boundary of the 2+1d Abelian Chern-Simons theory has operators attached to the Wilson lines in the bulk, and they are not mutually local with each other, with the non-locality reproduced by the braiding of anyons in the bulk. Likewise, the boundary theory of the 4+1d 2-form gauge theory is a $U(1)$ gauge theory, where the basic electric and magnetic particles are not mutually local, and the non-locality reproduces the $\mathbb{Z}_N$ braiding between the electric and the magnetic loop excitations in the bulk. We call such $U(1)$ gauge theory "fractional Maxwell theory". Condensing suitable particles in the boundary theory reproduces the gapped boundaries of the $\mathbb{Z}_N$ 2-form gauge theory.

The paper is structured as follows: Section II discusses the bulk properties of the model, both in terms of the lattice model and field theory. Properties being studied include the ground state degeneracy, the fractional loop excitations, and their possible statistics. We point out that, while it seems from lattice model analysis that self-statistics of the dyonic loop can be nontrivial, field theory analysis makes it clear that it can be trivialized. In Section III, we turn to look at the gapped boundary conditions of the model. We show how the electric, magnetic, and dyonic loops can all condense on the boundary, consistent with the expectation that they are all 'bosonic'. Moreover, similar construction can be used to construct invertible topological domain walls in the bulk, which include the domain wall across which magnetic and dyonic loops can tunnel into each other. Again discussion is carried out both in terms of the lattice model and field theory. In Section IV, we write down the explicit unitary circuit that maps the magnetic loop excitation to the dyonic loop excitation while keeping the ground space invariant. This is achieved by taking the 3+1d Lagrangian of the invertible domain wall, interpreting it as a 4 spatial dimensional transformation, and writing it in terms of $\mathbb{Z}_2$ degrees of freedom on a discrete lattice. In Section V, we discuss the generalization of the 4+1d $\mathbb{Z}_2$ 2-form gauge theory to the 4+1d

$\mathbb{Z}_N$ 2-form gauge theory. In particular, we discuss the difference between even $N$ and odd $N$ in terms of possible gapped boundary types as well as in terms of the generators of the $SL(2,\mathbb{Z}_N)$ loop-permutation symmetry. Section VI discusses the obstruction in gauging the $SL(2,\mathbb{Z}_N)$ loop permutation symmetry in the bulk for even $N$. In Section VII, we discuss a gapless boundary condition which we call the 'fractional Maxwell theory' and show how it can be Higgsed into the gapped boundary conditions discussed in previous sections.

## II.   MODEL AND BASIC PROPERTIES

In this section, we focus on the bulk properties of the 4+1d toric code model. We first discuss the lattice Hamiltonian of the loop-only toric code on the 4+1d hypercubic lattice. We then discuss the loop excitations and their braiding properties. Subsequently we give the low energy description of the 4+1d toric code as a $\mathbb{Z}_2$ 2-form gauge theory. Using this low energy field theory description of the toric code, we show that the self-statistics of all loop excitations and their composites are trivial.

### A.   Lattice Hamiltonian

The 4+1d toric code we consider has only membrane logical operators and no string logical operators. In other words, it has only loop-like excitations and no particle-like excitations. The Hamiltonian of the 4+1d toric code [9] is defined on the 4D hypercubic lattice. A representation of the unit cell of the 4D hypercubic lattice, called the 4-cell, is shown in Fig 1 (a). We denote the coordinates for the 4 spatial directions as $x$, $y$, $z$ and $w$ and denote the $n$-cells for $n = 1, 2, 3$ by the coordinates they are supported on. For example, we label the four types of cubes as $xyz$, $xyw$, $yzw$ and $xzw$, the six types of faces as $xy$, $xz$, $xw$, $yz$, $yw$ and $zw$ and the four types of edges as $x$, $y$, $z$ and $w$. The degrees of freedom are qubits, each supported on one face (a two-dimensional cell) of the hypercubic lattice. The Hamiltonian of this model is written as follows,

$$H = -\sum_e A_e - \sum_c B_c \tag{1}$$

$A_e = \prod_{e \subset f} X_f$ is the $X$-stabilizer term supported on the six faces neighboring an edge $e$ and $B_c = \prod_{f \subset c} Z_f$ is the $Z$-stabilizer term supported on the six faces of a 3-cube $c$. We refer to

the $A_e$ stabilizer terms as the charge terms and to the $B_e$ stabilizer terms as the flux terms. This is a Calderbank-Shor-Steane (CSS) Pauli stabilizer model because each of the Pauli interaction terms are made of purely $X$ or $Z$ Paulis and commute with each other. The 4D hypercubic lattice and the stabilizer generators are illustrated in Fig. 1 (b)-(e). There are four types of edges $x$, $y$, $z$ and $w$ on which the charge terms (the $X$-stabilizers) are supported and four types of cubes $xyz$, $xyw$, $xzw$ and $yzw$ on which the flux terms (the $Z$-stabilizers) are supported.

On a 4D torus, the ground state degeneracy of the 4+1d toric code is $2^6$ and thus, can encode six topological qubits. We do a counting of the relations among stabilizer generators and find the number of topological qubits by subtracting the number of independent stabilizer generators from the total number of physical qubits on the 4-torus. There is a local relation on each 4-cell among the flux terms supported on the eight 3-cells (their product being identity). There is a similar local relation on the dual 4-cell among the charge terms. Besides these local relations, there are global relations involving the cube flux terms on 3 dimensional sub-manifolds on the 4-torus. For example, the product of all $xyz$ flux terms on the 3-torus defined by fixed $w$ and periodic coordinates $x$, $y$ and $z$ is identity. Similar relations hold for the $xyw$, $yzw$ and $xzw$ flux terms and there are dual relations among the charge terms on the 4-torus. A simple counting of the independent number of stabilizers keeping in mind these relations and the total number of physical qubits gives the number of logical qubits to be 6. Alternatively, one can find the degeneracy from the homology computation as explained in the appendix, which applies more generally. The logical operators for the six topological qubits are deformable non-contractible membrane operators.

$$U^E(\Sigma) = \prod_{f \in \Sigma} Z_f, \quad U^M(\tilde{\Sigma}) = \prod_{f \cap \tilde{\Sigma} \neq 0} X_f , \qquad (2)$$

where $M$ is a non-contractible membrane in the original lattice and $\tilde{M}$ is a non-contractible membrane in the dual lattice. A basis can be chosen for them such that for each of the six logical operator pairs, one of them is of pure $X$-type on the $\{\alpha\beta\}$ plane of the original lattice and the other is of pure $Z$ type on the $\{\gamma\delta\}$ plane of the dual lattice, $\{\alpha, \beta, \gamma, \delta\} = \{x, y, z, w\}$. This pair of membrane operators anticommute because the membranes intersect at a single point.

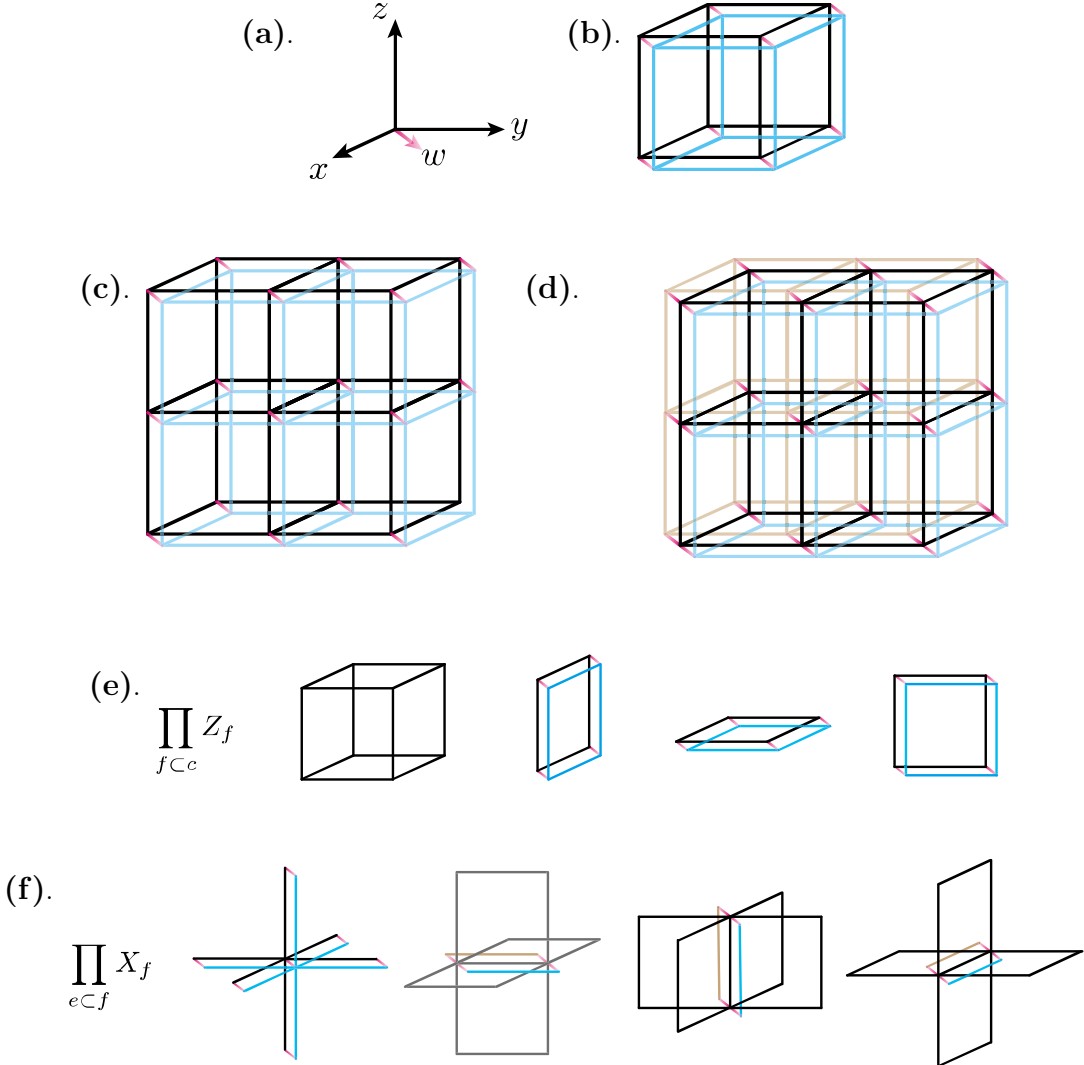

FIG. 1. Hamiltonian of the 4+1d toric code model on 4D hypercubic lattice. (a) Coordinates $x$, $y$, $z$ and $w$ to represent 4 spatial dimensions (b) a representation of the 4-cell using two 3-cubes shifted along $w$ direction. $w = 0$ is shown in black and $w = 1$ is shown in blue and the $w$ edge between $w = 0$ and $w = 1$ is shown in pink. (c) a $2 \times 2 \times 2 \times 2$ lattice using this 4-cell (d) a $2 \times 2 \times 2 \times 3$ lattice; $w = -1$ is added in maroon to the lattice in (c) with the $w$ edge from $w = -1$ to $w = 0$ shown in green. The qubits in the $(2, 2)$ 4+1d toric code live on the 2-cells (e) $Z$ stabilizers supported on the 3-cubes $xzw$, $yzw$, $xyz$ and $xyw$. (f) $X$ stabilizers associated with the edges $w$, $y$, $z$ and $x$.

## B.   Lattice: Loop excitations

All fractional excitations in the 4+1d toric code are loop excitations, created by a truncated membrane operators. The truncated $Z$-membrane and $X$-membrane create different types of loop excitations *i.e.* in different superselection sectors. We call these superselection sectors as $e$ and $m$ respectively. There is a well-defined braiding process in 4+1d for the $e$ and $m$ loops that can be done in a finite region of space such that the loop excitations themselves never intersect. Under this braiding process, the $X$ and $Z$ membrane operators intersect once in space, leading to an overall braiding phase of $(-1)$. In order to find the statistical phase associated with braiding the $e$-loop around the $m$-loop, we start with vacuum, locally create $e$ and $m$ loops far away from each other, grow the $e$-loop in the $xy$ plane at $w < 0$ and the $m$ loop in the $yz$ plane at $w = 0$. Then, we bring the $e$ loop to $w = 0$ such that the two loops are linked at the end of the process. In this process of moving the $e$ loop from $w < 0$ to $w = 0$, the loop excitations never intersect. Next, the $e$-loop can be moved from $w = 0$ to $w > 0$ such that the final configuration is two unlinked loops, $e$ at $w > 0$ and $m$ at $w = 0$. Finally, we reach the vacuum state by shrinking the loop excitations down to vacuum. In another process, we first create the $e$-loop and move it from $w < 0$ to $w = 0$ and then $w > 0$. We then finally annihilate the $e$-loop at $w > 0$. This copies the movement of the $e$ loop in the first process except for the fact that there is no $m$ loop to link with $w = 0$. We then create and grow the $m$ loop at $w = 0$ and annihilate it to vacuum at the same $w$, $w = 0$. In the second processes, the loops are never linked and the membrane operators don't intersect in space (and don't link in spacetime). We quotient the phases obtained in the two processes to obtain the statistical phase associated with the braiding of $e$ and $m$ loops. We illustrate this process in Fig. 2. The overall process leads to the braiding phase of $-1$ due to the nontrivial commutation between the membrane operators of the $e$ and $m$ loop excitations respectively.

In two dimensions, we know that the braiding of quasiparticle excitations in different superselection sectors as well as the self-statistics of a quasiparticle in a given superselection sector are well-defined. For example, the self-statistics of the fermion in the 2D toric code, which is a composite of the $e$ and $m$ particles, can be retrieved by doing a figure eight shape process [20] or a T shape process [21]. A few natural questions arise for the $e$ and $m$ super-selection sectors in the 4D toric code: what is the nature of the dyonic $em$ loop

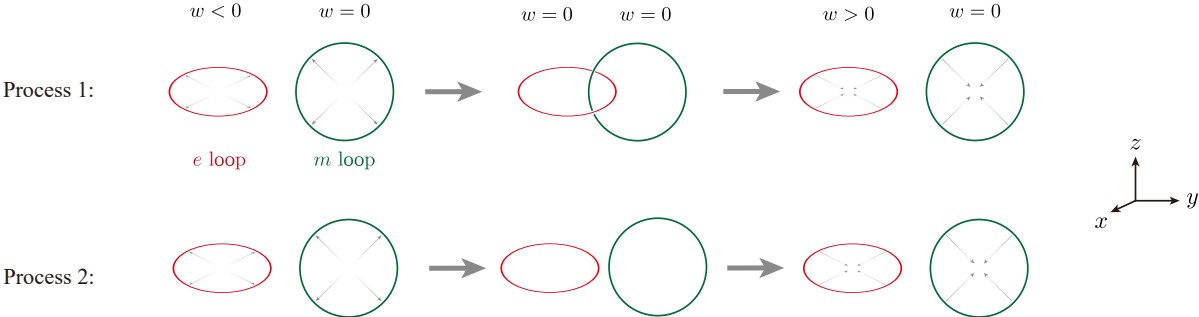

FIG. 2. Braiding $e$ and $m$ loop excitations.

excitation? Does self-statistics make sense for loop excitations in the same way that it does for the point excitations in two dimensions? Naively, it might seem the answer is yes: The membrane operator of the dyonic loop has an $X$ part and a $Z$ part which anti-commute with each other. This is feature led to fermionic statistics in 2+1d. It seems something similar should work in 4+1d as well. In fact, the answer is no and we are going to get some important insight from the field theory formulation of the model.

## C.   Field theory description

The 4+1d $\mathbb{Z}_2$ toric code is described as a $\mathbb{Z}_2$ 2-form gauge theory in 4+1d. A 2-form gauge field $b = \frac{1}{2} b_{\mu\nu} dx^\mu dx^\nu$ (repeated indices are summed over) has two space-time indices $\mu,\nu$, as compared to a vector gauge field $a = a_\mu dx^\mu$ with only one space-time index $\mu$. Moreover, $b$ is anti-symmetric, meaning that $b_{\mu\nu} = -b_{\nu\mu}$. The toric code can be described by $U(1)$ 2-form gauge fields $b^{(1)}, b^{(2)}$ with Lagrangian

$$\mathcal{L} = \frac{2}{2\pi} b^{(1)} db^{(2)} \ . \tag{3}$$

$b^{(1)}$, $b^{(2)}$ take value in $U(1)$. $b^{(1)} db^{(2)}$ is a short-hand notation for $\epsilon^{\mu\nu\tau\eta\lambda} b^{(1)}_{\mu\nu} \partial_\tau b^{(2)}_{\eta\lambda} d^5 x$, $\epsilon$ being the anti-symmetric Levi-Civita symbol. Throughout the paper, we use Greek letters ($\mu$, $\nu$, ...) to label space-time dimensions and English letters ($i$, $j$, ...) to label spatial dimensions. For a simple introduction to differential forms, see appendix A.

A useful comparison to make is with the 2+1d theory

$$\mathcal{L}_{2+1D} = \frac{2}{2\pi} a^{(1)} da^{(2)} \tag{4}$$

with 1-form (vector) $U(1)$ gauge fields $a^{(1)}$ and $a^{(2)}$ which describes the 2+1d toric code. The two theories are similar in many aspects but also different in important ways. We now

briefly review how things work in the 2+1d theory $\mathcal{L}_{2+1D}$. Solving the equations of motion, we find that $a_0^{(m)}$ are Lagrangian multipliers which enforce the flat connection constraints,

$$\partial_x a_y^{(m)} - \partial_y a_x^{(m)} = 0. \tag{5}$$

Here $a_x^{(m)}$ and $a_y^{(m)}$ satisfy canonical equal time commutation relations given by

$$\left[ a_x^{(m)}(x,y), a_y^{(\tilde{m})}(x',y') \right] = (1 - \delta_{m\tilde{m}}) \frac{i2\pi}{2} \delta_{xx'} \delta_{yy'} \tag{6}$$

where $(x,y)$ and $(x',y')$ are space-time coordinates for $a^{(m)}$ and $a^{(m)'}$ respectively. Due to the flat connection constraint, the only dynamical DOF comes from Wilson loop operators around nontrivial spatial cycles $C$,

$$W^{(m)}(C) = \exp(i \oint_C a^{(m)}) \ . \tag{7}$$

Consider the space to be a torus with periodic coordinates $x, y$, and we take one Wilson loop along the nontrivial cycle in $x$ direction and another one in $y$ direction, their commutation can be deduced from that of the $a$ fields to be [1]

$$W_x^{(m)}(C_x) W_y^{(\tilde{m})}(C_y) = e^{i(1-\delta_{m\tilde{m}})\pi} W_y^{(\tilde{m})}(C_y) W_x^{(m)}(C_x) \tag{9}$$

From this relation we see that even though the $a^{(m)}$'s are $U(1)$ gauge fields, $\mathcal{L}_{2+1D}$ describes a $\mathbb{Z}_2$ gauge theory — the 2+1d toric code [12–14]. The commutation relation of the Wilson loops gives the braiding statistics of the anyons in the toric code. The braiding statistics can also be seen in the space-time formulation by calculating the correlation functions (expectation values) of linked space-time Wilson loop configurations

$$\langle W^{(1)}(C) W^{(2)}(\tilde{C}) \rangle = \langle e^{i \oint_C a^{(1)}} e^{i \oint_{\tilde{C}} a^{(2)}} \rangle \ . \tag{10}$$

We briefly review the 2+1d calculation here, which is to be contrasted to the 4+1d case discussed later. Calculating this correlation function is equivalent to modifying the Lagrangian by a source term

$$\mathcal{L}_{C\tilde{C}} = \frac{2}{2\pi} a^{(1)} da^{(2)} + a^{(1)} \delta^{\perp}(C) + a^{(2)} \delta^{\perp}(\tilde{C}) \ , \tag{11}$$

---

[1] Using the Baker–Campbell–Hausdorff formula and the canonical commutation relation, we have

$$e^{i \oint_{C_x} a^{(m)}} e^{i \oint_{C_y} a^{(\tilde{m})}} = e^{i \oint_{C_y} a^{(\tilde{m})}} e^{i \oint_{C_x} a^{(m)}} e^{-\int \int [a^{(m)}(x), a^{(\tilde{m})}(y)] d^2x d^2x' \delta_x(C_x)^{\perp} \delta_{x'}(C_y)^{\perp}}$$

$$= e^{i \oint_{C_y} a^{(\tilde{m})}} e^{i \oint_{C_x} a^{(m)}} e^{\frac{-2\pi i}{2}(1-\delta_{m\tilde{m}}) \int \delta_x(C_x)^{\perp} \delta_{x'}(C_y)^{\perp}} = e^{i \oint_{C_y} a^{(\tilde{m})}} e^{i \oint_{C_x} a^{(m)}} e^{-i\pi(1-\delta_{m\tilde{m}})} \ , \tag{8}$$

where $\delta_x(C)^{\perp}$ is the delta function 1-form that restricts the space integral $\int d^2x$ to $C$.

where $\delta^\perp(C)$ is the delta function 2-form that restricts the space time integral to $C$. The equations of motion for $a^{(1)}$ and $a^{(2)}$ become

$$da^{(2)} + \pi\delta^\perp(C) = 0, \ da^{(1)} + \pi\delta^\perp(C') = 0 \tag{12}$$

As long as $C$ and $\tilde{C}$ are contractible space-time loops, $a^{(1)}$ and $a^{(2)}$ can be solved to be

$$a^{(1)} = -\pi\delta^\perp(\tilde{D}), \ a^{(2)} = -\pi\delta^\perp(D) \tag{13}$$

where $D$ and $\tilde{D}$ are disks with $C$ and $\tilde{C}$ as boundaries and $d\delta^\perp(D) = \delta(C), d\delta^\perp(\tilde{D}) = \delta(\tilde{C})$. Substituting (13) back into the correlation function, we find

$$\langle W^{(1)}(C)W^{(2)}(\tilde{C})\rangle = e^{\int -i\pi\delta^\perp(D)\delta^\perp(\tilde{C})} = e^{-i\pi\mathrm{Link3D}(C,\tilde{C})} \tag{14}$$

Note that the linking number in three dimensions, $\mathrm{Link_{3D}}$ satisfies the property that $\mathrm{Link_{3D}}(C,\tilde{C}) = \mathrm{Link_{3D}}(\tilde{C},C)$.

Similar analysis applies to the 4+1d Lagrangian $\mathcal{L}_{4+1\mathrm{D}} = \frac{2}{2\pi}b^{(1)}db^{(2)}$, but also with crucial differences. The $b_{0i}^{(m)}$ fields are Lagrangian multipliers which enforces the 'flat connection' constraints

$$\sum_{jkl} \epsilon^{jkl}\partial_j b_{kl}^{(m)} = 0 \ . \tag{15}$$

The spatial components of the fields satisfy the following equal time commutation relation,

$$\left[b_{ij}^{(m)}, b_{kl}^{(\tilde{m})}\right] = \epsilon^{ijkl}(m - \tilde{m})\frac{i2\pi}{2} \tag{16}$$

Note that in 2+1d, the commutation relation (Eq. 6) is symmetric under the exchange of $m$ and $\tilde{m}$, while the one in 4+1d is anti-symmetric. This is related to the fact that the 2+1d Lagrangian is symmetric under the exchange of $a^{(1)}$ and $a^{(2)}$ while the 4+1d one is anti-symmetric. We see below that this anti-symmetry leads to the conclusion that self-statistics does not exist in 4+1d although self-statistics is well defined in the closely related 2+1d theory.

Due to the 'flat connection' constraint, the only dynamical DOF comes from the Wilson surface operators around a nontrivial plane $\Sigma$

$$\mathcal{U}_{ij}^{(m)}(\Sigma_{ij}) = \exp\left(i\oint_{\Sigma_{ij}} b_{ij}^{(m)}dx_i dx_j\right) \tag{17}$$

which satisfy the commutation relations

$$\mathcal{U}_{ij}^{(m)}\mathcal{U}_{kl}^{(\tilde{m})} = e^{i\epsilon_{ijkl}(m-\tilde{m})\pi}\mathcal{U}_{kl}^{(\tilde{m})}\mathcal{U}_{ij}^{(m)} \ . \tag{18}$$

Therefore, similar to the 2+1d case, even though the $b^{(m)}$'s are $U(1)$ gauge fields, $\mathcal{L}_{4+1D}$ describes a $Z_2$ gauge theory. The correlation function of linked space-time Wilson surfaces is given by, following a similar derivation as given above,

$$\langle\mathcal{U}^{(m)}(\Sigma)\mathcal{U}^{(\tilde{m})}(\tilde{\Sigma})\rangle = e^{-i\pi(m-\tilde{m})\mathrm{Link}_{5D}(\Sigma,\tilde{\Sigma})} \tag{19}$$

which corresponds exactly to the braiding statistics discussed in section II B. Note that the linking number in five dimensions satisfies the properties that $\mathrm{Link}_{5D}(\Sigma,\tilde{\Sigma}) = -\mathrm{Link}_{5D}(\tilde{\Sigma},\Sigma)$. Note the linking number also equals to the intersection number of $\Sigma$ and the volume $\mathcal{V}'$ that bounds $\tilde{\Sigma}$, and minus the intersection number of $\tilde{\Sigma}$ and the volume $\mathcal{V}$ that bounds $\Sigma$.

When properly discretized, $e^{i\int_f b^{(1)}}$ integrated over a face $f$ becomes the $Z_f$ operator on that face while $e^{i\int_{\tilde{f}} b^{(2)}}$ integrated over the dual face becomes the $X_f$ operator on $f$. Putting $m = 1$ and 2 in Eq. 15 gives the constraints that $B_c = 1$ and $A_e = 1$ for the Hamiltonian terms in Eq. 1 respectively. The Wilson surface operators $\mathcal{U}^{(m)}(\Sigma)$ become the electric and magnetic membrane operators $U^E(\Sigma)$ and $U^M(\tilde{\Sigma})$ for $m = 1$ and 2 respectively. We use $\mathcal{U}^E$ and $\mathcal{U}^M$ in the following discussion to label the Wilson surface operators.

We now see what the field theory formulation tells about self-statistics, in particular whether the dyonic $em$ loop is fermionic. The anti-symmetry of the Lagrangian under the exchange of (1) and (2) indicates that the answer is no because the intuition from 2+1d is that the self-statistics of the $em$ loop is 'half' of the sum of the braiding statistics of $e$ with $m$ and the braiding statistics of $m$ with $e$. In 2+1d, these two are equal, while in 4+1d, they are opposite to each other. Hence the sum, which is intuitively expected to be the self-statistics, becomes zero. We also see this by calculating the correlation function of a dyon surface operator $\langle e^{i\oint_\Sigma b^{(1)}+b^{(2)}}\rangle$. To do this, we need to 'split' the dyon surface operator into an electric part on $\Sigma$ and a magnetic part $\tilde{\Sigma}$. $\Sigma$ and $\tilde{\Sigma}$ are close to each other but do not overlap. That is, we want to calculate $\langle e^{i\oint_\Sigma b^{(1)}+i\oint_{\tilde{\Sigma}} b^{(2)}}\rangle$ which we already know is equal to $e^{-i\pi\mathrm{Link}(\Sigma,\tilde{\Sigma})}$. But there seems to be a sign ambiguity because if we have split the dyon operator so that the electric part is on $\tilde{\Sigma}$ and the magnetic part on $\Sigma$, we get the opposite phase factor. (This may not seem to be an issue for $Z_2$ but will be an issue for $Z_n$.) The way to resolve this ambiguity is to notice that for $\Sigma, \tilde{\Sigma}$ close to each other, we take the volumes

$\mathcal{V}, \tilde{\mathcal{V}}$ that bound $\Sigma, \tilde{\Sigma}$ respectively to be close to each other in the second computation, so the difference

$$\epsilon_2 \equiv \delta(\tilde{\mathcal{V}})^\perp - \delta(\mathcal{V})^\perp \, , \tag{20}$$

is small and has support only near the surface $\Sigma$ (or $\tilde{\Sigma}$, which is close to $\Sigma$). We use $\delta(\Sigma)^\perp = d\delta(\mathcal{V})^\perp$, and $\int \delta(\mathcal{V})^\perp d\delta(\mathcal{V})^\perp = 0$ on orientable spacetime, to write the correlation function computed by the splitting as

$$(-1)^{q_e q_m \int \delta(\tilde{\mathcal{V}})^\perp d\delta(\mathcal{V})^\perp} = (-1)^{q_e q_m \int \left(\epsilon_2 + \delta(\mathcal{V})^\perp\right) d\delta(\mathcal{V})^\perp}$$

$$= (-1)^{q_e q_m \int \epsilon_2 d\delta(\mathcal{V})^\perp} = (-1)^{q_e q_m \int \epsilon_2 \delta(\Sigma)^\perp} = (-1)^{q_e q_m \int_\Sigma \epsilon_2} \, . \tag{21}$$

In other words, the correlation function computed in the second method is given by a local term on the surface! We note that it is important that $\Sigma, \tilde{\Sigma}$ are close for this to be a local term supported on the surface.

We remark that the discussion in (21) would not work in 2+1d, where $\Sigma$ is replaced by a curve $\gamma$ that describes the worldline of particles, and $\mathcal{V}$ is replaced by a surface $S$, and $\delta(\mathcal{V})^\perp$ is replaced by $\delta(S)^\perp$, which a delta function 1-form. Then $\delta(\mathcal{V})^\perp d\delta(\mathcal{V})^\perp$ is replaced by a Chern-Simons term $\delta(S)^\perp d\delta(S)^\perp$, which is not a total derivative, and the correlation function $(-1)^{\int \delta(S)^\perp d\delta(S)^\perp} = (-1)^{\int_\gamma \delta(S)^\perp}$ depends on the surface $S$ that bounds $\gamma$, and it cannot be written as an integral on $\gamma$ of a local density that depends only on $\gamma$ but not on $S$.

What does this observation mean for the lattice model? In the following sections, we find that there is a way to 'dress up' each local piece of the lattice membrane operator of the dyonic loop excitation so that the local pieces commute with each other, like in the $E$ and $M$ membranes, hence the self-statistics becomes trivial. To find out how to 'dress up' the dyon membrane operator, we look at the boundary of the system.

## III. BOUNDARY CONDITION / DOMAIN WALL

The previous section focused on the bulk properties of the 4+1d toric code model, while in this section, we look at gapped boundaries and domain walls. It turns out that the boundaries/domain walls give important insight into the bulk properties of the model and help address the issue about self-statistics of the loop excitations. We explicitly construct different gapped boundary conditions where the $e$, $m$, $\psi$ loops can respectively condense,

indicating the triviality of their self-statistics. Similar constructions can be carried out for domain walls in the bulk such that $m$ and $\psi$ loop can tunnel into each other across the domain wall, further supporting the idea that $m$ and $\psi$ do not differ in self-statistics. We are going to present the discussion in terms of both lattice model and field theory.

## A.   Lattice: smooth and rough boundaries

Similar to toric code in 2+1d [22], two natural gapped boundary conditions for the 4+1d model are the 'rough' and 'smooth' boundary conditions. Suppose that we have a boundary in the $+w$ direction. The Hamiltonian terms on the boundary are shown in Fig. 3. The smooth boundary involves five-body charge terms ($X$ stabilizers) and six-body flux terms ($Z$ stabilizers); the rough boundary involves six-body charge terms ($X$ stabilizers) and five-body flux terms ($Z$ stabilizers).

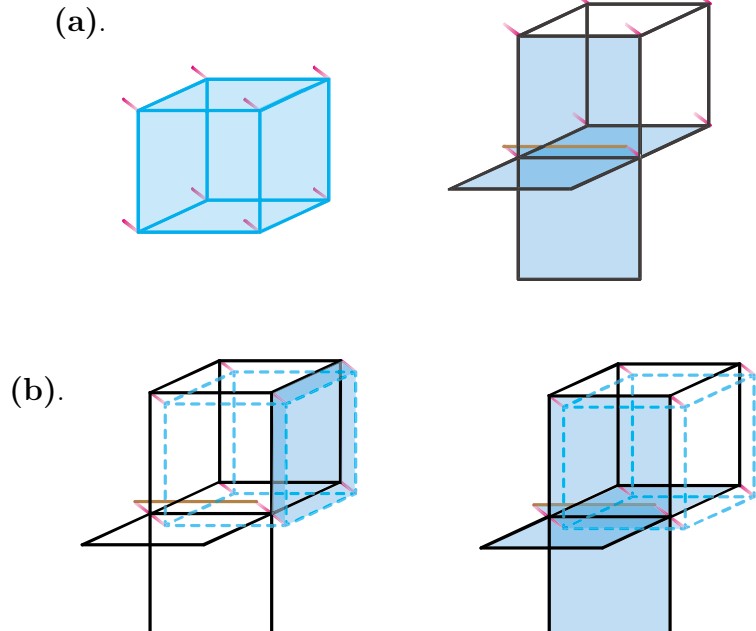

FIG. 3. a) Smooth and b) Rough boundary conditions. The terms on the left are the flux terms ($Z$ stabilizers) which are 6-body for smooth boundary conditions shown in (a) and 5-body for rough boundary conditions shown in (b). The terms on the right are the charge terms ($X$ stabilizers) which are 5-body for smooth boundary conditions shown in (a) and 6-body for rough boundary conditions shown in (b).

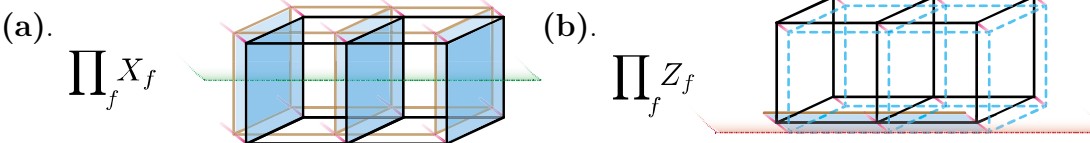

FIG. 4. $e$ loop and $m$ loop condensation. $m$ loop condenses on the smooth boundary and the $e$ loop condenses on the rough boundary.

It is straight forward to check that first all the Hamiltonian terms, either in the bulk or on the boundary, still commute. Therefore the boundaries are gapped. Moreover, the $m$ loop excitation condenses on the smooth boundary. That is, if we create an $m$ loop excitation in the bulk, bring it partially to the boundary as shown in Fig. 4(a), then the part that lies on the boundary disappears (is no longer excitation). The closed loop excitation becomes an open loop excitation with end points on the boundary. The excitations at the endpoints in the 3+1d boundary are confined due to being joined by a string-like excitation in the 4+1d bulk. There is no excitation on the boundary between the end points. If the whole loop is brought to lie on the boundary, then the excitation completely disappears. Similarly, the $e$ loop condenses on the rough boundary as shown in Fig. 4(b).

## B. Lattice: decorated smooth boundary

The smooth boundary can be further 'decorated' to realize new types of gapped boundary conditions. Following [23], in a D+1d gauge theory of gauge group $G$, on the gauge-symmetry-preserving (no gauge charge condensation) boundary we attach a $D-1+1$d gauge theory of group $G$. The $D-1+1$d gauge theory can be twisted, hence giving rise to a variety of boundary types. In Appendix D, we discuss a simpler example of the $\mathbb{Z}_2 \times \mathbb{Z}_2$ gauge theory in 2+1d whose boundary can be attached a 1+1d $\mathbb{Z}_2 \times \mathbb{Z}_2$ twisted gauge theory. Here we have a 4+1d 2-form $\mathbb{Z}_2$ gauge theory, hence the boundary can be attached to a (twisted) 3+1d 2-form $\mathbb{Z}_2$ gauge theory. The "ungauged" version of the theory is an SPT phase with $\mathbb{Z}_2$ 1-form symmetry, which has $\mathbb{Z}_4$ classification [24, 25], and is realized by lattice model in [25]. In the following, we are going to show how to attach the elementary twisted gauge theory among the four to the smooth boundary of the 4+1d toric code and how the $\psi$ loop excitation condenses on it.

The elementary 3+1d twisted 2-form $\mathbb{Z}_2$ gauge theory is equivalent to the 3+1d semionic

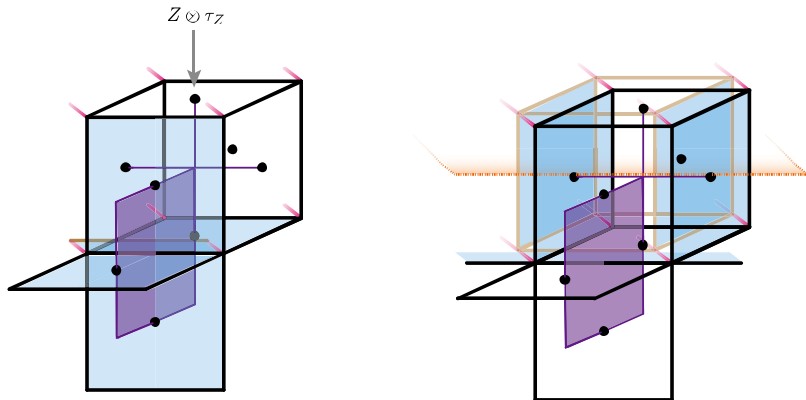

FIG. 5. Decorated smooth boundary. Left figure shows the coupling between smooth boundary and WW domain wall. Right figure shows the condensation of $\psi$ loop on the decorated smooth boundary.

Walker-Wang (WW) model (see appendix for a review of the semionic Walker-Wang model). In the WW model, $\mathbb{Z}_2$ DOFs are on the edges of a 3D lattice and they satisfy the closed loop condition $\prod_{e \supset v} \tau_Z^e = 1$ around each vertex $v$. If we dualize the 3D lattice so that its edges become the plaquettes of dual 3D lattice and vice versa, the $\mathbb{Z}_2$ DOFs would be on plaquettes (of the dual lattice) and they satisfy the condition $\prod_{p \subset c} \tau_Z^p = 1$ around each cube $c$. Hence the DOFs can be interpreted as 2-form $Z_2$ gauge fields that satisfy the zero flux condition. In the following discussion, we are going to use the original formulation of the WW model (DOFs on edges) to be consistent with previous literature on the WW model. To couple this elementary twisted 3+1d 2-form $\mathbb{Z}_2$ gauge theory to the smooth boundary of the 4+1d toric code in the $+w$ direction, we align the two lattices such that each edge in the 3+1d WW model goes through a dual plaquette on the 4+1d boundary, as shown in Fig. 5 (a). The pair of DOFs are coupled with a $Z \otimes \tau_Z$ term. With this coupling, the flux term from the 4+1d boundary becomes equivalent to the flux term from the 3+1d model – they measure the same flux. The electric fields from the two then have to act together. In particular, the plaquette term from the 3+1d WW model needs to be combined with the charge term from the 4+1d boundary, as shown in Fig. 5 (a). After making this coupling, it can be directly checked that all the terms in the model, both in the bulk and on the boundary, still commute with each other and we have a new gapped boundary condition.

Now we see from this lattice construction how the $\psi$ loop can condense on this boundary. Consider a $\psi$ membrane operator extending in the $wy$ plane and terminating on the boundary along a line in the $y$ direction. Due to the $Z \otimes \tau_Z$ coupling on the boundary, the $m$ part

of the membrane – a product of $X$ on $xz$ plaquettes dual to the $wy$ plane (shaded blue plaquettes in Fig. 5 (b)) – now need to be decorated by a string operator in the WW model along the line dual to the $xz$ plaquettes (dashed red line in Fig. 5 (b)). The most natural way to do this in the WW model is to apply a semion string operator, which is a product of $\tau_X$ together with some phase factors in the $\tau_Z$ basis along the line. The decorated $m$ part of the membrane operator commutes with the $Z \otimes \tau_Z$ coupling, commutes with the flux term on the boundary, but anticommutes with the 'decorated' charge term on the boundary because the semion string anticommutes with the plaquette operators in the WW model that it threads. This anti-commutation relation is cancelled by the $e$ part of the membrane operator (shaded orange plaquettes in Fig. 5 (b)) which anticommutes with the charge term on the boundary. The composite $\psi$ membrane operator hence commute with all terms along the line where the membrane terminates on the boundary and the $\psi$ loop excitation condenses on the boundary.

## C.   Lattice construction of $m \leftrightarrow \psi$ domain wall

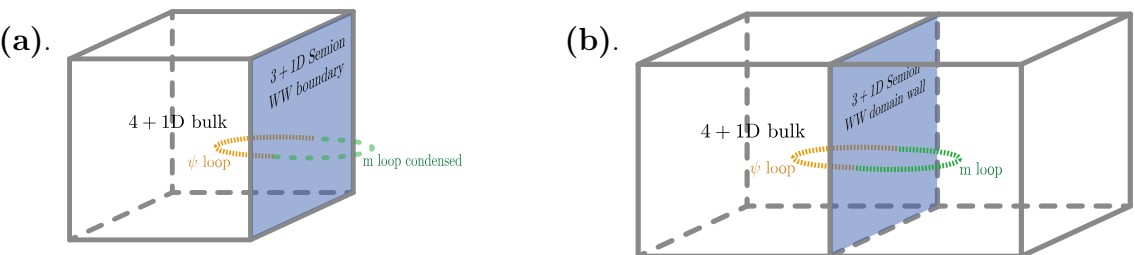

FIG. 6. (a) 3+1d semion WW gapped boundary. (b) 3+1d semion WW gapped domain wall which maps $m$ loop to $\psi$ loop and vice versa.

The construction in the last section can be carried out in a similar way in the bulk of the 4+1d toric code to generate a domain wall such that the $m$ loop excitation tunnels through the domain wall and becomes the $\psi$ loop excitation, and vice versa, as shown in Fig. 6.

Instead of attaching the WW model to the boundary of the 4+1d bulk, we insert it into the middle of the bulk (at a fixed $w$). The folding approach [26] implies that a WW domain wall in the bulk of the 4+1d toric code is equivalent to having a WW gapped boundary for two copies of the 4+1d toric code. We arrange the DOFs from the two models to be aligned in a similar way (edges in the WW dual to plaquettes in the 3+1d submanifold with fixed

$w$ inside the 4+1d bulk) and coupled in a similar way (with a $Z \otimes \tau_Z$ coupling). The flux term from the bulk and the WW model become equivalent to each other. The charge term from the bulk gets attached to the WW plaquette term on the dual plaquette. Now imagine acting with an $m$ membrane operator across the domain wall, intersecting the domain wall on a line in the $y$ direction. The membrane operator gets attached to a semion string on the domain wall. The undecorated $m$ membrane operator commute with all terms but violate the $Z \otimes \tau_Z$ coupling. By attaching the semion string, the $Z \otimes \tau_Z$ coupling is respected but the semion string anti-commutes with the WW plaquette operator it threads on the domain wall. Such anti-commutation can be canceled if we apply a half $e$ membrane operator on one side of the domain wall, terminating along the same $y$ direction line on the domain wall. Therefore, across the domain wall, $m$ and $\psi$ loop excitations map into each other.

### D.   Field theory description of boundary and domain wall

We discussed above the gapped boundaries of the 4+1d toric code lattice model. We now the discuss the gapped boundaries from the perspective of the low energy effective field theory on spacetime $M$, whose action as discussed before, is given by

$$S = \int_M \frac{2}{2\pi} b^{(1)} db^{(2)} . \tag{22}$$

#### 1.   Electric $e$ loop condensed boundary

In the presence of a boundary, the equation of motion for varying $b^{(2)} \to b^{(2)} + \Delta b^{(2)}$ picks up a boundary term:

$$\Delta S = - \int_M \frac{2}{2\pi} db^{(1)} \Delta b^{(2)} + \frac{2}{2\pi} \int_{\partial M} b^{(1)} \Delta b^{(2)} . \tag{23}$$

Thus the equation of motion (requirement that $\Delta S = 0$ for any $\Delta b^{(2)}$) implies that $db^{(1)} = 0$ in the bulk and $b^{(1)} = 0$ on the boundary. Thus the electric surface operator $e^{i \int b^{(1)}}$ becomes trivial on the boundary, indicating the electric loop condenses on the boundary. The boundary is $\mathbb{Z}_2$ 2-form gauge theory with gauge field $b^{(2)}$, it describes a $\mathbb{Z}_2$ topological order with deconfined charges, where the loop excitation is the bulk magnetic loop. To describe the boundary of the $\mathbb{Z}_2$ 2-form gauge theory, we express it as a $U(1)$ 2-form gauge field $b^{(2)}$ and a Lagrangian multiplier $U(1)$ 1-form $a^{(1)}$ with the action $\frac{2}{2\pi} \int_{\partial M} da^{(1)} b^{(2)}$. Then

the equation of motion for $b^{(2)}$, as a $U(1)$ 2-form gauge field, implies $b^{(1)} + da^{(1)} = 0$ on the boundary. The boundary has gauge invariant open surface operator $\int_\Sigma b^{(1)} + \oint_{\partial\Sigma} a^{(1)}$, and thus the electric loop is described by $e^{\oint_{\partial\Sigma} a^{(1)}}$ on the boundary.

Generalizing the Beigi-Shor-Whalen (BSW) construction [23] to 4+1D and 2-form gauge fields, the $e$ condensed gapped boundary of the 4+1D toric code is constructed by coupling to a 3+1D untwisted $Z_2$ topological order which can be the 3+1D toric code with an emergent boson or the 3+1D toric code with an emergent fermion[2] Hence, there are two variants of the $e$ condensed boundary. In field theory this corresponds to adding the boundary term that does not alter the equation of motion mod 2 (up to a background) as follows,

$$\frac{2^2}{4\pi} \int b^{(2)} b^{(2)} = \pi \int \frac{2b^{(2)}}{2\pi} \frac{2b^{(2)}}{2\pi} \ . \tag{24}$$

For $b^{(2)}$ whose holonomy takes value in $0, \pi$ mod $2\pi$, such boundary term is equivalent to $\pi \int \frac{2b^{(2)}}{2\pi} w_2(TM)$ by the Wu formula [29], and the equation of motion for $b^{(2)}$ gives the boundary condition $b^{(1)}| = \pi w_2(TM)$. Since the boundary of $\int w_2(TM)$ has the framing dependence of a fermion, the electric loop on the boundary carries an additional worldline of a spin 1/2 particle. The operator $e^{i \int b^{(1)}}$ can end on the boundary by a closed curve, and this additional term has the effect of changing the particle whose worldline is the closed curve from a boson to a fermion [24]. In other words, the boundary term changes the framing on the condensed $e$ loop on the boundary by attaching to it a worldline of a spin 1/2 particle [24]. We remark that the self-statistics of the loop remains unchanged, since this modification is localized on the loop. The boundary term is also the low energy effective action for the fermion Walker-Wang model.

### 2. Magnetic $m$ loop condensed boundary

We modify the boundary by adding the boundary term

$$-\frac{2}{2\pi} \int_{\partial M} b^{(1)} b^{(2)} \ . \tag{25}$$

The boundary term contributes to the equation of motion under the variation

$$b^{(1)} \to b^{(1)} + \Delta b^{(1)}, \quad b^{(2)} \to b^{(2)} + \Delta b^{(2)} \tag{26}$$

---

[2] The $\mathbb{Z}_2$ topological order in 3+1d is classified in [27, 28].

as

$$-\frac{2}{2\pi}\int_{\partial M}(\Delta b^{(1)}b^{(2)}+b^{(1)}\Delta b^{(2)})\ . \tag{27}$$

The part depending on $\Delta b^{(2)}$ compensates for the variation from the bulk action (Eq. 23), and thus the equation of motion for $b^{(2)}$ is trivially satisfied on the boundary. The part depending on $\Delta b^{(1)}$ implies $b^{(2)}=0$ on the boundary. Thus the magnetic surface operator $e^{i\int b^{(2)}}$ becomes trivial on the boundary, indicating that the magnetic loop condenses on the boundary. The boundary is a $\mathbb{Z}_2$ 2-form gauge theory with gauge field $b^{(1)}$, it describes a $\mathbb{Z}_2$ topological order, where the loop excitation is the bulk electric loop. We describe the boundary $\mathbb{Z}_2$ 2-form gauge theory using $U(1)$ 2-form gauge field $b^{(1)}$ and a Lagrangian multiplier $U(1)$ 1-form gauge field $a^{(2)}$, with the action $-\frac{2}{2\pi}\int_{\partial M}b^{(1)}da^{(2)}$ with 1-form $a^{(2)}$. Then the equation of motion from $b^{(1)}\to b^{(1)}+\Delta b^{(1)}$ gives $b^{(2)}+da^{(2)}=0$ on the boundary. The boundary has gauge invariant open surface operator $\int_{\Sigma}b^{(2)}+\oint_{\partial\Sigma}a^{(2)}$: the magnetic loop on the boundary is $\oint_{\partial\Sigma}a^{(2)}$.

Similar to the $e$ condensed boundary, there are two variants of the $m$ condensed boundary, corresponding to whether the boundary topological order, that is coupled to, is a 3+1D toric code with an emergent boson or an emergent fermion. The obvious difference is that the m-loop at the boundary comes from m-loop of the bulk in the m-boundary case and from e-loop of the bulk in the e-boundary case. In field theory, the coupling corresponds to adding the boundary term that does not alter the equation of motion mod 2:

$$\frac{2^2}{4\pi}\int b^{(1)}b^{(1)}=\pi\int\frac{2b^{(1)}}{2\pi}\frac{2b^{(1)}}{2\pi}\ . \tag{28}$$

The additional term has the effect of attaching to the condensed $m$ loop on the boundary the worldline of a spin 1/2 particle [24].

We remark that the magnetic loop condensed boundary can be obtained from the electric loop condensed boundary by colliding it with the topological domain wall that generates the $e\leftrightarrow m$ duality,

$$S:\quad (b^{(1)},b^{(2)})\to(b^{(2)},-b^{(1)})\ . \tag{29}$$

To see this, we note that such duality domain wall can be described as follows. We insert the wall at $x=0$, and on the left of the wall $x<0$ we perform the $e\leftrightarrow m$ duality,

$$-\frac{2}{2\pi}\int_{x<0}b^{(2)}db^{(1)}+\frac{2}{2\pi}\int_{x>0}b^{(1)}db^{(2)}\ . \tag{30}$$

Using Stokes' theorem and integration by parts we rewrite it as

$$\frac{2}{2\pi} \int b^{(1)} db^{(2)} - \frac{2}{2\pi} \int_{x=0} b^{(1)} b^{(2)} \ . \tag{31}$$

Thus the domain wall at $x = 0$ is precisely the boundary term that we added to modify from the $e$ loop condensed boundary to the $m$ loop condensed boundary.

The two variants of the $m$ condensed boundary correspond to colliding the boundary by the topological domain defect decorated with $\frac{4}{4\pi} b^{(1)} b^{(1)}$. As discussed below, this is the domain wall defect that generates the $T^2$ transformation in the bulk.

### 3. Dyonic $\psi$ loop condensed boundary

We start with the $m$ loop condensed boundary, where the bulk effective action can be written as

$$-\frac{2}{2\pi} \int_M b^{(2)} db^{(1)} = \frac{2}{2\pi} \int_M b^{(1)} db^{(2)} - \frac{2}{2\pi} \int_{\partial M} b^{(1)} b^{(2)} \ . \tag{32}$$

Then, we modify the boundary by adding the boundary term

$$\frac{2}{4\pi} \int_{\partial M} b^{(1)} b^{(1)} \ . \tag{33}$$

The boundary term contributes to the equation of motion under $b^{(1)} \to b^{(1)} + \Delta b^{(1)}$

$$\frac{2}{2\pi} \int_{\partial M} b^{(1)} \Delta b^{(1)} \ . \tag{34}$$

Then combing with the variation from the bulk, we find $b^{(2)} + b^{(1)} = 0$ on the boundary. Thus the dyon surface operator $e^{i \int b^{(1)} + b^{(2)}}$ becomes trivial on the boundary, indicating the dyon $\psi$ loop excitation condenses on the boundary. The boundary is a $\mathbb{Z}_2$ 2-form gauge theory with gauge field $b^{(1)}$ (or $b^{(2)}$), describing an invertible topological order. The $\mathbb{Z}_2$ 2-form gauge theory on the boundary can be described by $U(1)$ 2-form gauge field $b^{(1)}$ and $U(1)$ Lagrangian multiplier 1-form gauge field $a$, with the action

$$\frac{2}{2\pi} \int_{\partial M} b^{(1)} da \ , \tag{35}$$

then the equation of motion for $b^{(1)}$, as a $U(1)$ valued field, implies $b^{(1)} + b^{(2)} + da = 0$ on the boundary. In particular, the boundary has gauge invariant open surface operator $\int_\Sigma b^{(1)} + b^{(2)} + \oint_{\partial \Sigma} a$: the dyonic loop on the boundary is described by $\oint_{\partial \Sigma} a$.

We remark that the boundary term is the effective action of the semion Walker-Wang model.

Similar to the $e$ condensed and $m$ condensed boundaries, there are two variants of the $\psi$ condensed boundary, corresponding to whether the $\mathbb{Z}_2$ topological order that is coupled to at the boundary is the semion Walker-Wang model or the anti-semion Walker-Wang model.. In field theory this corresponds to adding the boundary term that does not alter the equation of motion mod 2:

$$\frac{2^2}{4\pi}\int b^{(1)}b^{(1)} = \pi\int \frac{2b^{(1)}}{2\pi}\frac{2b^{(1)}}{2\pi} \; . \tag{36}$$

It has the effect of changing the spin of the dyon particle corresponding to the condensed $\psi$ loop by $1/2$ on the boundary [24].

The dyonic loop condensed boundary can be obtained from the magnetic loop condensed boundary by colliding it with the topological domain wall that generates the exchange symmetry $m \leftrightarrow \psi$

$$T: \quad (b^{(1)}, b^{(2)}) \to (b^{(1)}, b^{(2)} + b^{(1)}) \; . \tag{37}$$

In the above, $T$ is not to be confused with time-reversal symmetry; $S, T$ are the common notation to denote the generators of $SL(2, \mathbb{Z}_2)$ symmetry [17].[3] To see this, we note that such a domain wall can be described as follows. We insert the wall at $x = 0$, and on the left of the wall $x < 0$ we perform the $m \leftrightarrow \psi$ duality,

$$\frac{2}{2\pi}\int_{x<0} b^{(1)}d(b^{(2)} + b^{(1)}) + \frac{2}{2\pi}\int_{x>0} b^{(1)}db^{(2)} \; . \tag{38}$$

Using Stokes' theorem and integration by parts we rewrite it as

$$\frac{2}{2\pi}\int b^{(1)}db^{(2)} + \frac{2}{4\pi}\int_{x=0} b^{(1)}b^{(1)} \; . \tag{39}$$

Thus the domain wall at $x = 0$ is precisely the boundary term that we added to modify from the $m$ loop condensed boundary to the $\psi$ loop condensed boundary.

The two variants of the dyonic loop condensed boundary corresponds to colliding the boundary with the topological domain wall decorated with $\frac{4}{4\pi}\int b^{(1)}b^{(1)}$, which is the domain wall defect that generates the $T^{(2)}$ transformation in the bulk.

We remark that these six gapped boundaries (three kinds of loop condensations, and two variants for each) are in one-to-one correspondence with the Lagrangian subgroups of the

---

[3] We note that the $S^2 = C : (b^{(1)}, b^{(2)}) \to (-b^{(1)}, -b^{(2)})$ does not permute the excitations.

bulk 2-form $\mathbb{Z}_2$ gauge theory and extra $\mathbb{Z}_2$ "refinement". The bulk $\mathbb{Z}_2$ 2-form gauge theory has $(q_e, q_m) \in \mathbb{Z}_2 \times \mathbb{Z}_2$ fusion algebra, and there are three Lagrangian $\mathbb{Z}_2$ subgroups, generated by $(1,0), (0,1), (1,1)$ respectively. The refinement corresponds to colliding extra $T^2$ transformation domain wall defect to the boundary in the case of $m, \psi$ condensed boundaries, and $ST^2S$ in the case of $e$ condensed boundary. Such a connection is discussed in [30, 31], where the gapped boundaries are argued to be in one-to-one correspondence with the "refined polarizations".

## IV.   MAPPING BETWEEN $m$ LOOP AND $\psi$ LOOP

In this section we construct the unitary $Q$ that explicitly realizes the $m \leftrightarrow \psi$ exchange symmetry (37). This unitary operator satisfies the property that, given any closed surface $\Sigma$,

$$Q\,\mathcal{U}^{(0,1)}(\Sigma)Q^\dagger = \mathcal{U}^{(1,1)}(\Sigma), \qquad Q\,\mathcal{U}^{(1,1)}(\Sigma)Q^\dagger = \mathcal{U}^{(0,1)}(\Sigma) \tag{40}$$

In other words $Q$ transforms closed $m$ membrane operators into closed $\psi$ membrane operators and vice versa. We also write down the explicit lattice representation of this unitary as a finite depth quantum circuit composed of control-S gates, that can be implemented at the lattice level *i.e.* for the 4+1d toric code on the 4+1d hypercubic lattice. The action of $Q$ on a single Pauli $Z$ operator is Identity, so it leaves an open $Z$ membrane operator invariant. The action of $Q$ on a single Pauli $X$ operator dresses it with a Pauli $Z$ and a phase dependent on Pauli $Z$ operators. This implies that the open $X$ membrane operator maps to an $X \otimes Z$ membrane operator times $Z$-dependent phases. These $Z$-dependent phases cancel in the interior of the membrane leaving an $X \otimes Z$ membrane operator that is dressed at the boundary. As a warm-up for the reader, we discuss the topological domain wall and the unitary associated with it in 2+1d $\mathbb{Z}_2 \times \mathbb{Z}_2$ gauge theory in appendix D.

### A.   A unitary that implements $m \leftrightarrow \psi$ transformation at low energy

We give a construction of the unitary that implements $m \leftrightarrow \psi$ transformation via the low energy field theory *i.e.* the unitary we describe here is a symmetry of the theory in the low energy subspace with zero flux. To motivate the construction, we use the property that the domain wall implementing the $m \leftrightarrow \psi$ symmetry is decorated with the semion Walker-Wang

model. We change the time direction and reinterpret the domain wall as a domain wall in spacetime that acts on the entire space at a time slice. Then the unitary that implements the symmetry can be described by the Euclidean effective action of the semion Walker-Wang model in terms of $\mathbb{Z}_2$ 2-form gauge field, which is identified with the $\mathbb{Z}_2$ 2-form gauge field of the bulk 4+1d $\mathbb{Z}_2$ 2-form gauge theory.

The semion Walker-Wang model has effective action given by the gauged SPT phase with $\mathbb{Z}_2$ 1-form symmetry that is the root state generating the $\mathbb{Z}_4$ classification of the SPT phases [17, 24, 32, 33]. In the continuum, the effective action can be written as [17, 34]

$$\frac{2}{4\pi}b^{(1)}b^{(1)} + \frac{2}{2\pi}b^{(1)}da' \ , \tag{41}$$

where $b^{(1)}$ is a $U(1)$ 2-form gauge field, and $a$ is a $U(1)$ 1-form gauge field. Integrating out $a'$ enforces $b^{(1)}$ to have $\mathbb{Z}_2$ valued holonomy, with $b^{(1)}$ having holonomy $0, \pi$. On the lattice, we consider general $\mathbb{Z}_2$ variable $b = 0, 1$ that is closed $\delta b = 0$ mod 2 but it might not admit a $U(1)$ gauge field description $b^{(1)}$, where $b^{(1)} \sim \pi b$. The effective action for such a gauged SPT phase on the lattice is given by

$$\frac{\pi}{2}\int_{4d} \mathcal{P}(b), \quad \mathcal{P}(b) \equiv b \cup b - b \cup_1 \delta b \ , \tag{42}$$

where the part involving the higher cup product[4] $\cup_1$ is a correction that makes the effective action well-defined for $\mathbb{Z}_2$ variable $b$ [32]. To see this, we shift $b \to b + 2c$ with integral two-cochain $c$, the action changes by

$$\pi \int (b \cup c + c \cup b - b \cup_1 \delta c) \ \mathrm{mod} \ 2\pi \ , \tag{43}$$

where the first two terms come from $b \cup b$, and they are not equal since the cup product is not commutative, and they do not add up to zero mod 2. Instead, $b \cup c = c \cup b + \delta(b \cup_1 c) + b \cup_1 \delta c$ mod 2, where we used $\delta b = 0$ mod 2. This non-commutativity is compensated by the last term arising from $b \cup_1 \delta b$ in the action, and thus the action is well-defined mod $2\pi$ for $\mathbb{Z}_2$ variable $b$.

To describe the unitary operator on the lattice, we introduce $\mathbb{Z}_2$ variable $b$ on each face on the lattice, defined as $b = (1 - Z)/2$ that has eigenvalue $0, 1$. At low energy, $b$ is the $\mathbb{Z}_2$

---

[4] We went from a wedge product notation to cup products since we now have a discrete gauge group *i.e.* $\mathbb{Z}_2$ instead of the continuous gauge group $U(1)$. For a review of the cup products, see Appendix C

2-form gauge field, and $\delta b = 0$ mod 2. The unitary that implements $m \leftrightarrow \psi$ transformation can then be written as[5]

$$Q = i^{\int \mathcal{P}(b)} . \tag{45}$$

The $Q$ operator commutes with the loop toric code Hamiltonian at the zero flux low energy subspace, and thus it is a symmetry of the low energy theory. We note that $Q$ has order 4,

$$Q^2 = (-1)^{\int \mathcal{P}(b)} = (-1)^{\int b \cup b - b \cup_1 \delta b} . \tag{46}$$

Furthermore, $Q$ exchanges the $m$ excitation created by $\mathcal{U}^{(0,1)}$ and the $\psi$ excitation created by $\mathcal{U}^{(1,1)}$, while $Q^2$ does not change the types of excitations, as shown below.

We now show explicitly how the unitary $Q$ acts on the low energy theory. We use the identity

$$X_f i^b X_f = i^{b + \tilde{f}} (-1)^{b(f)\tilde{f}} , \tag{47}$$

where $\tilde{f}$ is the integral two-cochain that equals the identity operator on face $f$ and zero otherwise, $b$ is an operator-valued two-cochain, on face $f'$ it is $b(f') = (1 - Z_{f'})/2$. The last term is a correction that follows from taking a lift of $b = 0, 1$ in $\mathbb{Z}_4$; in the eigenbasis of $Z_f$, $i^{b(f)} = \text{diag}(1, i)$ and this leads to the above commutation relation when evaluated on face $f$. On any other face $f' \neq f$, $X_f i^{b(f')} X_f = i^{b(f')}$ and this is reproduced by the above commutation relation using $\tilde{f}(f') = 0$ by the definition of $\tilde{f}$.

The conjugation on $X_f$ for fixed face $f$ by $Q$ gives

$$Q X_f Q^{-1} = X_f i^{\int \mathcal{P}(b + \tilde{f}) - \mathcal{P}(b)} (-1)^{b(f) \int (b \cup \tilde{f} + \tilde{f} \cup b + b \cup_1 \delta \tilde{f} + \tilde{f} \cup_1 \delta b)} \tag{48}$$

where we used $\tilde{f} \cup \tilde{f} = 0, \tilde{f} \cup_1 \delta \tilde{f} = 0$. The equation can be simplified using

$$\mathcal{P}(b + \tilde{f}) - \mathcal{P}(b) = 2\tilde{f} \cup b + \delta z - 2\tilde{f} \cup_1 \delta b + \delta \tilde{f} \cup_2 \delta b, \quad z = \tilde{f} \cup_1 b + \delta \tilde{f} \cup_2 b . \tag{49}$$

And thus the conjugation gives

$$Q X_f Q^{-1} = X_f (-1)^{\int \tilde{f} \cup b} \cdot (-1)^{\int \tilde{f} \cup_1 \delta b} i^{(1 + 2b(f)) \int \delta \tilde{f} \cup_2 \delta b} . \tag{50}$$

---

[5] Explicitly, on a triangular lattice, on a 4-simplex with vertices $(01234)$ it has the contribution

$$\mathcal{P}(b)(01234) = (b \cup b - b \cup_1 \delta b)(01234) = b(012)b(234) - b(034)\,(b(123) - b(023) + b(013) - b(012))$$

$$- b(014)\,(b(234) - b(134) + b(124) - b(123)) , \tag{44}$$

where $b(v_0, v_1, v_2) = (1 - Z_{(v_0, v_1, v_2)})/2$.

In other words, the conjugation changes $X_f$ to $X_f \prod_{f':\int \tilde{f}\cup\tilde{f}'=1 \bmod 2} Z_{f'}$ along with a product of operators that are trivial on the zero flux sector $\delta b = 0 \bmod 2$ or localized on the excitation (depends on $f$ by $\delta\tilde{f}$).

Let us show the unitary $Q$ changes the superselection sector of the magnetic loop into that of the dyonic loop. Consider the state with infinite-length magnetic string on hypercubic lattice created by $\mathcal{U}^{(0,1)}(\Sigma) = \prod X_f$ where the product is over parallel faces $f$ that intersect perpendicularly with a semi-infinite half plane on the dual lattice (the boundary of the half plane is the magnetic string). Such state belongs to the magnetic superselection sector.[6] Then the operator conjugated by $Q$ creates a state with infinite-length dyonic string: denote the ground state by $|0\rangle$,

$$Q\mathcal{U}^{(0,1)}(\Sigma)Q^{-1}|0\rangle = \mathcal{U}^{(1,1)}(\Sigma)i^{\int \delta\tilde{\Sigma}\cup_2\delta b}|0\rangle = i^{\int \delta\tilde{\Sigma}\cup_2\delta b}\mathcal{U}^{(1,1)}(\Sigma)|0\rangle , \tag{51}$$

where $\tilde{\Sigma} = \sum \tilde{f}$, and $i^{\int \delta\tilde{\Sigma}\cup_2\delta b}$ is localized on the string, since it depends on $\Sigma$ only by $\delta\tilde{\Sigma}$. We have used the identity $\int \delta\tilde{f}\cup_2\delta\tilde{f}' = 0$ for any two faces $f, f'$ that do not share a common vertex. Such state belongs to the dyon superselection sector.

In other word, we modify the dyon operator to be the conjugation of the magnetic operator by $Q$ without changing the dyon superselection sector, and the new dyon creation operators commute among themselves:

$$\mathcal{U}'^{(1,1)}(f) \equiv Q\mathcal{U}^{(0,1)}(f)Q^{-1}, \quad \mathcal{U}'^{(1,1)}(f)\mathcal{U}'^{(1,1)}(f') = \mathcal{U}'^{(1,1)}(f')\mathcal{U}'^{(1,1)}(f) . \tag{52}$$

We remark that the factor of $i$ implies that $\mathcal{U}^{(1,1)}, \mathcal{U}'^{(1,1)}$ transform differently under time-reversal symmetry.[7]

While $Q$ permutes $m \leftrightarrow \psi$, $Q^2$ does not permute the types of excitations,

$$Q^2\mathcal{U}^{(0,1)}(f)Q^{-2} = Q\mathcal{U}'^{(1,1)}(f)Q^{-1} = QX_fQ^{-1}(-1)^{\int \tilde{f}\cup b} \cdot (-1)^{\int \tilde{f}\cup_1\delta b}i^{(1+2b(f))\int \delta\tilde{f}\cup_2\delta b}$$
$$= X_f(-1)^{\int \delta\tilde{f}\cup_2\delta b} = \mathcal{U}^{(0,1)}(f)(-1)^{\int \delta\tilde{f}\cup_2\delta b} . \tag{53}$$

It follows that for $\Sigma$ given by union of faces intersecting with a half plane,

$$Q^2\mathcal{U}^{(0,1)}(\Sigma)Q^{-2}|0\rangle = (-1)^{\int \delta\tilde{\Sigma}\cup_2\delta b}\mathcal{U}^{(0,1)}(\Sigma)|0\rangle , \tag{54}$$

---

[6] More precisely, the state is the complex value linear functional $\alpha_M$ on the $\mathbb{C}^*$ algebra of quasi-local observables $\{\mathcal{O}\}$, $\alpha_M(\mathcal{O}) = \alpha(\mathcal{U}^{(0,1)}(\Sigma)\mathcal{O}(\mathcal{U}^{(0,1)}(\Sigma))^{-1})$, where $\alpha(\mathcal{O})$ is the expectation value of $\mathcal{O}$ on the vacuum $|0\rangle$, and $\alpha_M(\mathcal{O})$ is the expectation value of $\mathcal{O}$ on the state with infinite magnetic string.

[7] If the theory is enriched by time-reversal symmetry, then we consider dyonic loop on unorientable space-time or in the presence of time-reversal domain walls. Then the dyon surface operators have non-trivial correlation functions as discussed in [35], where such loop excitations are referred to as "exotic loops".

and thus $\mathcal{U}^{(0,1)}$ conjugated by $Q^2$ creates an excitation that belongs to the same superselection sector as the magnetic excitation.

## B. Circuit interpretation for the lattice model

The unitary in eqn (45) can be written as

$$U = \exp\left[i\frac{\pi}{2}\left(\int b \cup_0 b - \int b \cup_1 \delta b\right)\right]. \tag{55}$$

We now express it as a quantum circuit in terms of local gates acting on the $\mathbb{Z}_2$ 4+1d toric code's degrees of freedom placed on the 4+1d hypercubic lattice. In order to do this, we use the definition of the higher cup product on the hypercubic lattice and necessary notation from Ref. [36]. We review these definitions and the notation in appendix C. Here $b$ is an operator-valued 2-cochain whose value on face $f$ is $b(f) = (1 - Z_f)/2$, and $\delta$ is a coboundary operator. The two integrals in the exponent commute with each other and hence we split the exponential. Both terms specify a set of Control Phase gates which we describe using the notation as used in [36] for the cells of the hypercubic lattice. In this notation, as shown in fig. 7, the 4-cell is denoted as $(\bullet, \bullet, \bullet, \bullet)$ where $\bullet$ refers to an $x$, $y$, $z$ or $w$ coordinate for the 4-cell for the $\bullet$ at first, second, third or fourth position respectively.

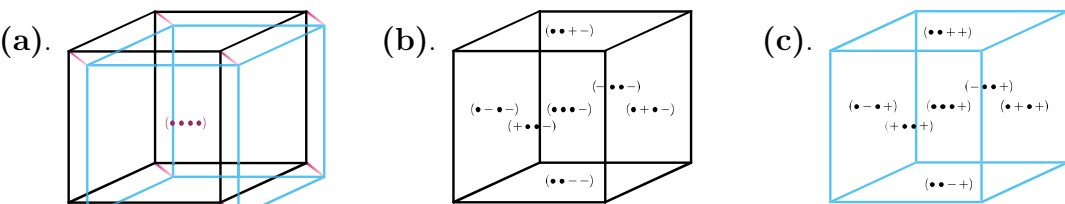

FIG. 7. Examples for the bullet notation used for the 4D hypercubic lattice (a) 4-cell denoted by $(\bullet, \bullet, \bullet, \bullet)$. $xyz$ 3-cells at $w = 0$ and $w = 1$ denoted by $(\bullet, \bullet, \bullet, -)$ and $(\bullet, \bullet, \bullet, +)$ shown in (b) and (c) along with the 2-cell boundaries. The $-$ and $+$ in a coordinate denote the position of the $i$-cell at the boundary relative to the center of the $i + 1$-cell.

Replacing any one $\bullet$ by $+$ or $-$ gives a 3-cell on the boundary of the 4-cell. For example, $(+, \bullet, \bullet, \bullet)$ and $(-, \bullet, \bullet, \bullet)$ are two $yzw$ 3-cells on the boundary of the 4-cell, one is at a negative position in $x$ direction relative to the center of the 4-cell and the other one is at a positive relative position. Using Eq. C2 in appendix C, we write the action of $b \cup_0 b$ on a

4-cell as

$$(b \cup_0 b)(\bullet, \bullet, \bullet, \bullet)$$

$$=b(\bullet, \bullet, +, +)b(-, -, \bullet, \bullet) + b(\bullet, +, +, \bullet)b(-, \bullet, \bullet, -) + b(+, +, \bullet, \bullet)b(\bullet, \bullet, -, -)$$

$$+ b(+, \bullet, \bullet, +)b(\bullet, -, -, \bullet) - b(+, \bullet, +, \bullet)b(\bullet, -, \bullet, -) - b(\bullet, +, \bullet, +)b(-, \bullet, -, \bullet) \,,$$

$$(56)$$

where the terms in the brackets with two $\bullet$ indicate a 2-cell in the boundary of the 4 cell. The action of $b \cup_1 \delta b$ on a 4-simplex denoted as $(\bullet, \bullet, \bullet, \bullet)$ is given by

$$(b \cup_1 \delta b)(\bullet, \bullet, \bullet, \bullet)$$

$$= - b(+, +, \bullet, \bullet)\delta b(\bullet, \bullet, -, \bullet) + b(+, \bullet, +, \bullet)\delta b(\bullet, -, \bullet, \bullet) - b(\bullet, +, +, \bullet)\delta b(-, \bullet, \bullet, \bullet)$$

$$-b(+, +, \bullet, \bullet)\delta b(\bullet, \bullet, \bullet, +) - b(\bullet, +, \bullet, -)\delta b(-, \bullet, \bullet, \bullet) + b(+, \bullet, \bullet, -)\delta b(\bullet, -, \bullet, \bullet)$$

$$-b(+, \bullet, -, \bullet)\delta b(\bullet, \bullet, \bullet, +) + b(+, \bullet, \bullet, -)\delta b(\bullet, \bullet, +, \bullet) - b(\bullet, \bullet, -, -)\delta b(-, \bullet, \bullet, \bullet)$$

$$-b(\bullet, -, -, \bullet)\delta b(\bullet, \bullet, \bullet, +) - b(\bullet, \bullet, -, -)\delta b(\bullet, +, \bullet, \bullet) + b(\bullet, -, \bullet, -)\delta b(\bullet, \bullet, +, \bullet)$$

$$(57)$$

where $b$ acts on the 2-cells and $\delta b$ acts on the 3-cells. The action of the coboundary operator $\delta$ is given by $\delta b(\text{3-cell}) = b(\partial(\text{3-cell}))$ where $\partial(\text{3-cell})$ denotes the boundary of the 3-cell. For example, in eq.(57), $\delta b(\bullet, \bullet, -, \bullet)$ can be expanded as

$$\delta b(\bullet, \bullet, -, \bullet)$$

$$=b(+, \bullet, -, \bullet) - b(-, \bullet, -, \bullet) - b(\bullet, +, -, \bullet) + b(\bullet, -, -, \bullet) - b(\bullet, \bullet, -, +) + b(\bullet, \bullet, -, -).$$

$$(58)$$

where we used $\delta b(..., \bullet_{i_1}, ..., \bullet_{i_2}, ..., \bullet_{i_n}, ...) = \sum_{p=1, \pm \in \{+,-\}}^{n} \mp(-1)^p b(..., \pm, ...)$ [36]. The action of coboundary operator $\delta$ on the other cells appearing in (57) can be written down similarly. After expanding Eq. 57 using Eq. 58, all pairs of qubits between which the Control Phase gate acts, are specified in Eq. 56 and Eq. 57 as products of two 2-cells. The $\pm$ sign in front of such products indicates whether it is the Control Phase gate from the first qubit as control to second qubit as target, or its inverse.

## V. GENERALIZATION TO $\mathbb{Z}_N$ 2-FORM GAUGE THEORY IN 4+1D

In this section, we generalize from the $\mathbb{Z}_2$ 4+1d toric code to the $\mathbb{Z}_N$ 4+1d toric code. Thus, the low energy description is generalized to 2-form $\mathbb{Z}_N$ gauge theory in 4+1d. It can be described by $U(1) \times U(1)$ 2-form gauge fields $b^{(1)}, b^{(2)}$ with the "mixed Chern-Simons action"

$$\frac{N}{2\pi} b^{(1)} db^{(2)} \ . \tag{59}$$

We interpret the gauge field $b^{(2)}$ as the "Lagrangian multiplier" that enforces $b^{(1)}$ to have holonomy valued in integer multiple of $\frac{2\pi}{N}$, and thus $b^{(1)}$ is effectively a $\mathbb{Z}_N$ 2-form gauge field.

### A. Symmetries

*a. 2-form symmetry* In the $\mathbb{Z}_N$ gauge theory, we have the electric and magnetic surface operators, $\mathcal{U}^E(\Sigma)$ and $\mathcal{U}^M(\Sigma)$ respectively, whose commutation relation generalizes to

$$\mathcal{U}^E(\Sigma)\mathcal{U}^M(\tilde{\Sigma}) = \mathcal{U}^M(\tilde{\Sigma})\mathcal{U}^E(\Sigma)e^{\frac{2\pi i}{N}\#(\Sigma,\tilde{\Sigma})} \ , \tag{60}$$

as can be verified using the clock and shift algebra obeyed by the generalized Pauli operators in $\mathbb{Z}_N$, $\mathcal{X}_f, \mathcal{Z}_f$. Here $\#(\Sigma, \tilde{\Sigma})$ counts the (signed) intersection number of the surfaces $\Sigma, \tilde{\Sigma}$. Besides the electric and magnetic surface operators, the dyon surface operators $\mathcal{U}^{(q_e,q_m)}(\Sigma)$ with electric and magnetic charges $(q_e, q_m)$, are defined as follows,

$$\mathcal{U}^{(q_e,q_m)}(\Sigma) = (\mathcal{U}^E(\Sigma))^{q_e} (\mathcal{U}^M(\tilde{\Sigma}))^{q_m} \sim e^{iq_e \int b^{(1)} + iq_m \int b^{(2)}} \ , \tag{61}$$

where we choose $\tilde{\Sigma}, \Sigma$ related by displacing in the $(1,1,1,1)$ direction on the hypercube lattice. The dyon surface operator $(q_e, q_m)$ on any closed surface commutes with the Hamiltonian, and thus it is a conserved charge, generating a 2-form symmetry [17]. In Appendix F, we give an introduction to the higher-form symmetries. The $N/\gcd(q_e, q_m, N)$th power of the dyon surface operator $(q_e, q_m)$ equals the identity, and thus the 2-form symmetry generated by the $(q_e, q_m)$ dyon surface operator is $\mathbb{Z}_{\frac{N}{\gcd(q_e,q_m,N)}}$.

The $\mathbb{Z}_N$ 2-form symmetry generated by the dyon surface operator is non-anomalous on orientable spacetime. To see this, we turn on the background $B_3$ for the 2-form symmetry:

$$\frac{N}{2\pi} \int \left( b^1 db^2 + (b^1 + b^2)B_3 \right) \ . \tag{62}$$

Under a 2-form background gauge transformation $b^1 \to b^1 - \lambda, b^2 \to b^2 + \lambda, B_3 \to B_3 - d\lambda$ for 2-form $\lambda$, the action changes by a total derivative and thus it is invariant on orientable spacetime.

b. *Time-reversal symmetry* The theory also has time-reversal symmetry that transforms the 2-form gauge fields as $\mathcal{T} : (b^{(1)}, b^{(2)}) \to (b^{(1)}, -b^{(2)})$, which can be combined with other unitary symmetries to define other anti-unitary symmetries. Due to this action of the time-reversal symmetry, it leaves the electric loop invariant but conjugates the magnetic loops.

c. *Parity anomaly* We will show that the 2-form symmetry generated by the dyon surface operator can have a parity anomaly, *i.e.* a mixed 't Hooft anomaly with time-reversal symmetry. In other words, if we insist the theory to be time-reversal symmetric, then the theory will not be gauge invariant under the 2-form symmetry.

If we enrich the theory by time-reversal symmetry, we consider domain wall defect that implements the time-reversal transformation. The correlation function of the dyon surface operator can be computed in a similar fashion as in Section II.C, and it is $q_e q_m$ multiplied by the integral of a total derivative that becomes non-trivial in the presence of time-reversal domain wall. This implies that the 2-form symmetry generated by the dyon surface operator becomes anomalous, for suitable $q_e q_m$. The anomaly can be understood as follows. A background 2-form gauge transformation changes the theory by $-\frac{N q_e q_m}{2\pi} \int \lambda d\lambda$, which can be compensated by a 5+1d bulk $\mathbb{Z}_N$ 2-form symmetry protected topological phase that has the effective action

$$\frac{N q_e q_m}{2\pi} \int B_3 B_3 = \frac{2\pi q_e q_m}{N} \int BB, \quad B \equiv \frac{N B_3}{2\pi} . \tag{63}$$

Here, $B_3$ is a background 3-form gauge field. This term is non-trivial only for even $N$ and when $q_e q_m$ equals an odd multiple of $N/2$, where we used that $B \equiv \frac{N B_3}{2\pi}$ has integer flux, and the action is defined modulo a multiple of $2\pi$. In other cases, the term vanishes since 3-forms anticommute. One can show the anomaly is always trivial on orientable spacetime due to the Wu formula $x_3 x_3 = w_1 w_2 x_3$ [29] where $x_3$ is any $\mathbb{Z}_2$ three-cocycle and $w_1, w_2$ are the first and second Stiefel-Whitney classes of the tangent bundle, with $w_1$ being trivial on orientable spacetime.

Let us restrict to even $N$. The dyon surface operator with $q_m = N/2$ is invariant under the time-reversal symmetry, and it generates $\mathbb{Z}_{N/\gcd(q_e, N/2)}$ 2-form symmetry, which has an

anomaly described by the 5+1d SPT phase with effective action

$$\pi q_e \gcd(q_e, N/2)^2 \int BB \ , \tag{64}$$

where $B$ is the background $\mathbb{Z}_{N/\gcd(q_e,N/2)}$ 3-form gauge field. For odd $q_e$ and $\gcd(q_e, N/2)$ (which implies $N/\gcd(q_e, N/2)$ is even), the above SPT phase equals $\pi \int Bw_1(TM)w_2(TM)$ mod $2\pi$. This implies that the 2-form symmetry generated by the dyon surface operator has an anomaly, and this anomaly can be resolved by choosing a local trivialization of $w_1(TM)w_2(TM)$ on the worldsheet of the dyonic loop. This implies that such dyon has a novel time-reversal transformation property, as studied in [35], which differs from the ordinary time-reversal transformation in the sense that it changes the framing dependence of the dyonic loop by stacking it with the worldline of a spin-1/2 particle that is attached to the loop. For instance, when $N = 2$, the half dyonic loop is attached to antisemions at the boundary, and the usual time-reversal transformation converts the antisemions to semions. Each semion needs to be stacked with a spin 1/2 particle (such that it is also attached to the end of the dyonic loop) to map back to an antisemion.

We stress that as emphasized in [37], the 't Hooft anomaly as discussed above is different from the higher-group symmetry. In particular, it is not related to the $H^4$ obstruction that we discuss later, which is related to 3-group symmetry.

d. *Loop permutation 0-form symmetry $SL(2, \mathbb{Z}_N)$* In addition to the 2-form symmetry, the $\mathbb{Z}_N$ theory also has a loop permutation symmetry $SL(2, \mathbb{Z}_N)$, generated by codimension-one gapped domain walls.

The $SL(2, \mathbb{Z}_N)$ symmetry is discussed on spin manifolds in [17]. For even $N$ it is generated by the transformation

$$\begin{aligned} S: \quad & b^{(1)} \to b^{(2)}, \quad b^{(2)} \to -b^{(1)} \\ T: \quad & b^{(1)} \to b^{(1)}, \quad b^{(2)} \to b^{(2)} + b^{(1)} \ . \end{aligned} \tag{65}$$

For odd $N$, we have the same $S$ transformation, but the $T$ transformation is

$$T: \quad b^{(1)} \to b^{(1)}, \quad b^{(2)} \to b^{(2)} + (N+1)b^{(1)} \ . \tag{66}$$

As we discuss in Section VI, for even $N$, although the $T^N$ induces a trivial permutation of the loop excitations but the corresponding domain wall defect is not completely trivial, and we denote it by $D^N$. The domain wall defects that generate the transformations can be described as follows.

- $S$ transformation. We perform the $S$ transformation on $x < 0$

$$-\frac{N}{2\pi}\int_{x<0} b^{(2)}db^{(1)} + \frac{N}{2\pi}\int_{x>0} b^{(1)}db^{(2)} = \frac{N}{2\pi}\int b^{(1)}db^{(2)} - \frac{N}{2\pi}\int_{x=0} b^{(1)}b^{(2)} \ . \quad (67)$$

Thus we find the domain wall defect at $x = 0$ is described by $\frac{N}{2\pi}b^{(1)}b^{(2)}$.

- $T^p$ transformation. We perform the $T^p$ transformation on $x < 0$

$$\frac{N}{2\pi}\int_{x<0} b^{(1)}d(b^{(2)} + pb^{(1)}) + \frac{N}{2\pi}\int_{x>0} b^{(1)}db^{(2)} = \frac{N}{2\pi}\int b^{(1)}db^{(2)} + \frac{Np}{4\pi}\int_{x=0} b^{(1)}b^{(1)} \ . \quad (68)$$

As mentioned, the above expression is for the $T$ transformation for even $N$. For odd $N$, we replace $p$ by $(N+1)p$. Thus we find the domain wall defect at $x = 0$ is described by $\frac{Np}{4\pi}b^{(1)}b^{(1)}$. This is the effective action of the Walker Wang model with boundary $\mathbb{Z}_N$ anyon of spin $-p/(2N)$ mod 1.

## B. Gapped boundaries

For the $\mathbb{Z}_N$ gauge theoey, the algebra of operators is simply generalized from $\mathbb{Z}_2$ to $\mathbb{Z}_N$ and hence, the Lagrangian description of gapped boundaries has modifications only in constant coefficients in terms, which change from 2 to $N$. Firstly we note that, as discussed for $\mathbb{Z}_2$ before, for even $N$, there are two variants of the $e$, $m$ and $\psi$ condensed boundaries, differing by a quadratic refinement [30, 31]. On the other hand, for odd $N$, there is only a single subtype of the $e$, $m$ and $\psi$ boundaries. We explain the subtypes of the boundaries for even and odd $N$ below.

Following [30, 31, 38], the gapped boundaries are the topological boundary conditions for the 2-form gauge theory. The distinct gapped boundaries are labelled by Lagrangian subgroups of $\mathbb{Z}_N \times \mathbb{Z}_N$, and for even $N$ there is extra $\mathbb{Z}_2$ refinement for each Lagrangian subgroup [30, 31] that corresponds to colliding the boundary with the $D^N$ or $SD^N S$ domain wall defect where the $S$ denotes the $S$ transformation mentioned in Eq. 67 (recall $D^N$ is the domain wall corresponding to the $N$th power of the $T$ transformation, and it does not permute the loop excitations). Due to the condensation on the boundary, when collided with the boundary, either one of these domain wall defects becomes trivial or the two become linearly dependent. For instance, in the case of the $(1,0)$ electric loop condensed boundary, $D^N$ becomes trivial while $SD^N S$ is not, and colliding with $SD^N S$ gives an additional boundary. Similarly, in the case of the $(p,1)$ loop condensed boundary, there are two cases.

For $p = 0$, the $SD^N S$ domain wall becomes trivial on the boundary while $D^N$ is not and gives rise to an additional boundary. For $p \neq 0$ the two domain wall defects become linearly dependent.

We first consider the $e \equiv (1,0)$ condensed boundary. This means the boundary condition $b^{(1)}| = da^{(1)}$ for 1-form gauge field $a^{(1)}$. In particular, the $e$ operator $\int b^{(1)}$ can end on the boundary. We take the boundary-bulk to be

$$\frac{N}{2\pi} \int_{\partial M} b^{(2)} da^{(1)} + \frac{N}{2\pi} \int_M b^{(1)} db^{(2)} \ , \tag{69}$$

where the equation of motion for $b^{(2)}$ has a boundary variation term that imposes $b^{(1)}| + da^{(1)} = 0$:

$$\frac{N}{2\pi} \Delta b^{(2)} (da^{(1)} + b^{(1)}) = 0 \text{ on boundary } \partial M \ . \tag{70}$$

where the last term is from the boundary variation of $b^{(1)} db^{(2)}$. The boundary is an untwisted $\mathbb{Z}_N$ 2-form gauge theory (also dual to a $\mathbb{Z}_N$ 1-form gauge theory). There are two variants of the $e$ condensed boundary for even $N$, which differ by a quadratic refinement [30, 31], In field theory this corresponds to adding the boundary term that does not alter the equation of motion mod $N$:

$$\frac{N^2}{4\pi} \int b^{(2)} b^{(2)} = \pi \int \frac{N b^{(2)}}{2\pi} \frac{N b^{(2)}}{2\pi} \ . \tag{71}$$

The additional term has the effect of changing the framing of the condensed $e$ loop on the boundary by stacking it with the worldline of a spin-1/2 particle attached to the loop [24].

The $m \equiv (0,1)$ condensed boundary is similar to the $e$ condensed boundary, with $b^{(1)}, b^{(2)}$ exchanged. The $m$ condensed boundary is obtained from the $e$ condensed boundary by colliding the $e$ condensed boundary with the topological domain wall that implements the $S$ transformation. Hence, there are again two variants of the $m$ condensed boundary for even $N$, which differ by a quadratic refinement [30, 31]. In field theory this corresponds to adding the boundary term that does not alter the equation of motion mod $N$:

$$\frac{N^2}{4\pi} \int b^{(1)} b^{(1)} = \pi \int \frac{N b^{(1)}}{2\pi} \frac{N b^{(1)}}{2\pi} \ . \tag{72}$$

The additional term has the effect of changing the framing of the condensed $m$ loop on the boundary by stacking it with the worldline of a spin 1/2 particle attached to the loop [24].

Finally for the dyon $(p, 1)$ condensed boundary, we have a boundary condition $(p b^{(1)} + b^{(2)})| = da$ for 1-form gauge field $a$. In particular, the dyon operator $\int p b^{(1)} + b^{(2)}$ can end

on the boundary. We take the boundary-bulk to be

$$\int_{\partial M}\left(\frac{N}{2\pi}b^{(1)}da + \frac{Np}{4\pi}b^{(1)}b^{(1)}\right) + \frac{N}{2\pi}\int_M b^{(2)}db^{(1)} \ , \tag{73}$$

where if $Np$ is odd we replace $p$ with $p + N$.[8] Then the equation of motion for $b^{(1)}$ has a boundary term that gives $b^{(2)}| + pb^{(1)}| + da = 0$.

$$\frac{N}{2\pi}\Delta b^{(1)}(da^{(1)} + pb^{(1)} + b^{(2)}) = 0 \text{ on boundary } \partial M \ , \tag{74}$$

where the last term is from the boundary variation of $b^{(2)}db^{(1)}$. The boundary is 2-form $\mathbb{Z}_N$ gauge theory that describes a gauged $\mathbb{Z}_N$ 1-form symmetry SPT phase, and it is also the effective action for the Walker-Wang model for $\mathbb{Z}_N$ Abelian anyons generated by an anyon of spin $-\frac{p}{2N}$. The dyon $(p, 1)$ condensed boundary is obtained from the $e$ condensed boundary by colliding the $e$ condensed boundary with the topological domain wall that implements the $T^p$ transform. There are again two variants of each dyon condensed boundary for even $N$, which differ by a quadratic refinement [30, 31]. In field theory this corresponds to adding the boundary term that does not alter the equation of motion mod $N$:

$$\frac{N^2}{2\pi}\int b^{(1)}b^{(1)} = \pi\int \frac{Nb^{(1)}}{2\pi}\frac{Nb^{(1)}}{2\pi} \ . \tag{75}$$

The additional term changes $p$ to $p + N$, and it has the effect of changing the framing of the condensed dyonic loop on the boundary by stacking it with the worldline of a spin $1/2$ particle attached to the loop [24].

## VI.   AN "$H^4$ OBSTRUCTION" TO $m \leftrightarrow \psi$ SYMMETRY FOR EVEN $N$

In this section, we discuss the obstruction to gauging the permutation automorphism between $m$ and $\psi$ loop excitations discussed in a previous section.

We focus on the global symmetry of the $\mathbb{Z}_N$ 2-form gauge theory. We show that for even $N$, the $SL(2, \mathbb{Z}_N)$ loop-permutation symmetry in fact has an "$H^4$ obstruction", similar to the "$H^3$ obstruction" to symmetry fractionalization in symmetry-enriched topological phase in 2+1d [15, 37, 39]. Such an obstruction means that the permutation symmetry combines with the 2-form symmetry to form a 3-group symmetry [37]. Here, the permutation symmetry is not a subgroup and thus it does not make sense to gauge the permutation symmetry

---

[8] Otherwise the boundary would have extra neutral fermion particles.

alone without also gauging the 2-form symmetry. Note that, for odd $N$, the permutation symmetry does not combine with the 2-form symmetry in such an intricate way and can be gauged alone. We stress that the $H^4$ obstruction that we discuss means that the symmetry participates in a 3-group, and it is different from the 't Hooft anomaly [37].

The defining feature of the $H^3$ obstruction in symmetry-enriched topological phases in 2+1d is that fusing the domain wall defects that generate the 0-form symmetry can produce an additional Abelian anyon, which generates the 1-form symmetry [15, 37, 39]. Such fusion algebra holds true even when the domain wall is supported on closed submanifolds.[9] Equivalently, performing a 0-form symmetry transformation in the presence of background gauge field, which is specified by the configuration of 0-form symmetry domain wall defects, produces a 1-form symmetry background gauge field [37]. This is the global symmetry analogue of the Green-Schwarz mechanism in string theory. As a consequence, it is not possible to gauge the 0-form symmetry, where we sum over all possible insertions of the 0-form symmetry defects, without also summing over the insertions of 1-form symmetry defect, since the fusion of the former produces the latter. The 0-form symmetry and 1-form symmetry combine into a 2-group symmetry, and thus, the 0-form symmetry is not a "subgroup": wenot gauge the 0-form symmetry alone in a 2-group symmetry. This is familiar in the group extension theory: consider $\mathbb{Z}_4 = \{1, \omega, \omega^2, \omega^3\}$ extension of $\mathbb{Z}_2$ by the $\mathbb{Z}'_2 = \{1, \omega^2\}$ subgroup, with $\omega = i$. The element $\omega$ fuses into an element of $\mathbb{Z}'_2$, and thus wenot gauge the "$\mathbb{Z}_2$ symmetry" generated by $\omega$ without also gauging $\mathbb{Z}'_2$. In other words, we need to gauge the entire extension $\mathbb{Z}_4$.

In the following, we show that there is a similar obstruction for the $\mathbb{Z}_N$ loop permutation symmetry $T : (q_e, q_m) \rightarrow (q_e, q_m + q_e)$ in the $\mathbb{Z}_N$ 2-form gauge theory in 4+1d. We show that fusing the domain wall defects that generate the 0-form symmetry produces the 2-form symmetry defect. This means the 0-form symmetry has an "$H^4$ obstruction", or equivalently it participates in a 3-group symmetry together with the 2-form symmetry. This means wenot gauge the 0-form symmetry alone, since summing over the 0-form symmetry defect insertions necessarily contains the sum over the 2-form symmetry defect insertions that are produced by the fusion of the 0-form symmetry defects.

---

[9] There is another phenomena where fusing the domain wall defects produces Abelian anyons only on its boundary, but not when the domain wall is supported on closed manifolds. This is an example of trivial $H^3$ obstruction. We thank M. Cheng for bringing this point to our attention.

## A. $H^4$ obstruction: a 3-group global symmetry

There are several ways to understand the $H^4$ obstruction in this example, or the 3-group symmetry.

We begin by noting that the order $N$ symmetry $T$ that transforms $(q_e, q_m) \rightarrow (q_e, q_m + q_e)$ is generated by a defect $D$ that, for even $N$, supports a gauged SPT phase of order $2N$ with $\mathbb{Z}_N$ 1-form symmetry, and thus $D$ has order $2N$. In ordinary bosonic systems, where the spacetime does not have an additional structure, there is no problem in gauging this $\mathbb{Z}_{2N}$ symmetry associated with $D$, that has the same loop permutation action as $T$: one simply proliferates the domain wall defects[10]. Alternatively, we include local fermions in the system, then the domain wall defect has order $N$, and there is again no obstruction to gauging the $\mathbb{Z}_N$ permutation symmetry in the system with local fermions.

On the other hand, for a general bosonic system, fusing $N$ copies of the domain wall defects together does not produce the identity domain wall defect. We call the resulting domain wall defect $D^N$.

As we will see, such a domain wall $D^N$ has the same correlation function as a particular electric loop excitation. The appearance of lower-dimensional defects in the correlation function of domain walls is similar to the $H^3$ obstruction in symmetry fractionalization in 2+1d [15, 37, 39] where the two ways of fusing the domain wall defects, defined by the two $F$ moves, differ by an Abelian anyon. The domain wall action for $D^N$ is

$$\frac{N^2}{4\pi} \int b^{(1)} b^{(1)} = \pi \int \frac{Nb^{(1)}}{2\pi} \frac{Nb^{(1)}}{2\pi} \ . \tag{76}$$

The effective action is the same as the effective action that describes the $\mathbb{Z}_N$ 1-form gauge theory where the electric particle of charge $N/2$ is a fermion [17, 27, 40]; equivalently, the magnetic particle in the $\mathbb{Z}_N$ 2-form gauge theory of magnetic charge $N/2$ is a fermion. Thus, the domain wall defect $D^N$ has the effect of changing the magnetic loop excitation by an additional fermion restricted to the loop. Equivalently, whenever we change the framing of the loop, we insert an electric loop that braids with the magnetic loop nontrivially such that the change in framing comes with an additional sign. We also use the mathematical identity

$$\frac{N^2}{4\pi} \int b^{(1)} b^{(1)} = \pi \int \frac{Nb^{(1)}}{2\pi} \frac{Nb^{(1)}}{2\pi} = \pi \int \frac{Nb^{(1)}}{2\pi} w_2(TM)| \tag{77}$$

---

[10] Here when we refer to the domain wall, we do not imply that there is a finite tension dynamical excitation associated with spontaneously broken 0-form symmetry; instead, these domain walls are external defects.

where $w_2(TM)|$ is the restriction of the second Stiefel-Whitney class of the tangent bundle. We used the Wu formula $x_2 \cup x_2 = x_2 \cup (w_2(TM) + w_1(TM)^2)$ for $\mathbb{Z}_2$ two-cocycle $x_2$ [29], and we assumed the ambient spacetime is orientable. The right hand is an insertion of the operator $\oint b^{(1)}$ that describes the electric loop excitation. Thus we find that the domain wall $D^N$ is non-trivial (and thus there is an obstruction to gauging the $\mathbb{Z}_N$ symmetry). But it is completely equivalent to a higher-codimension operator: it is the same as insertion of the electric surface operator, that generates $\mathbb{Z}_N$ 2-form symmetry, at a suitable locus in any correlation function. Thus the fusion of 0-form symmetry defects produces a 2-form symmetry defect, indicating the 0-form symmetry mixes with the 2-form symmetry and it participates in a 3-group. In particular, wenot gauge the 0-form symmetry alone.

We now repeat the discussion using background gauge fields instead of using the symmetry defects. The symmetry defects and the corresponding background gauge fields are related by the Poincaré duality. Let us turn on the background gauge field $B_1$ for the $\mathbb{Z}_N$ 0-form symmetry and the background gauge field $B_3$ for the $\mathbb{Z}_N$ 2-form symmetry. The generator for the $m \to \psi$ symmetry is the domain wall $\oint \frac{N}{4\pi} b^{(1)} b^{(1)}$, and thus, turning on the background $B_1$ for such a symmetry modifies the action by the following coupling (we also turn on background $B_3$ for the 2-form symmetry generated by $\oint b^{(1)}$). These backgrounds couple to the symmetry charges for the generators $e^{\frac{Ni}{4\pi} \oint b^{(1)} b^{(1)}}, e^{i \oint b^{(1)}}$ respectively, and thus they modify the action as

$$\int \frac{N}{4\pi} b^{(1)} b^{(1)} B_1 + \int b^{(1)} B_3 , \qquad (78)$$

where we normalized $\oint B_1, \oint B_3 = 0, 1, \cdots, N-1 \bmod N$. We now investigate the relation between the background gauge fields, which manifest in the fusion algebra, by demanding the above coupling to be consistent. In particular, if we extend the fields to a 5+1d bulk, the dynamical field $b^{(1)}$ should be independent of the bulk extension. The bulk dependence is given by

$$\int_{6d} \frac{N}{2\pi} b^{(1)} db^{(1)} B_1 + \pi \frac{N b^{(1)}}{2\pi} \frac{N b^{(1)}}{2\pi} \frac{dB_1}{N} + db^{(1)} B_3 + b^{(1)} dB_3$$
$$= \pi \int_{6d} \frac{N b^{(1)}}{2\pi} \frac{N b^{(1)}}{2\pi} \frac{dB_1}{N} + b^{(1)} dB_3 , \qquad (79)$$

where we simplified the first line using the property that $\frac{db^{(1)}}{2\pi}, \frac{N}{2\pi} b^{(1)}$ have integer periods.

We further simplify the equation using the identity

$$
\pi \int \frac{Nb^{(1)}}{2\pi} \frac{Nb^{(1)}}{2\pi} \frac{dB_1}{N} = \pi \int Sq^2 \left( \frac{Nb^{(1)}}{2\pi} \right) \frac{dB_1}{N}
$$

$$
= \pi \int Sq^2 \left( \frac{Nb^{(1)}}{2\pi} \frac{dB_1}{N} \right) + \left( Sq^1 \frac{Nb^{(1)}}{2\pi} \right) \left( Sq^1 \frac{dB_1}{N} \right) + \frac{Nb^{(1)}}{2\pi} Sq^2 \left( \frac{dB_1}{N} \right)
$$

$$
= \pi \int w_2(TM) \left( \frac{Nb^{(1)}}{2\pi} \frac{dB_1}{N} \right) + \frac{Nb^{(1)}}{2\pi} \left( \frac{dB_1}{N} \frac{dB_1}{N} \right) \, , \tag{80}
$$

where we used the property $Sq^2(z) = z^2$ for $\mathbb{Z}_2$ 2-form $z$, the Cartan formula $Sq^2(xy) = Sq^2 x y + Sq^1 x Sq^1 y + x Sq^2 y$ with $Sq^1 \frac{dB_1}{N} = 0$, and the Wu formula $Sq^2 x_4 = w_2(TM)x_4$ for $\mathbb{Z}_2$ 4-cocycle $x_4 = \frac{Nb^{(1)}}{2\pi} \frac{dB_1}{N}$ mod 2. Thus for the bulk term to be independent of the dynamical field $b^{(1)}$, the backgrounds should satisfy

$$
dB_3 = \frac{N}{2} \left( w_2(TM) \frac{dB_1}{N} + \frac{dB_1}{N} \frac{dB_1}{N} \right) \text{ mod } N \equiv \Theta_4 \, . \tag{81}
$$

Let us relate the above equation to the fusion algebra of the 0-form symmetry. The relation between the background gauge fields implies the background gauge transformations (for simplicity, we omit the transformation of $w_2(TM)$)

$$
B_1 \to B_1 + NC_1,
$$

$$
B_3 \to B_3 + \frac{N}{2} \left( w_2(TM)C_1 + C_1 dC_1 \right) \text{ mod } N \, , \tag{82}
$$

where we take a lift of the $\mathbb{Z}_N$ background gauge field $B_1$ in $\mathbb{Z}$, and we find that changing the lift by $NC_1$, which is a 0-form background gauge transformation, produces an additional background for the 2-form symmetry. This agrees with the previous finding that fusing $N$ of the $\mathbb{Z}_N$ 0-form symmetry domain wall defects gives a 2-form symmetry defect.

The second term on the right hand side in Eq. 81 can be explained as follows. Consider the junction of five domain walls that generate the $\mathbb{Z}_N$ 0-form symmetry. We first fuse two defects to form a codimension-two junction of three domain walls, then we add another domain wall to form a codimension-three junction of four domain walls, and adding another domain wall gives codimension-four junction of five domain walls. The term implies that the codimenion-four locus, i.e. one-dimensional, emits an electric loop. If the domain walls are $g_1, g_2, g_3, g_4, -(g_1 + g_2 + g_3 + g_4) \in \mathbb{Z}_N$, then the electric loop is $\frac{N}{2} \left( (g_1 + g_2 - [g_1 + g_2])/N \right) \left( (g_3 + g_4 - [g_3 + g_4])/N \right)$, where we used the addition as the group multiplication in $\mathbb{Z}_N$, and $[x] = x$ mod $N$.

The first term on the right hand side of Eq. 81 has the following interpretation. If the theory is consistent as a bosonic theory and the symmetries are internal symmetries, then the fermion parity symmetry should act trivially. Suppose $w_2(TM) = d\rho$, $\rho$ is a $\mathbb{Z}_2$ 1-form and it can be viewed as the background gauge field for the fermion parity symmetry. Then we find that the first term is equivalent to the shift $B_3 \to B_3 + \frac{N}{2}\rho\frac{dB_1}{N}$. If $B_1 = 0$, then the fermion parity acts trivially, as $\rho$ is decoupled from the rest of the theory. If there is non-zero $B_1$, the fermion parity couples as follows. Consider a junction of three domain walls that generate the $\mathbb{Z}_N$ 0-form symmetry, labelled by $g_1, g_2, -(g_1 + g_2)$, and the junction has codimension two. Then, we intersect the junction with the domain wall of the fermion parity symmetry, and obtain a new junction of codimension-three, *i.e.* two-dimensional in space time. The junction contain the electric membrane operator labelled by $\frac{N}{2}(g_1 + g_2 - [g_1 + g_2])/N$. Now, if we braid the electric membrane supported at the junction with the magnetic loop labelled by $q_m$, we find the sign $(-1)^{q_m(g_1+g_2-[g_1+g_2])/N}$. If the theory is bosonic and the symmetries are internal, then the fermion parity domain wall should act trivially on the theory; here we find that this is not the case. Concretely, if we take $g_1 = N - 1, g_2 = 1$, then $(g_1 + g_2 - [g_1 + g_2])/N = 1$. Thus, piercing the magnetic loop $q_m$ with the domain wall labelled by $N$ (more precisely, the domain wall obtained by fusing the $N-1$ and 1 domain wall) gives a point-like excitation with self-statistics $\pi q_m$, and when $q_m$ is odd, the point-like excitation is a fermion: when acted by the fermion parity domain wall, it produces a sign.

The relation (81) between the background gauge fields implies that we not only gauge the $\mathbb{Z}_N$ symmetry generated by $T$ with the gauge field $B_1$, but we must also gauge the 2-form symmetry with gauge field $B_3$. Thus, the right hand side of Eq. 81, denoted by $\Theta_4$, can be thought of as an obstruction to only gauging the 0-form symmetry. It is not an 't Hooft anomaly of the 0-form symmetry (whose group-cohomology classification is given by $H^6(BSO(5)_{\text{Lorentz}} \times B\mathbb{Z}_N, U(1))$); rather, it is a modification of the symmetry group [37]. The obstruction

$$\Theta_4 \equiv \frac{N}{2}\left(w_2(TM)\frac{dB_1}{N} + \frac{dB_1}{N}\frac{dB_1}{N}\right), \quad dB_3 = \Theta_4 , \tag{83}$$

is a beyond-group-cohomology $H^4$ obstruction that lives in $H^4(BSO(5)_{\text{Lorentz}} \times B\mathbb{Z}_N, \mathbb{Z}_N)$: if it belongs to the non-trivial class, one cannot remove it by redefining $B_3$. In other words, the symmetry is a 3-group symmetry that combines the emergent Lorentz symmetry, the

0-form loop permutation symmetry and the 2-form symmetry.

## B. Extend $\mathbb{Z}_N$ 0-form symmetry to $\mathbb{Z}_{2N}$: no obstruction

We now consider the extension $\mathbb{Z}_{2N}$ symmetry which is the extension of the $\mathbb{Z}_N$ symmetry generated by $T$, by the non-permutation symmetry generated by the domain wall defect $D^N$. As discussed earlier, the symmetry can be gauged. Indeed, the $H^4$ obstruction becomes trivial under the pullback for the odd $N$ projection map, $\mathbb{Z}_{2N} \to \mathbb{Z}_N$.

The $\mathbb{Z}_2$ symmetry generated by $D^N$ does not permute the loops but it leads to a symmetry fractionalization on loops: as explained before, when a magnetic loop passes through the domain wall, its framing dependence is changed by stacking with the worldline of an extra spin-1/2 particle attached to the loop. Since such a change of framing dependence involves the projective representation of the Lorentz group, the symmetry fractionalization is a "beyond group cohomology" fractionalization for the Lorentz symmetry [40].

## C. Gauging the $m \to \psi$ symmetry as $\mathbb{Z}_{2N}$ symmetry

We now gauge the $\mathbb{Z}_N$ symmetry corresponding to the $T$ transformation for even $N$. As discussed earlier, wenot gauge the $\mathbb{Z}_N$ quotient symmetry without also gauging the 2-form symmetry, but we gauge the extension $\mathbb{Z}_{2N}$. For even $N$, the $T$ transformation generates a symmetry that participates in a 3-group. Thus wenot only gauge the 0-form symmetry. Instead, we extend the $\mathbb{Z}_N$ transformation to $\mathbb{Z}_{2N}$ and gauge the symmetry. It does not participate in a 3-group, $i.e.$ there is no $H^4$ obstruction.

We introduce dynamical $\mathbb{Z}_{2N}$ gauge field $u$, written as a pair of $U(1)$ 1-form gauge field $u$ and $U(1)$ 3-form gauge field $v$ with action $\frac{2N}{2\pi}udv$. The gauge field $u$ couples to the domain wall defect which generates the 0-form symmetry:

$$\frac{N}{2\pi}b^{(1)}db^{(2)} + \frac{N}{4\pi}b^{(1)}b^{(1)}\frac{2Nu}{2\pi} + \frac{2N}{2\pi}udv \; . \tag{84}$$

The equation of motions are

$$\frac{N}{2\pi}db^{(2)} + \frac{N^2}{2\pi^2}b^{(1)}u = 0$$
$$-\frac{N}{2\pi}db^{(1)} = 0$$
$$\frac{2N}{2\pi}du = 0$$
$$\frac{N^2}{4\pi^2}b^{(1)}b^{(1)} + \frac{2N}{2\pi}dv = 0 \ . \tag{85}$$

Thus, the theory has non-Abelian surface and volume operators:

$$e^{i\oint_\Sigma b^{(2)} + \frac{N}{\pi}i\int_\mathcal{V} b^{(1)}u}, \quad e^{i\oint_{\mathcal{V}_3} v + i\frac{N}{4\pi}\int_{\mathcal{V}_4} b^{(1)}b^{(1)}} \ , \tag{86}$$

where the two-dimensional surface $\Sigma = \partial\mathcal{V}$ is on the boundary of the three-dimensional volume $\mathcal{V}$, and the 3-dimensional volume $\mathcal{V}_3 = \partial\mathcal{V}_4$ is on the boundary of the four-dimensional submanifold $\mathcal{V}_4$. The parts that are supported on $\mathcal{V}$ and $\mathcal{V}_4$, indicate that the surface operator and the volume operator depends on additional choices, and this leads to a non-Abelian fusion algebra and quantum dimension greater than one, as we discuss below.

We focus on the theory $N = 2$. Consider the surface operator $U = e^{i\oint_\Sigma b^{(2)} + \frac{2}{\pi}i\int_\mathcal{V} b^{(1)}u}$. We derive the fusion algebra of the surface operator using the property that the zero mode gauge transformation of $u$ (*i.e.* a global symmetry transformation) transforms $b^{(2)}$ by $b^{(2)} + b^{(1)}$ but leaves the gauge field $u$ invariant. Similarly, for any surface $\Sigma$ and a closed curve $\gamma \subset \Sigma$, we perform a global $\mathbb{Z}_2$ 1-form gauge transformation on $b^{(1)}$ with the $\mathbb{Z}_2$ 1-form parameter $\pi\delta(\gamma)^\perp$ where $\delta(\gamma)^\perp$ is the Poincare dual 1-form of the closed curve $\gamma$ with respect to the surface $\Sigma$, which leaves $b^{(1)}$ invariant. This leads to the following fusion algebra

$$U(\Sigma) \times W(\Sigma) = U(\Sigma), \quad U(\Sigma) \times V(\gamma) = U(\Sigma),$$
$$U(\Sigma) \times U(\Sigma) = \frac{|H^0(\Sigma, \mathbb{Z}_2)|}{|H^1(\Sigma, \mathbb{Z}_2)|} \sum_{\gamma \in H_1(\Sigma, \mathbb{Z}_2)} \sum_{\Sigma' \in H_2(\Sigma, \mathbb{Z}_2)} W(\Sigma')V(\gamma),$$
$$W(\Sigma') = e^{i\oint_{\Sigma'} b^{(1)}}, \quad V(\gamma) = e^{2i\oint_\gamma u} \ , \tag{87}$$

where $\times$ indicates the fusion and $+$ indicates the direct sum of multiple outcomes from the fusion.[11] Thus we conclude that the loop excitation described by surface operator $U$ obeys non-Abelian fusion rule, where $W$ is the $\mathbb{Z}_2$ electric surface operator. Note that when

---

[11] We remark that the first equation can be interpreted as $W$ condensed on $U$. Similar fusion algebra in 3+1d with $U$ replaced by a codimension-one non-invertible domain wall is discussed recently in [41, 42].

$\Sigma'$ is the trivial element in $H_2(\Sigma, \mathbb{Z}_2)$, $e^{i \oint_{\Sigma'} b^{(1)}}$ should be treated as the identity operator. The surface operator $W$ has $(-1)$ mutual braiding with the non-Abelian surface operator $U$ which is identified with the magnetic surface operator before we gauge the $\mathbb{Z}_{2N}$ symmetry.

## VII. A GAPLESS BOUNDARY AND THE CLASSIFICATION OF GAPPED BOUNDARIES

We consider the 4+1d gauge theory $\mathcal{L} = \frac{N}{2\pi} b^{(1)} db^{(2)}$ residing on the half-infinite space with $w < 0$. This system has a boundary at $w = 0$. In the bulk, the equation of motion enforces $db^{(1)} = db^{(2)} = 0$. The solution is given by $b^{(1)} = d\phi, b^{(2)} = d\theta$ where $\phi$ and $\theta$ should be viewed as U(1) 1-form gauge fields. When there is a boundary, we consider the boundary condition $b^{(1)}| = d\phi|$, $b^{(2)}| = d\theta|$, where $|$ means restricted to the boundary (we omit $|$ in the following discussion when the context is clear). In the bulk, $\phi$ and $\theta$ do not represent physical degrees of freedom as they simply parametrize the gauge transformation of the 2-form gauge fields $b^{(1)}$ and $b^{(2)}$. However, we disallow gauge transformation of $b^{(1)}$ and $b^{(2)}$ on the boundary here to keep the entire system gauge-invariant without changing the boundary condition; those gauge transformations that do not vanish on the boundary are global symmetries on the boundary. Consequently, $\phi$ and $\theta$ on the boundary are in fact physical degrees of freedom. We choose the axial gauge for the 2-form gauge fields, *i.e.* $b_{ti}^{(1)} = b_{ti}^{(2)} = 0$ for $i = x, y, z, w$, and further gauge-fix the U(1) gauge fields such that $\phi_t = \theta_t = 0$. After the gauge fixing, plugging the solution $b^{(1)} = d\phi$, $b^{(2)} = d\theta$ into the bulk action $\int_{w<0} \frac{N}{2\pi} b^{(1)} db^{(2)}$ yields a boundary term $\int d^3\vec{r} dt \frac{N}{2\pi} \epsilon_{\alpha\beta\gamma} \partial_t \phi_\alpha \partial_\beta \theta_\gamma \Big|_{w=0}$ with $\alpha, \beta, \gamma = x, y, z$. This boundary term dictates, after canonical quantization, the commutation relation between the fields $\phi$ and $\theta$. In addition, we consider extra kinetic energy terms and write down the following Lagrangian density for the boundary

$$\mathcal{L}_{bdy} = \frac{N}{2\pi} \epsilon_{\alpha\beta\gamma} \partial_t \phi_\alpha \partial_\beta \theta_\gamma - v(\partial_\alpha \phi_\beta - \partial_\beta \phi_\alpha)^2 - v(\partial_\alpha \theta_\beta - \partial_\beta \theta_\alpha)^2. \tag{88}$$

The last two terms are the kinetic energies of $\phi$ and $\theta$ fields with $v$ the velocity of the boundary modes. We also consider the more general case where the velocities of $\phi$ and $\theta$ are different. Such generalization does not lead to a conceptual change to the physics described below. Hence, we continue the discussion keeping the velocity of the two gauge fields the same. The form of the kinetic energies are constrained by the residue gauge transformation

$\phi_\alpha \to \phi_\alpha + \partial_\alpha f^{(1)}$ and $\theta_\alpha \to \theta_\alpha + \partial_\alpha f^{(2)}$ for time-independent scalar functions $f^{(1)}$ and $f^{(2)}$. The boundary theory (88) is a 3+1d generalization of the 1+1D chiral Luttinger liquid that resides on the boundary of the 2+1d 1-form Chern-Simons theory $\frac{N}{2\pi} a^{(1)} da^{(2)}$. One can easily check that the boundary theory $\mathcal{L}_{bdy}$ has a gapless spectrum with a linear dispersion.

When we integrate out the field $\theta$ by "completing the square" in $\mathcal{L}_{bdy}$, the resulting Lagrangian is

$$\mathcal{L}_{\mathrm{Maxwell}}[\phi] = \frac{N^2}{16\pi^2 v}(\partial_t \phi)^2 - v(\partial_\alpha \phi_\beta - \partial_\beta \phi_\alpha)^2, \tag{89}$$

which is exactly the gauged-fixed version of the 3+1d Maxwell theory of the 1-form U(1) gauge field $\phi$. Similar, if we instead integrate out $\phi$ in $\mathcal{L}_{bdy}$, the resulting theory is the gauged-fixed version of the Maxwell theory of the 1-form U(1) gauge field $\theta$. In fact, the Maxwell theory of $\phi$ and that of $\theta$ are dual to each other via electromagnetic duality. The gauge fields $\phi$ and $\theta$ are dual U(1) gauge fields. Interestingly, as we see after the canonical quantization of $\mathcal{L}_{bdy}$, for integer $|N| > 1$, the electromagnetic duality between $\phi$ and $\theta$ is a fractionalized version of the electromagnetic duality in the standard 3+1d Maxwell theory. The fractionalization in $\mathcal{L}_{bdy}$ is consequence of the bulk topological order given by the 4+1d 2-form Chern-Simons theory $\mathcal{L} = \frac{N}{2\pi} b^{(1)} db^{(2)}$.

Via canonically quantizing the boundary theory $\mathcal{L}_{bdy}$ in the Coulomb gauge , *i.e.* $\partial_\alpha \phi_\alpha = \partial_\alpha \theta_\alpha = 0$, we obtain the commutation relation

$$[\phi_\alpha(r), \theta_\gamma(r')] = -\frac{2\pi i}{N} \frac{\epsilon_{\alpha\beta\gamma}(r_\beta - r'_\beta)}{4\pi|r - r'|^3}, \tag{90}$$

where $r$ and $r'$ represent the 3-dimensional spatial coordinates on the boundary. We define the elementary Wilson loop operator $W_{(1,0)}(C)$ and $W_{(0,1)}(C)$ of the gauge fields $\phi$ and $\theta$ along a one-dimensional closed loop $C$ in the 3-dimensional space on the boundary:

$$W_{(1,0)}(C) \equiv \exp\left(i \int_C \phi\right), \quad W_{(0,1)}(C) \equiv \exp\left(i \int_C \theta\right). \tag{91}$$

The commutation relation between them is given by

$$W_{(1,0)}(C)W_{(0,1)}(C') = e^{i\frac{2\pi}{N}\mathrm{Link}_{3D}(C,C')} W_{(0,1)}(C')W_{(1,0)}(C) \tag{92}$$

where $\mathrm{Link}_{3D}(C, C')$ is the mutual linking number between the one-dimensional closed loops $C$ and $C'$ in the 3-dimensional space. The mutual linking number $\mathrm{Link}_{3D}(C, C')$ is defined by the integral

$$\mathrm{Link}_{3D}(C, C') = \oint_C dr_\alpha \oint_{C'} dr'_\gamma \frac{\epsilon_{\alpha\beta\gamma}(r_\beta - r'_\beta)}{4\pi|r - r'|^3} \tag{93}$$

which always carries integer values. The commutation relation (92) suggests that the Wilson loop operator $W_{(1,0)}$ carrying unit electric charge under the U(1) gauge field $\phi$ can be identified as the 't Hooft loop carrying $\frac{1}{N}$ magnetic charge of $\theta$ which is the dual to $\phi$. Similarly, the Wilson loop $W_{(0,1)}$ carrying unit electric charge under $\theta$ can be identified with the 't Hooft loop carrying $\frac{1}{N}$ magnetic charge of $\phi$. The existence of fractionally charged 't Hooft loops in the boundary theory (88) is a direct consequence of the non-trivial topological order in the 4+1d bulk. The Wilson loop operators $W_{(1,0)}$ and $W_{(0,1)}$ can also be identified respectively as the creation (or annihilation) operators of the $e$ and $m$ loops on the boundary. The commutation relation (92) captures the non-trivial braiding statistics between the $e$ and $m$ loop discussed earlier.

We consider the general form of Wilson line operators $W_{(q_e, q_m)}$ that carries $q_e$ electric charge under the gauge field $\phi$ and $q_m$ electric charge under $\theta$. Even though the fields $q_e \phi + q_m \theta$ at different (boundary) coordinates $r$ commute with each other, the singularity in $[\phi(r), \theta(r')]$ as $r' \to r$ still requires us to define the Wilson loop $W_{(q_e, q_m)}$ in the path-ordered fashion:

$$W_{(q_e, q_m)}(C) \equiv \mathcal{P} \exp \left\{ i \oint_C (q_e \phi + q_m \theta) \right\} \tag{94}$$

where $\mathcal{P}$ indicates the path ordering along $C$. The choice of the starting point of $C$ in the path ordering in fact does not affect the definition of $W_{(q_e, q_m)}$ because one can show that

$$W_{(q_e, q_m)}(C) = \left[ W_{(1,0)}(C) \right]^{q_e} \left[ W_{(0,1)}(C) \right]^{q_m} e^{-i \frac{2\pi}{2N} q_e q_m \mathrm{Link}_{3D}(C,C)}. \tag{95}$$

No path ordering is required for the definition of $W_{(1,0)}(C)$ and $W_{(0,1)}(C)$. Interestingly, the self-linking number $\mathrm{Link}_{3D}(C, C)$ of the closed loop $C$ occurs on the right hand side of (95). The self-linking number of $C$ depends on the framing of $C$. It can be obtained as the mutual linking number between the closed loop $C$ and a push-off copy of $C$ obtained from displacing each point of $C$ along the normal direction specified by the framing. A framed closed loop $C$ can be viewed as closed ribbon whose edges are given by $C$ and its push-off copy. In $W_{(q_e, q_m)}(C)$ with a framed loop $C$, the gauge fields $\phi$ and $\theta$ should be integrated along $C$ and its push-off copy respectively. Note that the framing dependence of $W_{(q_e, q_m)}$ only enters through the self-linking number $\mathrm{Link}_{3D}(C, C)$. The elementary Wilson loops $W_{(1,0)}(C)$ and $W_{(0,1)}(C)$ remain independent from the framing of $C$. $W_{(q_e, q_m)}$ can be viewed as the creation (or annihilation operator) of the composite loop excitation that consists of $q_e$ copies of the

$e$ loops and $q_m$ copies of the $m$ loops. For a generic integer $N$, the framing dependence of $W_{(q_e, q_m)}$ arise from the coupling of the composite loop to the bulk 2-form gauge fields.

When $N = 1$, we encounter a special case in which the 4+1d bulk Chern-Simons theory $\mathcal{L} = \frac{1}{2\pi}b^{(1)}db^{(2)}$ is free of any non-trivial loop excitations and the boundary theory is reduced to the ordinary 3+1d Maxwell theory. Yet, the framing dependence of $W_{(q_e, q_m)}$ shown in (95) is still non-trivial for $N = 1$. Such framing dependence can be naturally explained as follows. The independence of the elementary loops $W_{(1,0)}$ and $W_{(0,1)}$ on the framing indicates that, in this ordinary Maxwell theory, the particles carrying only the electric charges of $\phi$ and those carrying only electric charges $\theta$ are bosonic particles. By the electromagnetic duality, we identify the electric charge of $\theta$ as the magnetic charge of $\phi$ (and vice versa). The framing dependence $e^{-i\frac{2\pi}{2N}q_e q_m \text{Link}_{3D}(C,C)}$ of $W_{(q_e, q_m)}(C)$, which takes value $\pm 1$ when $q_e q_m$ is odd, exactly captures the fermionic nature of the charge-$(q_e, q_m)$ dyons of the ordinary 3+1d Maxwell theory whose electric and magnetic charge are both odd.

As discussed earlier, for the 2-form Chern-Simons theory $\mathcal{L} = \frac{N}{2\pi}b^{(1)}db^{(2)}$ in the 4+1d bulk, the transformations $S$ and $T$ shown in Eq. 65 generates an $SL(2, \mathbb{Z}_{2N})$ or $SL(2, \mathbb{Z}_N)$ loop permutation symmetry for even or odd $N$. These symmetry transformations in bulk also induce the following $S$ and $T$ transformation on the boundary:

$$
\begin{aligned}
S: & \quad \phi \to \theta, \quad \theta \to -\phi \\
T: & \quad \phi \to \phi, \quad \theta \to \theta + \phi .
\end{aligned}
\tag{96}
$$

The theory $\mathcal{L}_{bdy}$ shown in Eq. (88) is only invariant under the $S$ transformation. Under the $T$ transformation, the first term of $\mathcal{L}_{bdy}$, the term that dictates the commutation relation between $\phi$ and $\theta$, stays invariant, while the last two terms, namely the kinetic energies of $\phi$ and $\theta$, transform non-trivially. Starting from $\mathcal{L}_{bdy}$, one can generate an $SL(2, \mathbb{Z})$ class of boundary theories via different combination of the $S$ and $T$ transformations. When $N = 1$, the boundary theory $\mathcal{L}_{bdy}$ can be viewed as the ordinary 3+1d Maxwell theory of which the $S$ and $T$ transformations generate the standard $SL(2, \mathbb{Z})$ duality group.

In the following, we use this gapless boundary theory $\mathcal{L}_{bdy}$ as a parent theory to study the gapped boundaries of the 4+1d 2-form Chern-Simons theory $\mathcal{L} = \frac{N}{2\pi}b^{(1)}db^{(2)}$ for a general $N$. To drive the gapless boundary theory described by (88) into fully gapped boundary phases, we consider Higgsing the gauge fields $\phi$, $\theta$, or their linear combinations.

In the boundary phase where we higgs the gauge field $\phi$, the vacuum expectation value

$\langle W_{(1,0)}(C) \rangle$ follows the "perimeter law", namely $\log \langle W_{(1,0)}(C) \rangle$ scales as the length of the loop $C$. It implies that $e$ loop excitation, which can be created by $W_{(1,0)}$ on the boundary, is now condensed on the boundary. Due to the commutation relation (90), any Wilson loop operator $W_{(q_e, q_m)}$ with $q_m \neq 0$ should follow the "area law" in this phase, $i.e.$ $\log \langle W_{(q_e, q_m)}(C) \rangle$ scales as the area of a 2-dimensional surface whose boundary is given by the loop $C$. In other words, the gauge fields $q_e \phi + q_m \theta$ with $q_m \neq 0$ are confined on the boundary. It implies that the $m$ loop becomes a confined excitation as it is moved towards the boundary. Since all the U(1) gauge fields on the boundary are either Higgsed or confined, the boundary now becomes fully gapped in this phase.

Similarly, we also consider the boundary phase associated with the Higgs phase of $\theta$. In this case, all U(1) gauge fields $q_e \phi + q_m \theta$ with $q_e \neq 0$ are confined. This boundary phase is again fully gapped. From the loop excitation perspective, this boundary phase can be understood as the condensate of the $m$ loop where the $e$ loop is now confined.

More generally, we consider the Higgs phase of the composite U(1) gauge field $p\phi + q\theta$ with $p, q \in \mathbb{Z}$. We argue that a consistent gapped boundary without leftover boundary degrees of freedom requires that (1) the two integers $p$ and $q$ are coprime and (2) $Npq$ is even. The first condition ensures that once $p\phi + q\theta$ is Higgsed, all U(1) gauge fields on the boundary are either Higgsed or confined leaving no residual boundary degrees of freedom. The second condition is due to the non-trivial framing dependence of the Wilson loop operators. When the composite U(1) gauge field $p\phi + q\theta$ is Higgsed, the boundary phase is a condensate of the composite loops, each of which consists of $p$ copies of $e$ loops and $q$ copies of $m$ loops. We expect the vacuum expectation value $\langle W_{(p,q)}(C) \rangle$ to not only follow the perimeter law but also the framing dependence given by $\langle W_{(p,q)}(C) \rangle \propto e^{-i \frac{2\pi}{2N} pq \mathrm{Link}_{3D}(C,C)}$. Such a framing dependence of $\langle W_{(p,q)}(C) \rangle$ arises from the coupling of the composite loop to the bulk 2-form gauge fields. However, for any coprime pair $(p, q)$, the framing dependence of $\langle W_{(Np, Nq)}(C) \rangle$ should be trivial because the composite loop consisting of $Np$ copies of $e$ loops and $Nq$ copies of $m$ loops is a topologically trivial excitation in the bulk. On the other hand, $\langle W_{(p,q)}(C) \rangle \propto e^{-i \frac{2\pi}{2N} pq \mathrm{Link}_{3D}(C,C)}$ implies that $\langle W_{(Np, Nq)}(C) \rangle \propto e^{-i \frac{2\pi}{2} Npq \mathrm{Link}_{3D}(C,C)}$. Hence, a consistent Higgs phase of $p\phi + q\theta$ requires $Npq$ to be even. Conceptually, we think of the Higgs phase of $p\phi + q\theta$ as a condensate of the framed loops created (or annihilated) by $W_{(p,q)}$. Loops with different framing should be considered as different loop configurations and, hence, are allowed to have different framing-dependent expectation values. The consistency of the

framing-dependent expectation values of loops requires $Npq$ to be even.

When we take $N = 2$, the bulk theory can be identified as the 4+1d $\mathbb{Z}_2$ toric code model. The boundary Higgs phase with $p = q = 1$ is identified with the condensate of the $\psi$ loop. The framing dependence $\langle W_{(1,1)} \rangle \propto e^{-i\frac{2\pi}{4}\text{Link}_{3D}(C,C)}$ is consistent with the lattice construction of this boundary phase based on the semion Walker-Wang model discussed in Sec. III. For $N = 1$, the requirement that $Npq$ should be even rules out the possibility of Higgsing the composite field $p\phi + q\theta$ with odd $p$ and $q$. This requirement is consistent with the fact that the boundary theory (88) with $N = 1$ is equivalent to an ordinary 3+1d Maxwell theory where the dyons carrying both odd electric charges and odd magnetic charges are fermions.

For the 4+1d 2-form Chern-Simons theory $\mathcal{L} = \frac{N}{2\pi}b^{(1)}db^{(2)}$ with a general integer $N$, we classify the gapped boundaries captured by the Higgs phases of the parent boundary theory (88) using the coprime integer pair $(p,q)$ such that $Npq$ is even. For the classification, we have the following equivalence relation. Two pairs $(p,q)$ and $(p',q')$ yield topologically equivalent gapped boundaries if (1) $(p,q) = \pm(p',q')$ mod $N$ and (2) $pq = p'q'$ mod $2N$. The reasoning for these two equivalence conditions are as follows. $(p,q)$ and $(p',q')$ mod $N$ indicate the types of composite loop excitation that are condensed on the two boundaries. Note that when a composite loop $(p,q)$ (or $(p',q')$) is condensed, all loops of types $(np, nq)$ (or $(np', nq')$) for $n \in \mathbb{Z}$ should be considered as condensed. Different condensates of topologically distinct sets of composite loops should yield topologically distinguishable gapped boundaries. When $(p,q) = \pm(p',q')$ mod $N$ but $pq \neq p'q'$ mod $2N$, even though the condensed loops are of the same type in the two boundaries, the framing dependence of $\langle W_{(p,q)} \rangle$ and $\langle W_{(p',q')} \rangle$, which are given by $e^{-i\frac{2\pi}{2N}pq\text{Link}_{3D}(C,C)}$ and $e^{-i\frac{2\pi}{2N}p'q'\text{Link}_{3D}(C,C)}$, in the two condensates are different from each other by $(-1)^{\text{Link}_{3D}(C,C)}$; in other words, the framing dependence differs by that of a fermion . Hence, the types of boundary should still be viewed as distinct gapped boundary phases. With the equivalence relation between these Higgs phases defined, we classify the gapped boundaries of the 4+1d 2-form Chern-Simons theory $\mathcal{L} = \frac{N}{2\pi}b^{(1)}db^{(2)}$. For $N = 1$, there is only 1 type of gapped boundary. This statement is consistent with the fact that different Higgs and confined phases of the standard 3+1d Maxwell theory (in the absence of any higher-form symmetry) are topologically identical. Interestingly, for $N = 2$, there are 6 different types of gapped boundaries labeled by $(p,q) = (0,1), (1,0), (1,1), (1,2), (1,3), (2,1)$. This result is consistent with the analysis in Sec. III. For a general $N$, this classification of gapped boundaries of 4+1d 2-form Chern-

Simons theory $\mathcal{L} = \frac{N}{2\pi} b^{(1)} db^{(2)}$ discussed above using the fractional Maxwell theory as a parent theory is consistent with the classification of gapped boundaries based on polarizations as discussed in [30].

## VIII.   SUMMARY

In this work, we have shown that all loop excitations of the 4+1d loop-only $\mathbb{Z}_2$ toric code, whose low energy description is the 4+1d 2-form $\mathbb{Z}_2$ gauge theory, have trivial self-statistics. This was proven using an explicit construction of a unitary circuit that maps the dyonic loop excitation to the magnetic loop excitation which is known to have trivial self-statistics. The $\mathbb{Z}_2$ symmetry that exchanges the two loops has an obstruction and that implies the symmetry cannot be gauged by itself, unless extended to a $\mathbb{Z}_4$ symmetry. We show that this result generalizes to $\mathbb{Z}_N$ gauge theories as well, with the $\mathbb{Z}_N$ loop permutation symmetry having obstruction for even $N$ and free of obstruction for odd $N$. The permutation circuit is constructed using insights on gapped boundaries and invertible domain walls of the theory.

Recently, Refs. [43] and [44] have constructed lattice Hamiltonian models for a beyond group cohomology invertible phase without symmetry in 4+1d that has a 3+1d boundary with fermionic particle and fermionic loop excitations with mutual $\pi$ statistics. The fermionic loop in Refs. [43] and [44] is different from the dyonic loop discussed here: the fermionic loop there depends on the local $w_3$ structure [45], in contrast to the dyonic loops discussed here (and in Refs. [35]), and thus the fermionic loops in [43, 44] have non-trivial statistics.

We modify the $\mathbb{Z}_2$ 2-form gauge theory into a twisted $\mathbb{Z}_2$ 2-form gauge theory with the topological action

$$\frac{1}{2\pi} b^{(1)} db^{(1)} \ . \tag{97}$$

This is equivalent to $\int b^{(1)} w_3(TM)$ as shown in Refs. [30, 46], and it has the effect of changing the magnetic and dyonic loops described by $e^{i \oint b^{(2)}}, e^{i \oint b^{(1)} + b^{(2)}}$ into the fermionic loops of Refs. [43, 44], while the electric loop described by $e^{i \oint b^{(1)}}$ remains bosonic.[12] The twisted $\mathbb{Z}_2$ 2-form gauge theory can also be described by a local commuting projector Hamiltonian on

---

[12] This is equivalent to activating the background $w_3(TM)$ for the $\mathbb{Z}_2$ 2-form symmetry generated by the electric surface operator, and thus the surface operators that braid non-trivially with it are modified to correspond to fermionic loops following similar argument in [40]. Here we assumed the theory is not enriched by time-reversal symmetry.

the lattice [43]. It would be interesting to investigate the symmetry, boundary condition, domain wall properties etc. of these models in the future.

## ACKNOWLEDGEMENT

We would like to thank Fiona Burnell, Meng Cheng, Lukasz Fidkowski, Jeongwan Haah, Yi Ni, Xiao-Liang Qi, Nathan Seiberg, Shu-Heng Shao, Kevin Walker and Zhenghan Wang for valuable discussions. A.D. thanks Yu-An Chen for the useful discussion on higher cup products. W.S., A.D., and X.C. were supported by the Simons Foundation through the collaboration on Ultra-Quantum Matter (651438, XC, AD), the Walter Burke Institute of Theoretical Physics, the Institute for Quantum Information and Matter, an NSF Physics Frontiers Center (PHY-1733907), the National Science Foundation (DMR-1654340, XC) and the Simons Investigator Award (828078, XC). W.S. is also supported by a grant from the Simons Foundation (651444, WS). The work of P.-S. H. is supported by the U.S. Department of Energy, Office of Science, Office of High Energy Physics, under Award Number DE-SC0011632, by the Simons Foundation through the Simons Investigator Award, and by the Simons Collaboration on Global Categorical Symmetries. C.-M. J. is supported by a faculty startup grant at Cornell University. C. X. is supported by NSF Grant No. DMR-1920434 and the Simons Investigator program.

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

**Appendix A: Review of differential forms**

### 1. Tangent spaces, 1-forms and $m$-forms

Consider a curve in $\mathbb{R}^2$, $C \subset \mathbb{R}^2$ and a point on the curve, $p \in C$. The tangent space to $C$ at $p$, $T_pC$ is the set of all vectors tangent to $C$ at $p$. We assign coordinates to the points on the curve such that each point is mapped to two real numbers via $f_{(x,y)} : C \to \mathbb{R}^2$ such that $f_{(x,y)}(p) = (x(p), y(p))$. Similarly we assign a coordinate system for the vectors in the tangent space at $p$, $T_pC$ such that $f_{\langle dx,dy \rangle} : T_pC \to \mathbb{R}^2$. For example, for $v \in T_pC$, we get $f_{\langle dx,dy \rangle}(\vec{v}) = \langle dx(\vec{v}), dy(\vec{v}) \rangle$ where $dx(\vec{v})$ and $dy(\vec{v})$ are the $x$ and $y$ components of the vector $\vec{v} \in T_pC$ respectively.

A 1-form $\omega_p$ is a linear function from the tangent space $T_p\mathbb{R}^n$ at point $p$ in $\mathbb{R}^n$ to $\mathbb{R}$ i.e. $\omega_p : T_p\mathbb{R}^n \to \mathbb{R}$. This implies that $\omega_p$ is in the dual space of the tangent space i.e. $\omega_p \in (T_p\mathbb{R}^n)^\star$. We suppress the subscript $p$ denoting the point $p$ in $\omega_p$ from here on. In $\mathbb{R}^2$, we write an arbitrary one form as $adx + bdy$ where $a, b \in \mathbb{R}^2$ and $dx$ and $dy$ are elementary 1-forms that obey the definition as mentioned above. Intuitively, this 1-form is a scalar projection onto the line given by vector $\langle a, b \rangle \in T_p\mathbb{R}^2$. In general, in $\mathbb{R}^n$, we write the 1-form $\omega = a_1 dx_1 + a_2 dx_2 + ..... + a_n dx_n$.

In order to define 2-forms and $m$-forms where $m > 2$, we first define a 'wedge product' of two 1-forms, $\omega_1 \wedge \omega_2$ which is a linear function from two vectors in the tangent space at $p$ to $\mathbb{R}$ i.e. $\omega_1 \wedge \omega_2 : T_p\mathbb{R}^n \times T_p\mathbb{R}^n \to \mathbb{R}$. The action of this wedge product on two vectors $\vec{v}_1, \vec{v}_2 \in T_p\mathbb{R}^n$ is given by

$$\omega_1 \wedge \omega_2(\vec{v}_1, \vec{v}_2) = \text{Det} \begin{pmatrix} w_1(\vec{v}_1) & w_2(\vec{v}_1) \\ w_1(\vec{v}_2) & w_2(\vec{v}_2), \end{pmatrix} \tag{A1}$$

where Det denotes the determinant of the $2 \times 2$ matrix in the bracket. In $\mathbb{R}^2$, this determinant is exactly the signed area of the parallelogram spanned by the two vectors $\langle w_1(\vec{v}_1), w_2(\vec{v}_1) \rangle$ and $\langle w_1(\vec{v}_2), w_2(\vec{v}_2) \rangle$. Taking $w_1 = dx$ and $w_2 = dy$ and $\vec{v}_1, \vec{v}_2 \in T_p\mathbb{R}^2$, we get $dx \wedge dy(\vec{v}_1, \vec{v}_2) = \text{Det} \begin{pmatrix} \vec{v}_1 \\ v_2 \end{pmatrix}$ which is the area of the parallelogram spanned by the vectors $\vec{v}_1$ and $\vec{v}_2$. Similarly we define a wedge product of $m$ 1-forms, $\omega_1 \wedge \omega_2.... \wedge \omega_m$ which is a linear function from $m$ vectors in the tangent space at $p$ to $\mathbb{R}$ i.e. $\omega_1 \wedge \omega_2.... \wedge \omega_m : (T_p\mathbb{R}^n)^m \to \mathbb{R}$.

The action of $\omega_1 \wedge \omega_2 .... \wedge \omega_m$ on vectors $\vec{v}_1$, $\vec{v}_2, ... \vec{v}_m$ is given as follows

$$\omega_1 \wedge \omega_2 .... \wedge \omega_m(\vec{v}_1, \vec{v}_2, ..., \vec{v}_m) = \text{Det} \begin{pmatrix} w_1(\vec{v}_1) & w_2(\vec{v}_1) & . & . & w_m(\vec{v}_2) \\ w_1(\vec{v}_2) & w_2(\vec{v}_2) & . & . & w_m(\vec{v}_2) \\ . & . & . & . & . \\ . & . & . & . & . \\ w_1(\vec{v}_m) & w_2(\vec{v}_m) & . & . & w_m(\vec{v}_m) \end{pmatrix}. \tag{A2}$$

The wedge products of 1-forms in Eq. A1 and Eq. A2 also define 2-forms and $m$-forms respectively. In general, every $m$-form $\omega^{(m)}$ on $T_p\mathbb{R}^n$ can be expressed as

$$\omega^{(m)} = \sum_{1 \leq i_1 < i_2 < ... < i_m \leq n} a_{i_1, i_2, ..., i_m} dx_1 \wedge dx_2 \wedge ... \wedge dx_m, \tag{A3}$$

which we also write concisely as

$$\omega^{(m)} = \sum_I a_I dx_I \tag{A4}$$

where $I = (i_1, i_2, ..., i_m)$ and $1 \leq i_1 < i_2 < ... < i_m \leq n$. Given this form, we write the action of $m$-form $\omega^{(m)}$ on $m$ vectors $\vec{v}_1$, $\vec{v}_2, ..., \vec{v}_m$ as follows

$$\omega(\vec{v}_1, \vec{v}_2, ..., \vec{v}_m) = \sum_{1 \leq i_1 < i_2 < ... < i_m \leq n} a_{i_1, i_2, ..., i_m} \text{Det}\left(v_j^{i_1, i_2, ..., i_m}\right) \tag{A5}$$

where $v_j^{i_1, i_2, ..., i_m}$ is an $m \times m$ matrix made of elements of vectors $\vec{v}_j$, as implied through Eq. A2.

An important algebraic property of the wedge product is as follows. If $\omega^{(k)}$ is a $k$-form and $\omega^{(m)}$ is an $m$-form, then we have

$$\omega^{(k)} \wedge \omega^{(m)} = (-1)^{mk} \omega^{(m)} \wedge \omega^{(k)}. \tag{A6}$$

## 2. Differential $m$-forms and integration

To define a differential $m$-form, we replace the real numbers $a_I$ in Eq. A3 by differentiable functions in $\mathbb{R}^n$, $f_I : \mathbb{R}^n \to \mathbb{R}$. Thus, the differential $m$-form takes $m$-vector fields on $\mathbb{R}^n$ as input and outputs a function $g : \mathbb{R}^n \to \mathbb{R}$.

We now define an integral of a differential $m$-form over a surface $S$ in $\mathbb{R}^n$. We first consider an example of the integral of 2-form over a two-dimensional surface in $\mathbb{R}^3$ parametrized using

two variables $u_1$ and $u_2$ as $(x_1, x_2) = (\phi_1(u_1, u_2), \phi_2(u_1, u_2)) \equiv \vec{\phi}(u_1, u_2) \subset \mathbb{R}^3$. Then the integral of the 2-form, $\omega^{(2)}$ over $S$ is given as follows,

$$\int_S \omega^{(2)} = \int \int \omega^{(2)}_{\vec{\phi}_{(u_1,u_2)}} \left( \frac{\partial \vec{\phi}}{\partial u_1}, \frac{\partial \vec{\phi}}{\partial u_2} \right) du_1 du_2 \qquad (A7)$$

where the subscript $\vec{\phi}_{(u_1,u_2)}$ indicates the coordinates of the parametrized point. The 2-form $\omega^{(2)}_{\vec{\phi}_{(u_1,u_2)}}$ takes a pair of vectors in $T_{\vec{\phi}_{(u_1,u_2)}}\mathbb{R}^3$ *i.e.* $\frac{\partial \vec{\phi}}{\partial u_i}$ for $i = 1, 2$ as inputs. Here, the notation $\frac{\partial \vec{\phi}}{\partial u_i}$ is used for a vector $\langle \frac{\partial \phi_1(u_1,u_2)}{\partial u_i}, \frac{\partial \phi_2(u_1,u_2)}{\partial u_i} \rangle \in T_{\phi_{(u_1,u_2)}}\mathbb{R}^3$. The limits of integration for variables $u_1$ and $u_2$ which parametrize the surface $S$ embedded in $\mathbb{R}^3$ are chosen according to the definition of the surface.

The generalization to $m$-forms is straightforward *i.e.* the integral of an $m$-form $\omega^{(m)}$ over a $m$-dimensional hypersurface in $\mathbb{R}^n$ is given by

$$\int_{S \in \mathbb{R}^n} \omega^{(m)} = \int \int ... \int \omega^{(m)}_{\vec{\phi}_{(u_1,u_2,...,u_m)}} \left( \frac{\partial \vec{\phi}}{\partial u_1}, \frac{\partial \vec{\phi}}{\partial u_2}, ....., \frac{\partial \vec{\phi}}{\partial u_m} \right) du_1 du_2 ... du_m . \qquad (A8)$$

### 3. Exterior derivative

We now define the exterior derivative which is a map from the differential $m$-forms on $\mathbb{R}^n$ to the differential $m+1$-forms on $\mathbb{R}^n$. For example, if $\omega$ is a 0-form in $\mathbb{R}^n$ *i.e.* a function $\omega \equiv f(x_1, x_2, ..., x_n)$, then $d\omega \equiv df$ is a 1-form. In general, we act $d$ on Eq. A4 as follows,

$$d\omega^{(m)} = \sum_I \sum_{j=1}^n \frac{\partial f_I}{\partial x_j} dx_j \wedge dx_I \qquad (A9)$$

which makes it clear how we go from a differential $m$-form to a differential $m+1$-form. The exterior derivative obeys a product rule as follows

$$d(\omega^{(m)} \wedge \omega^{(k)}) = d\omega^{(k)} \wedge \omega^{(m)} + (-1)^m \omega^{(m)} \wedge d\omega^{(k)} \qquad (A10)$$

whose derivation is simple and uses the relation in Eq. A6. Another important property of the exterior derivative $d$ is that $d^2\omega^{(m)} = 0$.

### 4. Hodge operator

We consider the $dx_I$ used in the definition of $m$-form in Eq. A4 as a basis of a vector space of $m$-forms in $\mathbb{R}^n$ which we denote as $V^m(\mathbb{R}^n)$. The Hodge operator is used to define

the Hodge duality which is a relation between the vector spaces of $m$-forms and $n - m$ forms *i.e.* $V^m(\mathbb{R}^n)$ and $V^{n-m}(\mathbb{R}^n)$. For our purposes, it is sufficient to define the Hodge star $\star$ in $\mathbb{R}^n$ using coordinates $x_1, x_2, ..x_n$ via the equation

$$\star \, dx_I = dx_{I\star}, \tag{A11}$$

such that $dx_I \wedge dx_{I\star} = dx_1 \wedge dx_2 \wedge ... \wedge dx_n$ where the order is important as we know from the definition of the wedge product. In $\mathbb{R}^3$, for example, $\star dx_1 = dx_2 \wedge dx_3$. In general, the Hodge star maps an $m$-form to an $(n - m)$-form on $\mathbb{R}^n$.

Using the property of the exterior derivative $d^2 = 0$ and the definition of the Hodge operator, conventional vector calculus results in $\mathbb{R}^3$ are recovered. Under the identification of the functions $f$ to 0-forms $f$ and of vector fields $\vec{F} = f_1\hat{x}_1 + f_2\hat{x}_2 + f_3\hat{x}_3$ to 1-forms $\omega = f_1 dx_1 + f_2 dx_2 + f_3 dx_3$, one notes that $\text{grad}f \equiv df$, $\text{curl}(\vec{F}) \equiv (\star d)\omega$ and $\text{div}(\vec{F}) \equiv (\star d \star)\omega$. Since $d^2 = 0$, we get $\star d^2 f = 0$ and $(\star d \star)(\star d)\omega = \star d^2 \omega = 0$ which correspond to the identities $\text{curl}(\text{grad}(f)) = 0$ and $\text{div}(\text{curl}(F)) = 0$. Hence, the exterior derivative and differential $m$-forms generalize the vector calculus results of $\mathbb{R}^3$ to $\mathbb{R}^n$.

## 5. Generalized Stokes' theorem

The generalized Stokes' theorem states that the integral of a differential $m$-form, $\omega^{(m)}$ over the boundary of some orientable manifold $\mathbb{M}$, $\partial\mathbb{M}$ is equal to the integral of its exterior derivative over the manifold $\mathbb{M}$. This is written as follows,

$$\int_{\partial\mathbb{M}} \omega^{(m)} = \int_{\mathbb{M}} d\omega^{(m)}. \tag{A12}$$

We now consider the integral of the $m$-form $\omega$ over the boundary of a smooth $m + 1$-chain $\mathbb{C}_{m+1}$ in the smooth manifold $\mathbb{M}$. This gives the Stokes' theorem for chains as follows,

$$\int_{\partial\mathbb{C}_{m+1}} \omega^{(m)} = \int_{\mathbb{C}_{m+1}} d\omega^{(m)}. \tag{A13}$$

## Appendix B: Homology and toric codes in $n + 1$d

### 1. Review of homology

Homology groups are topological invariants because they are isomorphic for spaces that have the same homotopy type. For example, in two dimensions, the homology group $H_1(X_1)$

where $X_1$ is an annular disk is an abelian group isomorphic to $\mathbb{Z}$ corresponding to $\pm Nb$ where $N$ is an integer and $b$ is the "boundary" of the hole. We consider spaces $\mathbb{X}$ that are triangulable *i.e.* homeomorphic to a polyhedron $\mathbb{K}$ and define simplicial homology groups associated with the oriented simplices of $\mathbb{K}$. An oriented $K$-simplex is denoted in terms of 0-simplices $[v_i]$ by $\sigma^{(K)} = [v_0, ..., v_K]$ such that its boundary is given by

$$\partial\sigma^{(K)} = \sum_{i=1}^{K}(-1)^j[v_0, ..., \hat{v}_j, ..., v_K] \tag{B1}$$

where $[v_0, ..., \hat{v}_j, ..., v_K]$ is the $(K\text{-}1)$ simplex obtained by omitting $\hat{v}_j$. Note that $\partial[v_i] = 0$. Even or odd permutation with respect to a fixed base ordering of vertices determines the sign or orientation of a $K$-simplex.

For each $p$-simplex $\sigma_i^{(p)}$ of an $n$-dimensional simplicial complex $K$ (a "set of simplices"), there is an abelian group $C_p(K)$ called the chain group generated by the oriented $p$-simplices of $K$. Here, $p$ can run from 1 to $n$. Elements of $C_p(K)$ are given by

$$c_p = \sum_{i=1}^{l_p} f_i\sigma_i^{(p)}, \quad f_i \in \mathbb{Z} \tag{B2}$$

and the boundary operator Eq. (B1) is a homomorphism from $C_p(K)$ to $C_{p-1}(K)$ *i.e.* $\partial_p : C_p(K) \to C_{p-1}(K)$. Note that the coefficients $f_i$ are integers. The kernel of this homomorphism Eq. (B1) is the $p$-dimensional cycle group of $C_p(K)$ and denoted as $Z_p(K)$. The homeomorphic image $\partial_{p+1}C_{p+1}(K)$ is a subgroup of $C_p(K)$. This is the $p$-dimensional boundary group of $K$ denoted as $B_p(K)$. Because $\partial b_p = 0$ for any $b_p \in B_p(K)$, $B_p(K)$ is also a subgroup of $Z_p(K)$.

We define the $p$-dimensional homology group of $K$ denoted by $H_p(K)$ to be the quotient group of cycles and boundaries *i.e.*

$$H_p(K) = Z_p(K)/B_p(K) \tag{B3}$$

If $K$ is contractible, then

$$H_p(K) = \begin{cases} 0, & p \neq 0 \\ \mathbb{Z}, & p = 0 \end{cases} \tag{B4}$$

For two different triangulations of the same topological space, $Z_p(K)$ and $B_p(K)$ can be different but $H_p(K)$ which depends on the number of $p+1$-dimensional holes is the same.

The homology groups are hard to calculate; hence we define relative homology groups. By considering some subpolyhedron $\{K\} \subset K$, we relate homology of $H_p(K)$ to $H_p(\{K\})$.

The relative $p$-chain group or the $p$-dimensional chain group of $K$ modulo $\{K\}$ with integer coefficients is the quotient group

$$C_p(K; \{K\}) = C_p(K) | C_p(\{K\}), \quad p > 0 \tag{B5}$$

such that $c_p + C_p(\{K\}) \in C_p(K; \{K\})$ where $c_p \in C_p(K)$. Using this we defined the relative boundary map $\partial_p$ as a homomorphism from $C_p(K; \{K\})$ to $C_{p-1}(K; \{K\})$ which acts as

$$\partial(c_p + C_p(\{K\})) = \partial_p c_p + C_{p-1}(\{K\}) \tag{B6}$$

We define the relative cycle group and the relative boundary group analogously which leads to the following definition of the relative homology group.

The relative $p$-dimensional homology group of $K$ denoted by $H_p(K; \{K\})$ is the quotient group of relative cycles and relative boundaries *i.e.*

$$H_p(K; \{K\}) = Z_p(K; \{K\}) / B_p(K; \{K\}) \tag{B7}$$

The intuition is that in the relative homology group $H_p(K; \{K\})$, anything in $\{K\}$ is identity.

We defined the elements of the chain group to be linear combinations of oriented $p$-simplices with integer coefficients. Instead of integer coefficients, we could take the coefficients $g \in G$ where $G$ is some abelian group. This defines the chain groups of the complex $K$ over the abelian group $G$, denoted as $C_{p,G}(K)$.

## 2. Homology description of toric codes in $n+1$d and 4+1d

We define the $(i, n-i)$ $(n+1)$-dimensional $\mathbb{Z}_N$ toric code on the hypercubic lattice via the chain complex

$$C_{i+1} \overset{\partial_{i+1}}{\to} C_i \overset{\partial_i}{\to} C_{i-1}. \tag{B8}$$

Here $C_i$ refers to an $i$-chain group and each element of $C_i$, *i.e.* an $i$-chain is a formal linear combination of the $i$-cells in a cell complex. $\partial_i$ refers to the boundary map that acts on elements of $C_i$ and the composition $\partial_i \cdot \partial_{i+1}$ is 0. For $\mathbb{Z}_N$ toric codes, the chain groups $C_i \equiv C_i(\mathcal{L}, \mathbb{Z}_N)$ are defined for the lattice $\mathcal{L}$ with coefficients in $\mathbb{Z}_N$. We associate a $\triangle$-complex structure with the lattice $\mathcal{L}$, *i.e.*, $\mathcal{L}$ can be defined via a triangulation of the

manifold $\mathcal{M}^n$ in $n$ dimensions with a branching structure which follows from a total ordering of the vertices of the $n$-simplex. This implies that each element of the chain group $C_i$ is formally a linear combination $\sum_\alpha z_\alpha e_\alpha^{(i)}$ where $z_\alpha \in \mathbb{Z}_N$ and $e_\alpha^{(i)}$ is an $i$-simplex, a subsimplex of an $n$-simplex for $i < n$. Given an ordering of the vertices on the $n$-simplices, we define the action of the boundary map on an $i$-simplex $[v_0, ..., v_i]$ as

$$\partial_i [v_0, ..., v_i] = \sum_j (-1)^j [v_0, ....\hat{v}_j, ....., v_i]$$

where $v_0, ..., v_i$ represents the vertices of the $i$-simplex such that their order is consistent with the ordering of the vertices of the $n$-simplex. The $\hat{v}_j$ refers to the vertex that has been removed from the vertices of the $i$-simplex to give one face of the $i$-simplex $i.e.$ the $i - 1$ simplex on its boundary. Note that the vertices of the $i < n$ simplices are a subset of the vertices of an $n$-simplex and ordered according to the ordering of vertices in the $n$-simplex. The definition of the boundary map as defined above leads to the desired property for composition of boundary maps $i.e.$ $\partial_i \cdot \partial_{i+1} = 0$. We define the $(i, n - i)$ $\mathbb{Z}_N$ toric codes on a triangulation of the torus in $n$-dimensions via the chain complex above, such that qubits are placed on the $i$-simplex, $X$-type stabilizers are associated with the $(i - 1)$-simplices and the $Z$-type stabilizers with the $(i + 1)$-simplices.

Since we consider the hypercubic lattice, the $i$-cells are $i$-dimensional hypercube cells. Thus, the $(2, 2)$ 4+1d $\mathbb{Z}_N$ toric code $i.e.$ the membrane only toric code can be defined on the hypercubic lattice [36] via the chain complex

$$C_3 \xrightarrow{\partial_3} C_2 \xrightarrow{\partial_2} C_1. \tag{B9}$$

such that the qubits are on the 2-cells, the $Z$-type stabilizers associated with the faces $i.e.$ 2-cells are described by the boundary map on the 3-cells, $\partial_3$ while the edge $X$-type stabilizers are described by the transpose[13] of the boundary map $\partial_2$ $i.e.$ $\partial_2^T : C_1 \to C_2$ which acts on the 1-cells. The 4d lattice and the stabilizer generators are shown explicitly in Fig. 1.

The ground state degeneracy can be seen via the (co)homology groups associated with the chain complex described above. For example, the $X$-type membrane logical operators on a 4-torus $\mathcal{T}^4$ are described by the homology group $H_2(\mathcal{T}^4)$ while the $Z$-type membrane logical operators are described on the dual lattice by the same homology group. The number

---

[13] One could introduce the cochain complex to describe the edge stabilizers as the stabilizers associated with the 3-cell on the dual lattice but that is not necessary.

of encoded qubits which is also equal to the number of independent logical operator pairs, is hence given by the dimension of the homology group *i.e.* $\dim H_2(\mathcal{T}^4)$ which is equal to 6. Hence, the topological ground state degeneracy for the $\mathbb{Z}_N$ toric code is $N^6$.

Instead of the $(2,2)$ 4+1d $\mathbb{Z}_N$ toric code which is on a torus, we also consider the $(2,2)$ 4+1d $\mathbb{Z}_N$ surface code with gapped boundaries. We consider gapped boundaries of $e$ and $m$ type meaning that they can condense the loop excitations of $e$ and $m$ types respectively. The logical operators in the presence of the gapped boundaries [22] are associated with the relative homology groups $H_i(\mathcal{L}, B_a)$, where $\mathcal{L}$ denotes the 4+1d lattice and $B_a$ stands for a subcomplex corresponding to the gapped boundary of type $a$. To calculate the relative homology group, we use the fact that $H_i(\mathcal{L}, B_a) \cong H_i(\mathcal{L}/B_a)$ (for $i > 0$) where $\mathcal{L}/B_a$ stands for the quotient space such that the boundaries of type $a$ are identified. In 4+1d surface code, the logical-$Z$ membrane operators are associated with the non-trivial relative 2-cycle in the 2nd relative homology group, $H_2(\mathcal{L}, B_e)$, where $B_e$ stands for the $e$-boundaries. The logical-$X$ membrane operators are associated with the non-trivial relative 2-cycles in the 2nd relative homology group on the dual lattice $\mathcal{L}^*$, $H_2(\mathcal{L}^*, B_m^*)$, where $B_m^*$ stands for the dual subcomplex corresponding to the $m$-boundaries. The number of encoded qubits or the topological ground state degeneracy is given by the dimension of these relative homology groups. Consider the surface code with $e$-type boundaries at the $yzw$ and $xzw$ 3+1d boundaries and $m$-type boundaries at the $xyw$ and $xyz$ 3+1d boundaries. The relative homology calculation gives the number of encoded qubits for such a configuration to be 1. This is also obvious from the fact that the logical operators are $Z$ and $X$ membrane operators in the $xy$ and $zw$ planes respectively. The patches of $e$ and $m$ boundaries come with the domain walls between such boundaries which we refer to the $e-m$ domain walls. Since this is a 4+1d model, one could consider more nontrivial topologies of $e-m$ domain walls and calculate the number of encoded qubits via the relative homology groups accordingly. For example, one could consider the two-dimensional $e-m$ domain walls to be on a torus. In the presence of $n$ toric $e-m$ domain walls, we get the number of encoded qubits to be $n-2$. This is clear because there are $X$ membrane operators that are periodic along $z$ or $w$ and incident two of the $m$ boundaries. We call these $X$ membrane operators as $mz$ and $mw$. Similarly there are $ez$ and $ew$ $Z$ membrane operators that anticommute with $mw$ and $mz$ $X$ membrane operators respectively. Thus, the number of independent membrane operators that joins all pairs of the $m$ boundaries, which are $n/2$ in number, gives the number of encoded qubits.

There are $2(n/2 - 1) = n - 2$ such membrane operators.

## Appendix C: Review of higher cup products

We now briefly discuss the cup products on triangulated manifolds and lattices and also on hypercubic lattices. We refer the reader to Ref. [36] for more details.

### 1. Cup products on triangulations

We consider an $m$-cochain $c^{(m)}$ and an $n$-cochain $c^{(n)}$ with $\mathbb{Z}_N$ coefficients on a smooth triangulated manifold. The cup product $\cup_0$ of these two cochains is an $(m+n)$-cochain and is defined according to the action on an $(m+n)$-chain denoted as $(0, 1, ..., m+n)$ as follows,

$$c^{(m)} \cup_0 c^{(n)}(0, 1, ..., m+n) = c^{(m)}(0, 1, ..., m)c^{(n)}(m, m+1, ..., m+n) \qquad \text{(C1)}$$

Note that 0 in $\cup_0$ indicates the order of the cup product. We now define higher cup products $\cup_q$ where $q$ can be nonzero. The $\cup_q$ cup product of the cochains $c^{(m)}$ and $c^{(n)}$ is a $m+n-q$ cochain can be written as,

$$
\begin{aligned}
&c^{(m)} \cup_q c^{(n)}(0, 1, ..., m+n) \\
&= \sum_{0 \leq i_0 < ... i_q < m+n-q} (-1)^p c^{(m)}(0 \to i_0, i_1 \to i_2, ...)c^{(n)}(i_0 \to i_1, i_2 \to i_3, ...),
\end{aligned}
$$

where we used the notation of Ref. [36]. The numbers in the brackets showing the chains on the right are in ascending order and $i_1 \to i_2$ are integers from $i_1$ to $i_2$. Here, in RHS, $c^{(m)}$ acts on an $m$-chain and $c^{(n)}$ acts on an $n$-chain, meaning that the chain $(0 \to i_0, i_1 \to i_2, ...)$ has $m + 1$ numbers to specify the $m$-cell and the chain $(i_0 \to i_1, i_2 \to i_3, ...)$ has $n + 1$ numbers to specify the $n$-cell. The $p$ in $(-1)^p$ is the number of elementary permutations needed to go from the sequence $(0 \to i_0, i_1 \to i_2, ..., i_0 + 1 \to i_1 - 1, i_2 + 1 \to i_3 - 1, ...)$ to $(0 \to m+n-q)$. For example, for two 2-cochains $c_1^{(2)}$ and $c_2^{(2)}$, we have $c_1^{(2)} \cup_1 c_2^{(2)}(0123) = -c_1^{(2)}(013)c_2^{(2)}(123) + c_1^{(2)}(023)c_2^{(2)}(012)$ where $(0123)$ indicates a 3-cell and $(013)$, $(123)$, $(023)$ and $(012)$ indicate the 2-cells.

## 2. Cup products on hypercubic lattices

Now we consider the generalization of the definition of the higher cup product on triangulated manifolds to hypercubic lattices. We write the definition used in Ref. [36] which the reader can refer for more details.

For the hypercubic lattice in $\mathbb{R}^d$, we denote the $m$-cells via $d$ number of coordinate indices such that $d - m$ out of those $d$ can takes values in $+$ and $-$ and the remaining are fixed and denoted as $\bullet$. This is more clear from the example of a square 2-cell in $\mathbb{R}^2$ which we denote as $(\bullet, \bullet)$ and which we imagine to be the $x$ and $y$ coordinates of the origin at the center of the 2-cell. Keeping the origin in mind, we assign coordinates to the 1-cells and 0-cells at the boundary of this 2-cell. If we move in $-x$ direction to the vertical edge, we assign that edge/1-cell the coordinates $(-, \bullet)$ and if we move right, we assign $(+, \bullet)$. Similarly the top and bottom edges are $(\bullet, +)$ and $(\bullet, -)$. In Fig. 7, we show this notation for a 4-cell of a 4D hypercubic lattice.

Using this notation for the $m$-cells of the hypercubic lattice in $d$ dimensions and the corresponding $\mathbb{Z}_N$ chains defined on this lattice, we define the higher cup products. We again consider an $m$-cochain $c^{(m)}$ and an $n$-cochain $c^{(n)}$. The $\cup_q$ cup product of $c^{(m)}$ and $c^{(n)}$ is an $t = (m + n - q)$ cochain defined as follows via its action on an $t$-cell denoted with $t$ bullets out of the $d$ coordinates and the remaining can have $\pm$ coordinates. Since the $\pm$ coordinates will be fixed and same in the decomposition on the RHS, we don't keep them in the cell notations below

$$c^{(m)} \cup_q c^{(n)}(\bullet, \bullet, .., \bullet) = \sum_{(r_1, r_2, .., r_t), (s_1, s_2, ... s_t)} (-1)^p c^{(m)}(r_1, r_2, .., r_t) c^{(n)}(s_1, s_2, .., s_t) \qquad \text{(C2)}$$

where the sum is over $m$-cells denoted $\vec{r} = (r_1, r_2, .., r_t)$ and $n$-cells denoted $\vec{s} = (s_1, s_2, .., s_t)$ chosen from the set defined via

$$\left\{ (r_1, r_2, .., r_t), (s_1, s_2, .., s_t) \middle| r_{i_1} = s_{i_1} = ... = r_{i_q} = s_{i_q} = \bullet \right.$$
$$\left. \text{and } (r_j, s_j) \in \{((-1)^{n(j)}+, \bullet), (\bullet, (-1)^{n(j)}-))\} \right\}$$

i.e. $q = m + n - t$ coordinates are fixed to $\bullet$ in both cells and the remaining are fixed according to integer $n(j)$ for coordinate $j$. The integer $n(j)$ is defined according to the position of the coordinate relative to the bullet positions given $i_1$ to $i_q$ via $i_{n(j)} < j < i_{n(j)} + 1$. The $p$ in $(-1)^p$ is the sum of total number of $+$ signs in $\vec{r}$ and $\vec{s}$ and the number of elementary

permutations needed to go from the sequence $(1, 2, ...t)$ to $(i_1 \to i_q, b_1 \to b_{t-m}, a_1 \to a_{t-n})$. Here, $i_1 \to i_q$ denotes the coordinates at which $\vec{r}$ and $\vec{s}$ are $\bullet$, $b_1 \to b_{t-m}$ and $a_1 \to a_{t-n}$ denote the coordinates in $\vec{s}$ and $\vec{r}$, respectively, that are not $\bullet$.

### Appendix D: Topological domain wall in $\mathbb{Z}_2 \times \mathbb{Z}_2$ gauge theory in 2+1d

In this appendix, we give a lattice construction for the topological domain wall in $\mathbb{Z}_2 \times \mathbb{Z}_2$ gauge theory decorated with the gauged SPT phase with $\mathbb{Z}_2 \times \mathbb{Z}_2$ symmetry in 1+1d, as described by $H^2(\mathbb{Z}_2 \times \mathbb{Z}_2, U(1)) = \mathbb{Z}_2$. We introduce two qubits, labelled by $(i)$ with $i = 1, 2$ on each edge, acted by Pauli matrices $X^{(i)}, Y^{(i)}, Z^{(i)}$, and we denote the $\mathbb{Z}_2$ gauge field $a^{(i)} = (1 - Z^{(i)})/2$.

The Hamiltonian is two copies of toric code

$$H = -H^{(1)} - H^{(2)}, \quad H^{(i)} = -\sum_v \prod X_e^{(i)} - \sum_f \prod Z_e^{(i)} . \tag{D1}$$

We call the term summing over faces $f$ the flux terms, and the other terms the vertex terms. The low energy theory is two copies of $\mathbb{Z}_2$ gauge theory. We moreover decorate the vertex term with the projector to zero flux sector $\prod Z_e = 1$ for each face, without changing the low energy physics.

The unitary describing the topological domain wall is given by the non-trivial group cocycle in $H^2(\mathbb{Z}_2 \times \mathbb{Z}_2, U(1)) = \mathbb{Z}_2[(-1)^{a^{(1)} \cup a^{(2)}}]$:

$$U = (-1)^{\int a^{(1)} \cup a^{(2)}} = \prod_{e,e'} \left( CZ_{e,e'}^{(1),(2)} \right)^{\int \tilde{e} \cup \tilde{e}'} , \tag{D2}$$

where the integral is over the 2d space, and $CZ_{e,e'}^{(1),(2)}$ is the control-Z operator for the two consecutive edges $e, e'$ between the qubits (1) and (2). The unitary is described in Figure 8 on square lattice. The operator $U$ commutes with the Hamiltonian with the zero flux projector: it commutes with the flux term, and it commutes with the Hamiltonian in the zero flux sector.

We examine how the operator acts on the excitations by conjugation. It acts trivially on the electric excitations created by $Z^{(1)}, Z^{(2)}$; however, it changes the types of the magnetic excitations created by $X^{(1)}, X^{(2)}$ , since

$$U^{-1} X_e^{(1)} U = X_e^{(1)} (-1)^{\int \tilde{e} \cup a^{(2)}} , \tag{D3}$$

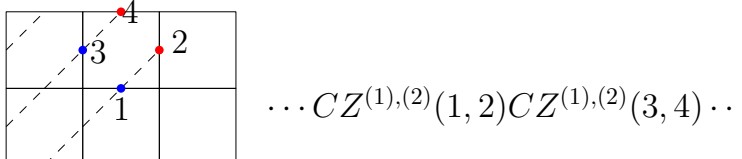

FIG. 8. The unitary that implements the $\mathbb{Z}_2$ 0-form symmetry described by $H^2(\mathbb{Z}_2 \times \mathbb{Z}_2, U(1))$. $CZ^{(1),(2)}(e_1, e_2)$ is the control-Z operator for the first qubit on edge $e_1$ and second qubit on edge $e_2$.

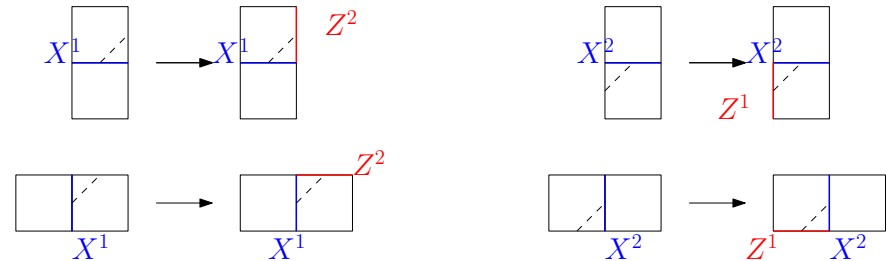

FIG. 9. The action of the electric operator on $x, y$ plane by conjugation.

where $\tilde{e}$ is the one-cochain that takes value 1 on the edge $e$ and zero on other edges, and we use the fact that conjugation by $X_e^{(1)}$ generates the shift $a^{(1)} \to a^{(1)} + \tilde{e}$. See Figure 9 for an illustration of the transformation. Equivalently, conjugating $X^{(1)}$ by the control-Z operator generates $Z^{(2)}$, and similarly conjugating $X^{(2)}$ by the control-Z operator generates $X^{(1)}$. Thus the unitary $U$ implements the anyon permutation

$$(e^{(1)}, m^{(1)}, e^{(2)}, m^{(2)}) \to (e^{(1)}, m^{(1)}e^{(2)}, e^{(2)}, m^{(2)}e^{(1)}) . \tag{D4}$$

## Appendix E: Review of Walker-Wang models

The Walker-Wang models [47] are a class of exactly solvable models for 3+1d topological orders. The basic intuition behind the Walker-Wang construction is simple. Given a 2+1d anyon theory, the model is constructed such that the ground state wave function is a superposition of "3D string-nets" labeled by the anyon types, which describe the 2+1d space time trajectories of the anyons. The coefficient in front of each configuration is equal to the topological amplitude of the corresponding anyon process. It can be evaluated by using the graphical rules depicting the algebraic data of the anyon theory, captured essentially by the $F$ and $R$ symbols defined in Fig. 11, which specify the fusion and braiding rules of the anyons, respectively. The bulk-boundary correspondence described above is similar in

spirit to the correspondence between quantum Hall wave functions and the edge conformal field theories. There, the systems are in one dimension lower, and the bulk wave function is expressed as a correlator in the boundary CFT [48].

Mathematically, the input anyon theory of a Walker-Wang model is described by a braided fusion category $\mathcal{A}$. If $\mathcal{A}$ is modular, which means that the only quasiparticle that braids trivially with itself and all other quasiparticles in $\mathcal{A}$ is the vacuum[14], the output theory would have a trivial bulk and a surface with topological order described by $\mathcal{A}$, and the model belongs to the first type of Walker-Wang models we introduced in the previous section. On the other hand, if $\mathcal{A}$ is non-modular, the output theory would have a nontrivial bulk, and the model belongs to the second type of Walker-Wang models. The surface theory in this case is more complicated because it contains not only the quasiparticles in $\mathcal{A}$, but also the endpoints of bulk loop excitations that are cut open by the system boundary.

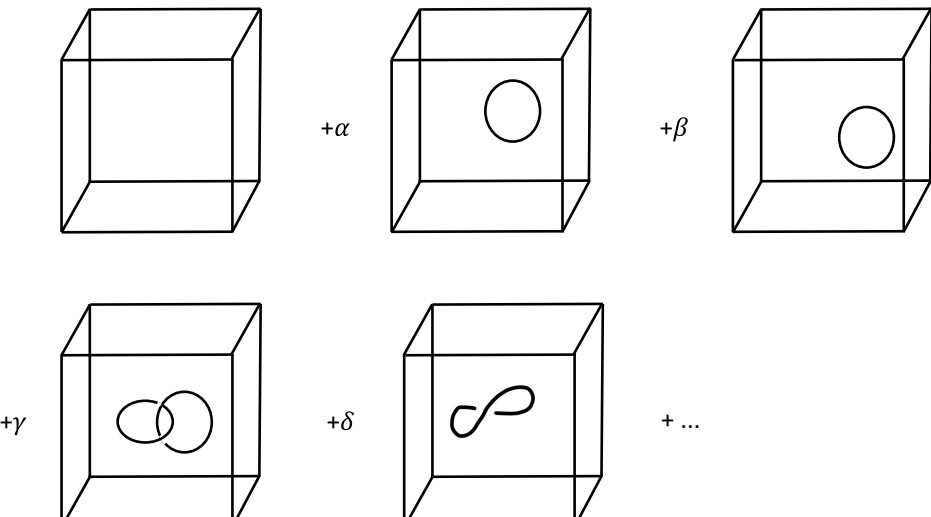

FIG. 10. An example of the ground state wave function of a Walker-Wang model. $a$ and $b$ here label the quasiparticle types.

---

[14] Premodular theories have transparent particles which can be condensed to get a modular theory.

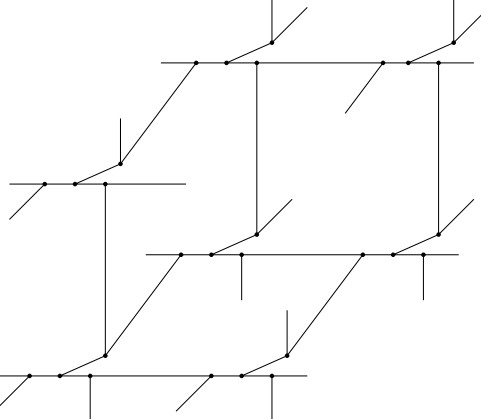

FIG. 12. Planar projection of a trivalent resolution of the cubic lattice.

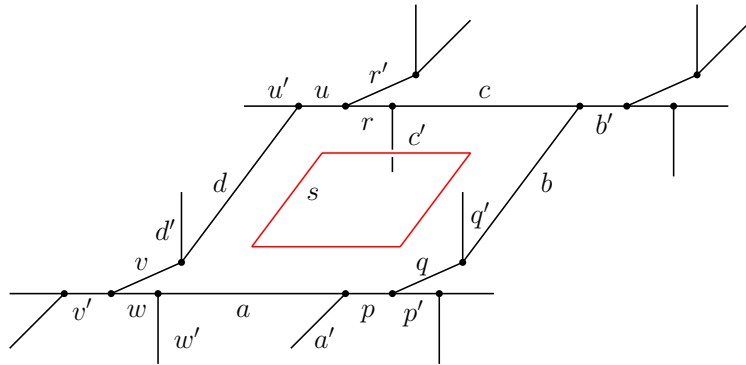

FIG. 13. Plaquette term in the Walker-Wang Hamiltonian.

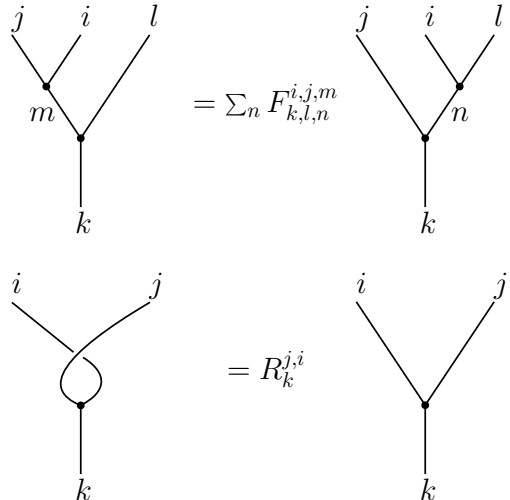

FIG. 11. Graphical definition of $F$ and $R$ symbols.

To be more concrete, we illustrate with three examples. First, we consider the simplest nontrivial input $\mathcal{A}$ possible, which consists of only the vacuum $I$ and a boson $e$. $\mathcal{A}$ is non-modular because $e$ is distinct from the vacuum but braids trivially with everything in $\mathcal{A}$.

A Walker-Wang model with such input describes the 3+1d $\mathbb{Z}_2$ gauge theory with $e$ being the $\mathbb{Z}_2$ gauge charge, which is deconfined in the bulk and on the boundary [49]. Next, we modify $\mathcal{A}$ a bit by replacing the boson $e$ with a semion $s$. $\mathcal{A}$ becomes modular in this case, because $s$ braids nontrivially with itself. A Walker-Wang model with the modified input has a trivial bulk and a deconfined semion excitation $s$ on the boundary [49]. One could also consider the input category consisting of vacuum and a fermion $f$. In that case, one obtains the 3D toric code with a fermionic charge.

In general, deconfined bulk quasiparticle excitations of a Walker-Wang model correspond to quasiparticles in the symmetric center $\mathcal{Z}(\mathcal{A})$ of the input braided fusion category $\mathcal{A}$. A quasiparticle belongs to $\mathcal{Z}(\mathcal{A})$ if it has trivial braiding with itself and all other quasiparticles in $\mathcal{A}$. If $\mathcal{A}$ is modular, $\mathcal{Z}(\mathcal{A})$ is trivial, which is consistent with the fact that a Walker-Wang model with modular input has a trivial bulk. If $\mathcal{A}$ is non-modular, it is known that there are two possibilities for $\mathcal{Z}(\mathcal{A})$ [50]: (1) $\mathcal{Z}(\mathcal{A})$ contains only bosons. In this case, it can be identified with the set of irreducible representations of some finite group $G$; (2) $\mathcal{Z}(\mathcal{A})$ contains at least one fermion. In this case, it can also be identified with the set of irreducible representations of some finite group $G$, but each representation comes with a parity, and the set is split into even and odd sectors, corresponding to the bosons and fermions in $\mathcal{Z}(\mathcal{A})$, respectively. Thus, the deconfined bulk quasiparticle excitations of a Walker-Wang model with non-modular input correspond to the irreducible representations of some finite group, and it is plausible that the bulk topological order of the model is a gauge theory of the corresponding group.

We now review some details of Walker-Wang models. We closely follow Ref. [47] and refer the reader there for further details. Walker-Wang models are defined on a fixed planar projection of a trivalent resolution of the cublic lattice as shown in Fig. 12. The Hilbert space of a model defined on the lattice is spanned by all labelings of the edges by the input anyon types. The Hamiltonian is of the form

$$H = -\sum_v A_v - \sum_p B_p, \tag{E1}$$

where $A_v$ is a vertex term which enforces the fusion rules at $v$ by giving an energy penalty to string configurations that violate the fusion rules at $v$, and $B_p$ is a plaquette term of the form $B_p = \sum_s d_s B_p^s$, where the summation is over all the input anyon types $s$, weighted by the quantum dimension of $s$. Each $B_p^s$ acts on the anyon labels of the edges

around plaquette $p$, in a way determined by the anyon labels of the edges adjoining $p$. More explicitly, the matrix element of $B_p^s$ sandwiched between states with plaquette edges $(a'', b'', c'', d'', p'', q'', r'', u'', v'', w'')$ and $(a, b, c, d, p, q, r, u, v, w)$ is given by

$$
\begin{aligned}
(B_p^s)_{(a,b,c,d,p,q,r,u,v,w)}^{(a'',b'',c'',d'',p'',q'',r'',u'',v'',w'')} &= R_q^{bq'} (R_c^{rc'})^* (R_{q''}^{b''q'})^* \times \\
&R_{c''}^{r''c'} F_{a'pa}^{sa''p''} F_{p'qp}^{sp''q''} F_{q'bq}^{sq''b''} F_{b'cb}^{sb''c''} F_{c'rc}^{sc''r''} F_{r'us}^{sr''u''} F_{u'du}^{su''d''} \times \\
&F_{d'vd}^{sd''v''} F_{v'wv}^{sv''w''} F_{w'aw}^{sw''a''},
\end{aligned}
$$

(E2)

The above expression looks rather complicated, but there is a simple graphical way of understanding the action of $B_p^s$. Namely, $B_p^s$ temporarily displaces certain links ($c'$ and $q'$ in Fig. 13) and fuses a loop with anyon label $s$ to the skeleton of $p$. One can check that all terms in the Hamiltonian commute, and the model is exactly solvable.

To be able to discuss point and loop excitations in Walker-Wang models, we also need to define string operators and membrane operators in these models. The string operators have a graphical definition analogous to that of the plaquette operators. Namely, to create a pair of quasiparticle excitations $\alpha \in \mathcal{A}$ at two points, we just need to lay an $\alpha$-string connecting the two points, and then fuse it to the edges of the lattice. Furthermore, one can show that the string operator commutes (resp. fails to commute) with the plaquette operators threaded by the string if $\alpha \in \mathcal{Z}(\mathcal{A})$ (resp. $\alpha \notin \mathcal{Z}(\mathcal{A})$), and the corresponding quasiparticles are deconfined (resp. confined) in the bulk. On the other hand, all quasiparticles in $\mathcal{A}$ are deconfined on the boundary, because string operators restricted to the boundary do not thread any plaquettes and hence there is no energy penalty associated with them. Unlike the string operators, in general, we do not know how to implement membrane operators in Walker-Wang models, but as we show below, we deduce the statistics of the loop excitations without explicitly writing down the membrane operators.

### 1. Semion Walker-Wang model

We now specify the semion Walker-Wang model which was briefly mentioned above in discussion of the nontrivial input braided fusion categories $\mathcal{A}$. The Semion Walker Wang is obtained when the input category $\mathcal{A}$ consists of vacuum $I$ and semion $s$. The Hamiltonian of the semion Walker-Wang model [49] on the trivalent lattice described above, is given as

follows,

$$H = -\sum_v \underbrace{\prod_{e(v)} \sigma_i^Z}_{A_v} + \sum_p \underbrace{(\prod_{i \in \partial p} \sigma_i^X)(\prod_{j \in e(p)} i^{n_j}) i^{\sum_{j,\text{red}} n_j - \sum_{j,\text{blue}} n_j}}_{B_p} \, .$$

where $A_v$ ($B_p$) refer to the terms associated with vertices $v$ (plaquettes $p$). $s(v)$ refers to the three edges emitting out from a vertex $v$ and $e(p)$ is the set of ten edges emitting out from the plaquette $p$. $\partial p$ is the set of ten edges on the boundary of the plaquette $p$. Note that $e(p)$ does not include any edge in $\partial p$. $\sigma_j^Z$ and $\sigma_j^X$ are the Pauli operators that act on the spin-1/2 degree of freedom at edge $j$ and $n_j = \frac{1}{2}(1 - \sigma_j^Z)$.

Consider the states $|\Psi\rangle$ that are eigenstates of the plaquette terms $i.e.$ $B_v|\Psi\rangle = |\Psi\rangle$. Such states are referred to as the loop gas states and each term in the Hamiltonian commutes with every other term when the Hilbert space is restricted to such states $i.e.$ the Hamiltonian is a sum of commuting operators. There is a unique ground state on a torus and it is an eigenstate of both vertex and plaquette operators with eigenvalues $B_v \equiv +1$ and $B_p \equiv -1$.

In Fig. 10, the relative amplitudes for different loop configurations with respect to the vacuum configuration are denoted as $\alpha$, $\beta$, $\gamma$ and so on. For the semion WW, the amplitude $\delta$ is given by $\iota$ because of the twist as shown and the configurations with single loops have amplitudes $\alpha = \beta = -1$.

The point excitations in the bulk of the semion WW are confined because a pair of such excitations cannot be separated without creating plaquette excitations along the path of separation. This is consistent with the facts that the point excitations are semionic and in three dimensions, deconfined excitations have only bosonic or fermionic statistics. Secondly, any deconfined topological excitation will have a string operator that can wrap around the torus. Since the semion WW does not have such excitations, it has a unique ground state on a torus.

We refer the reader to Ref. [49] for more details on the semion Walker-Wang model.

## Appendix F: Higher-form and higher group symmetries

### 1. Higher form symmetries

We start with the conventional global symmetry referred to as the 0-form symmetries and then later generalize to "Generalized global symmetries" or higher form symmetries.

Under the 0-form global symmetry, the charged objects are local or point-like operators $\mathcal{O}(\vec{x})$ and it is a codimension-1 surface that encloses or links such a local operator. Such a codimension-1 surface is topological which implies that the enclosed charge is invariant under its deformation as long as it doesn't intersect with other topological surfaces with enclosed charges.

For the 1-form symmetry, the charged objects are line-like operators and it is a topological codimension-2 surfaces that links such a charged object. In general, for an $n$-form symmetry, the charged objects are $n$-dimensional and it is a topological surface of codimension $n+1$ that links such a charged object.

There is a conserved $n+1$-form current associated with an $n$-form symmetry. For example, for a continuous 0-form $U(1)$ symmetry $U(1)^{(0)}$, where superscript $(0)$ is for 0-form, there is a conserved current $j^{(1)} = j_\mu dx^\mu$ such that $\partial^\mu j_\mu \equiv d \star j^{(1)} = 0$. We used the notation involving exterior derivative and Hodge star that was discussed in appendix A. In $\mathbb{R}^n$, we have the charged operator under $U(1)^{(0)}$ found by integrating the current $i.e.$ $Q(\Sigma_1) = \int_{\Sigma_1} \star j^{(1)}$ where $\Sigma_1$ is a codimension-1 topological surface. Now we consider a continuous $U(1)$ 1-form global symmetry $U(1)^{(1)}$. There is an associated 2-form conserved current $j^{(2)} \equiv j_{[\mu,\nu]}$ such that $d \star j^{(2)} = 0$. The associated charge is given by integrating the 2-form current over a codimension-2 topological surface i.e $Q(\Sigma_2) = \int_{\Sigma_2} \star j^{(2)}$.

We couple the $n+1$-form current, associated with the $n$-form symmetry, to an $n+1$-form non-dynamical or classical background gauge field as a following change in the action $S$,

$$\triangle S = \int A^{(n+1)} \wedge \star j^{(n+1)}. \tag{F1}$$

Current conservation $d \star j^{(n+1)} = 0$ implies that the partition function $Z[A^{(n+1)}]$ is invariant under the gauge transformation $A^{(n+1)} \to A^{(n+1)} + d\lambda^{(n)}$ where $\lambda^{(n)}$ is an $n$-form gauge parameter.

## 2.   2-group and 3-group symmetries

Consider a theory with both 0-form and 1-form symmetries. We consider a linear coupling of the associated currents $j^{(1)}$ and $j^{(2)}$ to the background gauge fields $A^{(1)}$ and $A^{(2)}$ respectively as follows,

$$\triangle S = \int A^{(1)} \wedge \star j^{(1)} + A^{(2)} \wedge \star j^{(2)}. \tag{F2}$$

For some theories, the gauge transformations for the background fields to keep the partition function $Z[A^{(1)}, A^{(2)}]$ invariant are given by

$$A^{(1)} \to A^{(1)} + d\lambda^{(0)} \tag{F3}$$

$$A^{(2)} \to A^{(2)} + d\lambda^{(1)}, \tag{F4}$$

where $\lambda^{(0)}$ and $\lambda^{(1)}$ are 0-form and 1-form gauge parameters. Sometimes when the 0-form and 1-form symmetries communicate in an intricate way, such gauge transformations for the symmetries are not sufficient to keep the partition function invariant.

$(A^{(1)}, A^{(2)})$ are said to form a 2-connection on a 2-group bundle when the gauge transformations that keep the partition function invariant up to c-number anomalies are given as follows

$$A^{(1)} \to A^{(1)} + d\lambda^{(0)} \tag{F5}$$

$$A^{(2)} \to A^{(2)} + d\lambda^{(1)} + \frac{\kappa}{2\pi}\lambda^{(0)}dA^{(1)} \tag{F6}$$

$$\tag{F7}$$

where $\kappa$ is quantized. When $\kappa = 0$, we recover the standard gauge transformations eqns (F3) and (F4). In the case of a discrete 2-group symmetry $G$, such background gauge transformations generalize with $\kappa$ belonging to a certain cohomology class $H^3_\rho(BG, G)$, twisted via $\rho$ which can be considered as an action of the 0-form on the 1-form symmetry. In the next subsection, we discuss $2 + 1$d TQFTs which form a class of examples of theories with a discrete 2-group symmetry.

We now consider the generalization to the 3-group symmetry. Instead of just 0-form and 1-form, we have the 0-form, 1-form and 2-form symmetries. We couple the associated currents $j^{(1)}$, $j^{(2)}$ and $j^{(3)}$ to the background gauge fields $A^{(1)}$, $A^{(2)}$ and $A^{(3)}$. The linear coupling looks as follows,

$$\triangle S = \int A^{(1)} \wedge \star j^{(1)} + A^{(2)} \wedge \star j^{(2)} + A^{(3)} \wedge \star j^{(3)}. \tag{F8}$$

In some theories, for example, we may need the following gauge transformations to keep the partition function $Z[A^{(1)}, A^{(2)}, A^{(3)}]$ invariant,

$$A^{(1)} \to A^{(1)} + d\lambda^{(0)} \tag{F9}$$

$$A^{(2)} \to A^{(2)} + d\lambda^{(1)} \tag{F10}$$

$$A^{(3)} \to A^{(3)} + d\lambda^{(2)} + \frac{\kappa}{2\pi}\lambda^{(0)}dA^{(2)}. \tag{F11}$$

In such cases, the $(A^{(1)}, A^{(2)}, A^{(3)})$ are said to form a 3-connection on a 3-group bundle.

### 3. 2+1d TQFTs and discrete 2-group symmetries

Topological Quantum Field theories (TQFTs) with a global symmetry form a general class of examples that have a higher group symmetry, in particular, the 2-group symmetry. In the literature on topological phases, this is more commonly known as the obstruction to symmetry fractionalization or the $H^3$ anomaly [15]. Without the introduction of a global symmetry, a TQFT is described by a unitary modular tensor category $\mathcal{C}$ containing certain line operators or anyon worldlines as the objects. These line operators form a finite set of observables and obey a commutative fusion algebra

$$a \times b = \sum_{c \in \mathcal{C}} N_{ab}^c c. \tag{F12}$$

where $N_{ab}^c = N_{ba}^c$ are non-negative integers. They obey an associativity rule $\sum_e N_{ab}^e N_{ec}^d = \sum_f N_{af}^d N_{bc}^f$. Here, $N_{ab}^c$ is the dimension of the vector space $V_{ab}^c$ associated with the set of trivalent vertices specified by the line operators or anyons $a$, $b$ and $c$. We refer the reader to Refs [15] for a review of the topological data associated with the UMTC. Here, we just specify that the topological data includes the quantum dimensions of anyons $d_a$ where $a \in \mathcal{C}$, S-matrix $S_{ab}$, topological spins $\theta_a$ and the $F$ and $R$ symbols. Given the topological data, we identify the intrinsic global symmetry of the 3d TQFTs. There exists a 0-form symmetry group of the TQFT. In order to define it, we first identify a set of invertible maps $\varphi : \mathcal{C} \to \mathcal{C}$ that preserve the topological data specified above. These act as permutation on the anyons in $\mathcal{C}$ up to a unitary action on the vector spaces $V_{ab}^c$. These transformations are called topological symmetries. There is a subset of such transformations in this 0-form symmetry group that do not permute the anyons and the action on a state in the vector space $V_{ab}^c$ is only an overall phase. This subset is called natural isomorphisms. It is conjectured that any topological symmetry that does not permute anyons is a natural isomorphism [15]. The 0-form symmetry group is the automorphism group of the category $\mathcal{C}$, $\mathrm{Aut}(\mathcal{C})$; in particular, it is the set of topological symmetries modulo the natural isomorphisms. There is also an intrinsic 1-form symmetry of the TQFT, generated by the worldlines of the subset of anyons in $\mathcal{C}$ that are abelian. The action of the intrinsic 0-form symmetry on the intrinsic 1-form symmetry is given by the permutation action of the elements of the automorphism group

Aut($\mathcal{C}$) on the set of abelian anyons. This leads to an intrinsic 2-group symmetry of the theory, meaning of which will be made clear below. In general, we also consider an extrinsic 2-group given by $(G_e, A_e)$ and couple the TQFT to it.

Consider a homomorphism from 0-form $G$ to Aut($\mathcal{C}$), $[\rho] : G \to$ Aut($\mathcal{C}$) such that $[\rho_g][\rho_h] = [\rho_{gh}]$ where $g, h \in G$. For $G =$ Aut($\mathcal{C}$), the intrinsic 0-form symmetry of the theory, we have the case $[\rho] = id$. Also, the action of $G$ on the 1-form symmetry group is given as $\rho : G \to$ Aut($\mathcal{A}$) where we restrict to the set of abelian anyons $\mathcal{A}$.

The action of $\rho_g$ on anyons $a$ is given as $\rho_g(a) = a' \in \mathcal{C}$ and on the state $|a, b; c\rangle \in V_{ab}^c$ is given by $\rho_g |a, b; c\rangle = U_g(a', b', c') |a', b'; c'\rangle$ where $U_g$ are unitary matrices. The group homomorphism classes $[\rho_g]$ obey $[\rho_g][\rho_h] = [\rho_{gh}]$ but $\rho_g$ has to be a group homomorphism only up to a natural isomorphism, $\kappa_{gh}$, $\kappa_{gh}\rho_g\rho_h = \rho_{gh}$. The natural isomorphism $\kappa_{gh}$ is dependent on the state $|a, b; c\rangle$ on which the elements $\rho_g$ and $\rho_h$ act and is given by $\kappa_{gh}(a, b; c)_{\mu\nu} = \frac{\eta_a(g,h)\eta_b(g,h)}{\eta_c(g,h)}\delta_{\mu\nu}$ for some phases $\eta_a(g, h), \eta_b(g, h), \eta_c(g, h)$. Using associativity of $\rho$, one obtains

$$\Omega_a(g, h, k)\Omega_b(g, h, k) = \Omega_c(g, h, k) \tag{F13}$$

where $\Omega_a(g, h, k) = \frac{\eta_{\rho_g^{-1}(a)}(h,k)\eta_a(g,hk)}{\eta_a(gh,k)\eta_a(g,h)}$. This gives $\Omega_a(g, h, k) = M_{a\beta(g,h,k)}^\star$ where $\beta(g, h, k) \in \mathcal{A}$ is an abelian anyon and $M_{ab}^\star = \frac{S_{ab}^\star S_{00}}{S_{0a}S_{0b}}$ is the monodromy scalar component defined in terms of S-matrix elements $S_{ab}$. Here, the abelian anyon $\beta(g, h, k) \in \mathcal{A}$ is also a twisted cocycle in $Z_\rho^3(BG, \mathcal{A})$ and different $\eta_a$ solutions for the same natural isomorphisms lead to a $\beta$ differing by an exact cocycle. Hence, we define an equivalence class $[\beta] \in H_\rho^3(BG, \mathcal{A})$.

Thus, from the action of the 0-form symmetry on the 1-form symmetry, we get a twisted cohomology class, $[\beta] \in H_\rho^3(BG, \mathcal{A})$ which is what was referred to as the obstruction to symmetry fractionalization or the $H^3$ anomaly above. The full 2-group symmetry of the TQFT is written as $\mathcal{G} = (G, \mathcal{A}, \rho, [\beta])$ where $[\beta]$ is more generally referred to as the Postnikov class.