# Peer review of "Loops in 4+1d Topological Phases"

_SciPost Physics_

## Round 1 · Referee Report · Anonymous (Referee 1) · 2022-2-7

Strengths

1. The results are mathematically elegant.

Weaknesses

1. The authors do not provide an adequate review of where this work fits into the literature.
2. The writing is of uneven quality, especially in the latter parts of the paper (e.g., sec VI).

Report

Overall, the parts that I understand in this paper appear correct. The paper is unfortunately not clearly written. The explanations presented are often rushed, especially so for the most technically difficult arguments. Not enough trouble has been taken to put the work in context within the field.

I'll try to summarise my understanding, and list questions as they arise. This paper examines a family of 4+1 D TQFTs with ZN gauge group and two 2 form fields L=N2πb(1)db(2).

The model thus described has loop-like excitations of two types: magnetic and electric. The authors first focus on the N=2 case, and consider bound states of electric and magnetic excitations. They show that, whereas such em bound states are fermionic in 2+1D, they have no universal non-trivial exchange statistic in 4+1D. The key difference is that the bilinear form αdβ is symmetric for two 1-forms in 2+1 D, but antisymmetric for two 2-forms in 4+1 D.

The authors note that the model has a number of different possible gapped boundary conditions corresponding to different 3+1D Walker-Wang models. You can condensed e loops, m loops, and dyonic psi loops. In the former two cases, one obtains 3+1 D toric code (or fermionic toric code models), in the last case one obtains a 3DSem model.

Already at this point, I have some serious confusions. Below Eq 23 the authors say:

"The boundary is Z2 2-form gauge theory .... describes a Z2 topological order with confined charges"

I'm confused: The 3+1 D toric code has **deconfined** charges! It's possible I've misunderstood the construction, but in any case the text is very confusing without further elaboration. Relatedly, just above eq 36 the authors say:

"Similar to the e condensed and m condensed boundaries, there are two variants of the ψ condensed boundary, corresponding to whether the particle in the boundary Z2 topological
order is a boson or fermion"

Again I'm confused: The 3D semion model has no deconfined particles, so it's meaningless to talk about exchange statistics of the particle. Again, the text is confusing.

Continuing through the paper. Related to these different boundary conditions, the paper notes the apparent SL(2,Z2) symmetry of the model, which can also be associated with permutations of the e,m, and psi loop excitations. The authors provide explicit unitary maps on an explicit lattice model implementing said automorphisms of the excitations. I don't have any problems with this section (IV).

Sec V and VI are very difficult to follow, and appear rather rushed.

To begin:

"As discussed for the Z2 gauge theory, this symmetry is non-anomalous on orientable spacetime, where the integral of total derivative vanishes"

I've searched the text, and cannot find the discussion of anomalies in the Z_2 case prior to the quoted sentence.

Continuing, I'm confused as to exactly what role time reversal symmetry is playing in the remainder of this paper. Does the H^4 obstruction only exist only when time reversal is imposed? I strongly suspect not, but the text above (63) suggests otherwise (or is a distracting aside whose relevance should be clarified).

Focussing on Sec VI: My understanding of the obstruction is that it captures the fact you cannot proliferate D^N defects in the N even case. Correct? I'm less clear as to why you cannot proliferate D^N defects/gauge mψ symmetry. Some questions:

Where does the final equality in (76) come from? Citation needed.

"Thus we find that the domain wall D^N is completely equivalent to insertion of the electric surface operator at a suitable
locus in any correlation function."
Why is this important? Does this prove there's an obstruction? Why?

I think 77-79 form the proof that you cannot gauge mψ symmetry. But I do not understand the construction in (77). Where does this come from? The authors need to spend more time explaining where (77) comes from, and why it corresponds to an attempt to gauge mψ symmetry.

Requested changes

1. The text should be modified to address each of the questions listed above/head-off the confusion described. Particularly sec VI, which will likely require a major rewriting for clarity.
2. The authors cite no other works in the introduction. This must be fixed, and the relation of the manuscript to existing work clarified.
3. Below (69) "am" -> "an"
4. Repeated reference [20] and [36].
5. Below (73) aciton-> action

  • validity: good
  • significance: good
  • originality: good
  • clarity: low
  • formatting: acceptable
  • grammar: reasonable

Author:  Arpit Dua  on 2022-11-01  [id 2971]

(in reply to Report 1 on 2022-02-07)
Category:
answer to question

Dear Editor and Referee,

Please have a look at the attached pdf in which we reply to the referee's comments and summarize the changes made in the manuscript.

Thanks,
Authors

Attachment:

reply.pdf

---

## Editorial Decision

resubmitted